# FedDR – Randomized Douglas-Rachford Splitting Algorithms for Nonconvex Federated Composite Optimization

**Quoc Tran-Dinh** and **Nhan H. Pham**
Department of Statistics and Operations Research, The University of North Carolina at Chapel Hill
318 Hanes Hall, UNC-Chapel Hill, NC 27599-3260
quoctd@email.unc.edu, nhanph@live.unc.edu

**Dzung T. Phan** and **Lam M. Nguyen**
IBM Research, Thomas J. Watson Research Center, Yorktown Heights, NY, USA.
phandu@us.ibm.com, lamnguyen.mltd@ibm.com

## Abstract

We develop two new algorithms, called, **FedDR** and **asyncFedDR**, for solving a fundamental nonconvex composite optimization problem in federated learning. Our algorithms rely on a novel combination between a nonconvex Douglas-Rachford splitting method, randomized block-coordinate strategies, and asynchronous implementation. They can also handle convex regularizers. Unlike recent methods in the literature, e.g., FedSplit and FedPD, our algorithms update only a subset of users at each communication round, and possibly in an asynchronous manner, making them more practical. These new algorithms can handle statistical and system heterogeneity, which are the two main challenges in federated learning, while achieving the best known communication complexity. In fact, our new algorithms match the communication complexity lower bound up to a constant factor under standard assumptions. Our numerical experiments illustrate the advantages of our methods over existing algorithms on synthetic and real datasets.

## 1 Introduction

Training machine learning models in a centralized fashion becomes more challenging and marginally inaccessible for a large number of users, especially when the size of datasets and models is growing substantially larger. Consequently, training algorithms using decentralized and distributed approaches comes in as a natural replacement. Among several approaches, federated learning (FL) has received tremendous attention in the past few years since it was first introduced in [18, 30]. In this setting, a central server coordinates between many local users (also called agents or devices) to perform their local updates, then the global model will get updated, e.g., by averaging or aggregating local models.

**Challenges.** FL provides a promising solution for many machine learning applications such as learning over smartphones or across organizations, and internet of things, where privacy protection is one of the most critical requirements. However, this training mechanism faces a number of fundamental challenges, see, e.g., [31]. First, when the number of users gets substantially large, it creates *communication bottleneck* during model exchange process between server and users. Second, the local data stored in each local user may be different in terms of sizes and distribution which poses a challenge: *data or statistical heterogeneity*. Third, the variety of users with different local storage, computational power, and network connectivity participating into the system also creates a major challenge, known as *system heterogeneity*. This challenge also causes unstable connection

35th Conference on Neural Information Processing Systems (NeurIPS 2021).

between server and users, where some users may be disconnected from the server or simply dropped out during training. In practice, we can expect only a subset of users to participate in each round of communication. Another challenge in FL is *privacy concern*. Accessing and sharing local raw data is not permitted in FL. In addition, distributed methods exchange the objective gradient of local users, and private data can be exposed from the shared model such as the objective gradients [51]. Therefore, FL methods normally send the global model to each user at the start of each communication round, each user will perform its local update and send back only the necessary update for aggregation.

**Our goal and approach.** Our goal in this paper is to further and simultaneously address these fundamental challenges by proposing two new algorithms to train the underlying common optimization model in FL. Our approach relies on a novel combination between randomized block-coordinate strategy, nonconvex Douglas-Rachford (DR) splitting, and asynchronous implementation. While each individual technique or partial combinations is not new, our combination of three as in this paper appears to be the first in the literature. To the best of our knowledge, this is the first work developing randomized block-coordinate DR splitting methods for nonconvex composite FL optimization models, and they are fundamentally different from some works in the convex setting, e.g., [7, 8].

**Contribution.** Our contribution can be summarized as follows.

(a) We develop a new FL algorithm, called **FedDR** (**Fed**erated **D**ouglas-**R**achford), by combining the well-known DR splitting technique and randomized block-coordinate strategy for the common nonconvex composite optimization problem in FL. Our algorithm can handle nonsmooth convex regularizers and allows inexact evaluation of the underlying proximal operators as in FedProx or FedPD. It also achieves the best known $\mathcal{O}\left(\varepsilon^{-2}\right)$ communication complexity for finding a stationary point under standard assumptions (Assumptions 2.1-2.2), where $\varepsilon$ is a given accuracy. More importantly, unlike FedSplit [33] and FedPD [49], which require full user participation to achieve convergence, our analysis does allow partial participation by selecting a subset of users to perform update at each communication round.

(b) Next, we propose an asynchronous algorithm, **asyncFedDR**, where each user can asynchronously perform local update and periodically send the update to the server for proximal aggregation. We show that **asyncFedDR** achieves the same communication complexity $\mathcal{O}\left(\varepsilon^{-2}\right)$ as **FedDR** (up to a constant factor) under the same standard assumptions. This algorithm is expected to simultaneously address all challenges discussed above.

Let us emphasize some key points of our contribution. First, the best known $\mathcal{O}\left(\varepsilon^{-2}\right)$ communication complexity of our methods matches the lower bound complexity up to a constant factor as shown in [49], even with inexact evaluation of the objective proximal operators. Second, our methods rely on a DR splitting technique for nonconvex optimization and can handle possibly nonsmooth convex regularizers, which allows us to deal with a larger class of applications and with constraints [47]. Furthermore, it can also handle both statistical and system heterogeneity as discussed in FedSplit [33] and FedPD [49]. However, FedSplit only considers the convex case, and both FedSplit and FedPD require all users to update at each communication round, making them less practical and applicable in FL. Our methods only require a subset of users or even one user to participate in each communication round as in FedAvg or FedProx. In addition, our aggregation step on the server is different from most existing works due to a proximal step on the regularizer. It is also different from [47]. Third, as FedProx [23], we allow inexact evaluation of users' proximal operators with any local solver (e.g., local SGD or variance reduced methods) and with adaptive accuracies. Finally, requiring synchronous aggregation at the end of each communication round may lead to slow-down in training due to the heterogeneity in computing power and communication capability of local users. It is natural to have asynchronous update from local users as in, e.g., [34, 35, 39]. Our asynchronous variant, **asyncFedDR**, can fairly address this challenge. Moreover, it uses a general probabilistic model recently introduced in [5], which allows us to capture the variety of asynchronous environments and architectures compared to existing methods, e.g., [39, 44].

**Related work and comparison.** Federated Averaging (FedAvg) is perhaps the earliest method used in FL. In FedAvg, users perform stochastic gradient descent (SGD) updates for a number of epochs then send updated models to server for aggregation. FedAvg's practical performance has been shown in many early works, e.g., [18, 29, 48] and tends to become the most popular method for solving FL applications. [26] show that local SGD where users perform a number of local updates before global communication takes place as in FedAvg may offer benefit over minibatch SGD. Similar comparison between minibatch SGD and local SGD has been done in [42, 43]. Analyzing convergence of FedAvg

was very challenging at its early time due to the complexity in its update as well as data heterogeneity. One of the early attempt to show the convergence of FedAvg is in [39] for convex problems under the iid data setting and a set of assumptions. [45] also considers local SGD in the nonconvex setting. Without using an additional bounded gradient assumption as in [39, 45], [41] improves the complexity for the general nonconvex setting while [11] uses a Polyak-Łojasiewicz (PL) condition to improve FedAvg's convergence results. In heterogeneous data settings, [17] analyzes local GD, where users performs gradient descent (GD) updates instead of SGD. The analysis of FedAvg for non-iid data is given in [24]. The analysis of local GD/SGD for nonconvex problems has been studied in [13]. However, FedAvg might not converge with non-iid data as shown in [33, 49, 50].

FedProx [23] is an extension of FedAvg, which deals with heterogeneity in federated networks by introducing a proximal term to the objective in local updates to improve stability. FedProx has been shown to achieve better performance than FedAvg in heterogeneous setting. Another method to deal with data heterogeneity is SCAFFOLD [16] which uses a control variate to correct the "client-drift" in local update of FedAvg. MIME [15] is another framework that uses control variate to improve FedAvg for heterogeneous settings. However, SCAFFOLD and MIME require to communicate extra information apart from local models. Compared to aforementioned works, our methods deal with nonconvex problems under standard assumptions and with composite settings.

FedSplit [33] instead employs a Peaceman-Rachford splitting scheme to solve a constrained reformulation of the original problem. In fact, FedSplit can be viewed as a variant of Tseng's splitting scheme [1] applied to FL. [33] show that FedSplit can find a solution of the FL problem under only convexity without imposing any additional assumptions on system or data homogeneity. [49] proposes FedPD, which is essentially a variant of the standard augmented Lagrangian method in nonlinear optimization. Other algorithms for FL can be found, e.g., in [6, 10, 12, 14, 25, 46].

Our approach in this paper relies on nonconvex DR splitting method, which can handle the heterogeneity as discussed in [33]. While the DR method is classical, its nonconvex variants have been recently studied e.g., in [9, 21, 40]. However, the combination of DR and randomized block-coordinate strategy remains limited [7, 8] even in the convex settings. Alternatively, asynchronous algorithms have been extensively studied in the literature, also for FL, see, e.g., [2, 34, 35]. For instance, a recent work [44] analyzes an asynchronous variant of FedAvg under bounded delay assumption and constraint on the number of local updates. [39] proposes an asynchronous local SGD to solve convex problems under iid data. However, to our best knowledge, there exists no asynchronous method using DR splitting techniques with convergence guarantee for FL. In addition, most existing algorithms only focus on non-composite settings. Hence, our work here appears to be the first.

**Content.** The rest of this paper is organized as follows. Section 2 states our FL optimization model and our assumptions. Section 3 develops **FedDR** and analyzes its convergence. Section 4 considers an asynchronous variant, **asyncFedDR**. Section 5 is devoted for numerical experiments. Due to space limit, all technical details and proofs can be found in Supplementary Document (Supp. Doc.).

## 2 Nonconvex Optimization Models in Federated Learning

The underlying optimization model of many FL applications can be written into the following form:

$$\min_{x \in \mathbb{R}^p} \left\{ F(x) := f(x) + g(x) = \frac{1}{n} \sum_{i=1}^{n} f_i(x) + g(x) \right\}, \tag{1}$$

where $n$ is the number of users, and each $f_i$ is a local loss of the $i$-th user, which is assumed to be nonconvex and $L$-smooth (see Assumptions 2.1 and 2.2 below), and $g$ is a proper, closed, and convex regularizer. Apart from these assumptions, we will not make any additional assumption on (1). We emphasize that the use of regularizers $g$ has been motivated in several works, including [47].

Let $\text{dom}(F) := \{x \in \mathbb{R}^p : F(x) < +\infty\}$ be the domain of $F$ and $\partial g$ be the subdifferential of $g$ [1]. Since (1) is nonconvex, we only expect to find a stationary point, which is characterized by the following optimality condition.

**Definition 2.1.** If $0 \in \nabla f(x^*) + \partial g(x^*)$, then $x^*$ is called a [first-order] stationary point of (1).

The algorithms for solving (1) developed in this paper will rely on the following assumptions.

**Assumption 2.1** (Boundedness from below). $\text{dom}(F) \neq \emptyset$ and $F^\star := \inf_{x \in \mathbb{R}^p} F(x) > -\infty$.

**Assumption 2.2** (*L*-smoothness)**.** All functions $f_i(\cdot)$ for $i \in [n] := \{1, \cdots, n\}$ are *L*-smooth, i.e., $f_i$ is continuously differentiable and there exists $L \in (0, +\infty)$ such that

$$\|\nabla f_i(x) - \nabla f_i(y)\| \le L\|x - y\|, \quad \forall x, y \in \text{dom}(f_i). \tag{2}$$

Assumptions 2.1 and 2.2 are very standard in nonconvex optimization. Assumption 2.1 guarantees the well-definedness of (1) and is independent of algorithms. Assuming the same Lipschitz constant $L$ for all $f_i$ is not restrictive since if $f_i$ is $L_i$-smooth, then by scaling variables of its constrained formulation (see (11) in Supp. Doc.), we can get the same Lipschitz constant $L$ of all $f_i$.

**Proximal operators and evaluation.** Our methods make use of the proximal operators of both $f_i$ and $g$. Although $f_i$ is *L*-smooth and nonconvex, we still define its proximal operator as

$$\text{prox}_{\eta f_i}(x) := \arg\min_y \left\{ f_i(y) + \tfrac{1}{2\eta}\|y - x\|^2 \right\}, \tag{3}$$

where $\eta > 0$. Even $f_i$ is nonconvex, under Assumption 2.2, if we choose $0 < \eta < \frac{1}{L}$, then $\text{prox}_{\eta f_i}$ is well-defined and single-valued. Evaluating $\text{prox}_{\eta f_i}$ requires to solve a strongly convex program. If $\text{prox}_{\eta f_i}$ can only be computed approximately up to an accuracy $\epsilon \ge 0$ to obtain $z$, denoted by $x_+ :\approx \text{prox}_{\eta f_i}(x)$, if $\|x_+ - \text{prox}_{\eta f_i}(x)\| \le \epsilon_i$. Note that instead of absolute error, one can also use a relative error as $\|x_+ - \text{prox}_{\eta f_i}(x)\| \le \epsilon_i\|x_+ - x\|$ as in [37]. For the convex function $g$, its proximal operator $\text{prox}_{\eta g}$ is defined in the same way as (3). Evaluating $\text{prox}_{\eta f_i}$ can be done by various existing methods, including local SGD and accelerated GD-type algorithms. However, this is not our focus in this paper, and therefore we do not specify the subsolver for evaluating $\text{prox}_{\eta f_i}$.

**Gradient mapping.** As usual, let us define the following gradient mapping of $F$ in (1).

$$\mathcal{G}_\eta(x) := \tfrac{1}{\eta}\big(x - \text{prox}_{\eta g}(x - \eta \nabla f(x))\big), \quad \eta > 0. \tag{4}$$

Then, the optimality condition $0 \in \nabla f(x^*) + \partial g(x^*)$ of (1) is equivalent to $\mathcal{G}_\eta(x^*) = 0$. However, in practice, we often wish to find an $\varepsilon$-approximate stationary point to (1) defined as follows.

**Definition 2.2.** If $\tilde{x} \in \text{dom}(F)$ satisfies $\mathbb{E}\big[\|\mathcal{G}_\eta(\tilde{x})\|^2\big] \le \varepsilon^2$, then $\tilde{x}$ is called an $\varepsilon$-stationary point of (1), where the expectation is taken overall the randomness generated by the underlying algorithm.

Note that, for $\mathcal{G}_\eta(\tilde{x})$ to be well-defined, we require $\tilde{x} \in \text{dom}(F)$. In our algorithms below, this requirement is fulfilled if $\tilde{x} \in \text{dom}(f)$, which is often satisfied in practice as $\text{dom}(f) = \mathbb{R}^p$.

## 3 FedDR Algorithm and Its Convergence Guarantee

Prior to our work, FedSplit [33] exploits similar update steps as ours by adopting the Peaceman-Rachford splitting method to solve the convex and non-composite instances of (1). FedSplit can overcome some of the key challenges as discussed earlier. Following this idea, we take the advantages of the DR splitting method to first derive a new variant to handle the nonconvex composite problem (1). This new algorithm is synchronous and we call it **FedDR**. The central idea is as follows: First, we reformulate (1) into (12) by duplicating variables. Next, we apply a DR splitting scheme to the resulting problem. Finally, we combine such a scheme with a randomized block-coordinate strategy.

The complete algorithm is presented in Algorithm 1, where its full derivation is in Supp. Doc. A.1.

Let us make the following remarks. Firstly, **FedDR** mainly updates of three sequences $\{\bar{x}^k\}$, $\{x_i^k\}$ and $\{y_i^k\}$. While $\bar{x}^k$ is an averaged model to approximately minimize the global objective function $F$, $x_i^k$ act as local models trying to optimize a regularized local loss function w.r.t. its local data distribution, and $y_i^k$ keeps track of the residuals from the local models to the global one. Secondly, we allow $x_i^k$ to be an approximation of $\text{prox}_{\eta f_i}(y_i^k)$ up to an accuracy $\epsilon_{i,k} \ge 0$ as defined in (3), i.e., $\|x_i^k - \text{prox}_{\eta f_i}(y_i^k)\| \le \epsilon_{i,k}$ for all $i \in [n]$ if $k = 0$ and for all $i \in \mathcal{S}_{k-1}$ if $k > 0$. If $\epsilon_{i,k} = 0$, then we get the exact evaluation $x_i^k := \text{prox}_{\eta f_i}(y_i^k)$. Approximately evaluating $\text{prox}_{\eta f_i}$ can be done, e.g., by local SGD as in FedAvg. Thirdly, Algorithm 1 is different from existing randomized proximal gradient-based methods since we rely on a DR splitting scheme and can handle composite settings. Here, three iterates $y_i^k$, $x_i^k$, and $\hat{x}_i^k$ at Step 5 are updated sequentially, making it challenging to analyze convergence. Lastly, the subset of active users $\mathcal{S}_k$ is sampled from a random set-valued mapping $\hat{\mathcal{S}}$. As specified in Assumption 3.1, this sampling mechanism covers a wide range of sampling strategies. Clearly, if $\mathcal{S}_k = [n]$ and $g = 0$, then Algorithm 1 reduces to FedSplit, but for the nonconvex case. Hence, our convergence guarantee below remains applicable, and the guarantee is sure. Note that both our model (1) and Algorithm 1 are completely different from [47].

---

**Algorithm 1** (FL with Randomized DR (**FedDR**))

---

1: **Initialization:** Take $x^0 \in \operatorname{dom}(F)$. Choose $\eta > 0$ and $\alpha > 0$, and accuracies $\epsilon_{i,0} \geq 0$ ($i \in [n]$).
   Initialize the server with $\bar{x}^0 := x^0$ and $\tilde{x}^0 := x^0$.
   Initialize each user $i \in [n]$ with $y_i^0 := x^0$, $x_i^0 :\approx \operatorname{prox}_{\eta f_i}(y_i^0)$, and $\hat{x}_i^0 := 2x_i^0 - y_i^0$.
2: **For** $k := 0, \cdots, K$ **do**
3:    [*Active users*] Generate a proper realization $\mathcal{S}_k \subseteq [n]$ of $\hat{\mathcal{S}}$ (see Assumption 3.1).
4:    [*Communication*] Each user $i \in \mathcal{S}_k$ receives $\bar{x}^k$ from the server.
5:    [*Local update*] **For each user** $i \in \mathcal{S}_k$ **do**: Choose $\epsilon_{i,k+1} \geq 0$ and update

$$y_i^{k+1} := y_i^k + \alpha(\bar{x}^k - x_i^k), \quad x_i^{k+1} :\approx \operatorname{prox}_{\eta f_i}(y_i^{k+1}), \quad \text{and} \quad \hat{x}_i^{k+1} := 2x_i^{k+1} - y_i^{k+1}.$$

6:    [*Communication*] Each user $i \in \mathcal{S}_k$ sends $\Delta \hat{x}_i^k := \hat{x}_i^{k+1} - \hat{x}_i^k$ back to the server.
7:    [*Sever aggregation*] The server aggregates $\tilde{x}^{k+1} := \tilde{x}^k + \frac{1}{n} \sum_{i \in \mathcal{S}_k} \Delta \hat{x}_i^k$.
8:    [*Sever update*] Then, the sever updates $\bar{x}^{k+1} := \operatorname{prox}_{\eta g}(\tilde{x}^{k+1})$.
9: **End For**

---

### 3.1 Convergence of Algorithm 1

Let us consider a proper sampling scheme $\hat{\mathcal{S}}$ of $[n]$, which is a random set-valued mapping with values in $2^{[n]}$, the collection of all subsets of $[n]$. Let $\mathcal{S}_k$ be an iid realization of $\hat{\mathcal{S}}$ and $\mathcal{F}_k := \sigma(\mathcal{S}_0, \cdots, \mathcal{S}_k)$ be the $\sigma$-algebra generated by $\mathcal{S}_0, \cdots, \mathcal{S}_k$. We first impose the following assumption about the distribution of our sampling scheme $\hat{\mathcal{S}}$.

**Assumption 3.1.** There exist $\mathbf{p}_1, \cdots, \mathbf{p}_n > 0$ such that $\mathbb{P}(i \in \hat{\mathcal{S}}) = \mathbf{p}_i > 0$ for all $i \in [n]$.

This assumption covers a large class of sampling schemes as discussed in [36], including non-overlapping uniform and doubly uniform. This assumption guarantees that every user has a non-negligible probability to be updated. Note that $\mathbf{p}_i = \sum_{\mathcal{S}:i \in \mathcal{S}} \mathbb{P}(\mathcal{S})$ due to Assumption 3.1. For the sake of notation, we also denote $\hat{\mathbf{p}} := \min\{\mathbf{p}_i : i \in [n]\} > 0$.

The following theorem characterizes convergence of Algorithm 1 with inexact evaluation of $\operatorname{prox}_{\eta f_i}$. Due to space limit, we refer the reader to Lemma A.6 in Sup. Doc. for more details about the choice of stepsizes and related constants. The proof of this theorem is defered to Sup. Doc. A.5.

**Theorem 3.1.** *Suppose that Assumptions 2.1, 2.2, and 3.1 hold. Let $\{(x_i^k, y_i^k, \hat{x}_i^k, \bar{x}^k)\}$ be generated by Algorithm 1 using stepsizes $\alpha$ and $\eta$ defined in* (33). *Then, the following holds*

$$\frac{1}{K+1} \sum_{k=0}^{K} \mathbb{E}\left[\|\mathcal{G}_\eta(\bar{x}^k)\|^2\right] \leq \frac{C_1[F(x^0) - F^\star]}{K+1} + \frac{1}{n(K+1)} \sum_{k=0}^{K} \sum_{i=1}^{n} \left(C_2 \epsilon_{i,k}^2 + C_3 \epsilon_{i,k+1}^2\right), \quad (5)$$

*where $\beta$, $\rho_1$, and $\rho_2$ are explicitly defined by* (35)*, and*

$$C_1 := \frac{2(1+\eta L)^2(1+\gamma_2)}{\eta^2 \beta}, \quad C_2 := \rho_1 C_1, \quad \text{and} \quad C_3 := \rho_2 C_1 + \frac{(1+\eta L)^2(1+\gamma_2)}{\eta^2 \gamma_2}.$$

*Let $\tilde{x}^K$ be selected uniformly at random from $\{\bar{x}^0, \cdots, \bar{x}^K\}$ as the output of Algorithm 1. Let the accuracies $\epsilon_{i,k}$ for all $i \in [n]$ and $k \geq 0$ at Step 5 be chosen such that $\frac{1}{n} \sum_{i=1}^{n} \sum_{k=0}^{K+1} \epsilon_{i,k}^2 \leq M$ for a given constant $M > 0$ and all $K \geq 0$. Then, if we run Algorithm 1 for at most*

$$K := \left\lfloor \frac{C_1[F(x^0) - F^\star] + (C_2 + C_3)M}{\varepsilon^2} \right\rfloor \equiv \mathcal{O}\left(\varepsilon^{-2}\right)$$

*iterations, then $\tilde{x}^K$ is an $\varepsilon$-stationary point of* (1) *in the sense of Definition 2.2.*

*Remark* 3.1. [**Choice of accuracies** $\epsilon_i^k$] To guarantee $\frac{1}{n} \sum_{i=1}^{n} \sum_{k=0}^{K+1} \epsilon_{i,k}^2 \leq M$ in Theorem 3.1 for a given constant $M > 0$ and for all $K \geq 0$, one can choose, e.g., $\epsilon_{i,k}^2 := \frac{M}{2(k+1)^2}$ for all $i \in [n]$ and $k \geq 0$. In this case, we can easily show that $\frac{1}{n} \sum_{i=1}^{n} \sum_{k=0}^{K+1} \epsilon_{i,k}^2 = \frac{M}{2} \sum_{k=0}^{K+1} \frac{1}{(k+1)^2} \leq M$. Note that, instead of using absolute accuracies, one can also use relative accuracies as $\|\epsilon_{i,k}\|^2 \leq \theta \|x_i^{k+1} - x_i^k\|^2$ for a given constant $\theta > 0$, which is more practical, while still achieving a similar convergence guarantee. Such an idea has been widely used in the literature, including [28] (see Supp. Doc. A.7).

*Remark* 3.2 (**Comparison**). Since (1) is nonconvex, our $\mathcal{O}\left(\varepsilon^{-2}\right)$ communication complexity is the state-of-the-art, matching the lower bound complexity (up to a constant factor) [49]. However, different from the convergence analysis of FedSplit and FedPD [49], our flexible sampling scheme allows us to update a subset of users at each round and still obtains convergence. This can potentially further resolve the communication bottleneck [22]. We note that FedSplit is a variant of the Peaceman-Rachford splitting method, i.e. $\alpha = 2$ and only considers convex non-composite case while we use a relaxation parameter $\alpha < 2$ and for a more general nonconvex composite problem (1).

The following corollary specifies the convergence of Algorithm 1 with a specific choice of stepsizes and exact evaluation of $\text{prox}_{\eta f_i}$, whose proof is in Sup. Doc. A.6.

**Corollary 3.1.** *Suppose that Assumptions 2.1, 2.2, and 3.1 hold. Let $\{(x_i^k, y_i^k, \hat{x}_i^k, \bar{x}^k)\}$ be generated by Algorithm 1 using stepsizes $\alpha = 1$, $\eta = \frac{1}{3L}$, and $p_i = \frac{1}{n}$. Under exact evaluation of $\text{prox}_{\eta f_i}$, i.e. $\epsilon_{i,k} = 0$ for all $i \in [n]$ and $k \geq 0$, the following bound holds*

$$\frac{1}{K+1}\sum_{k=0}^{K}\mathbb{E}\left[\|\mathcal{G}_\eta(\bar{x}^k)\|^2\right] \leq \frac{160Ln}{3(K+1)}[F(x^0) - F^\star]. \tag{6}$$

*Let $\tilde{x}^K$ be selected uniformly at random from $\{\bar{x}^0, \cdots, \bar{x}^K\}$ as the output of Algorithm 1. Then after at most*

$$K := \left\lfloor \frac{160Ln[F(x^0) - F^\star]}{3\varepsilon^2} \right\rfloor \equiv \mathcal{O}\left(\varepsilon^{-2}\right),$$

*communication rounds, $\tilde{x}^K$ becomes an $\varepsilon$-stationary point of (1) (defined by Definition 2.2).*

# 4 AsyncFedDR and Its Convergence Guarantee

**Motivation.** Although **FedDR** has been shown to converge, it is more practical to account for the system heterogeneity of local users. Requiring synchronous aggregation at the end of each communication round may lead to slow down in training. It is natural to have asynchronous update from local users as seen, e.g., in [35, 39]. However, asynchronous implementation remains limited in FL. Here, we propose **asyncFedDR**, an asynchronous variant of **FedDR**, and analyze its convergence guarantee. For the sake of our analysis, we only consider $\mathcal{S}_k := \{i_k\}$, the exact evaluation of $\text{prox}_{\eta f_i}$, and bounded delay, but extensions to general $\mathcal{S}_k$ and inexact $\text{prox}_{\eta f_i}$ are similar to Algorithm 1.

## 4.1 Derivation of asyncFedDR

Let us first explain the main idea of **asyncFedDR**. At each iteration $k$, each user receives a delay copy $\bar{x}^{k-d_{i_k}^k}$ of $\bar{x}^k$ from the server with a delay $d_{i_k}^k$. The active user $i_k$ will update its own local model $(y_i^k, x_i^k, \hat{x}_i^k)$ in an asynchronous mode without waiting for others to complete. Once completing its update, user $i_k$ just sends an increment $\Delta\hat{x}_{i_k}^k$ to the server to update the global model, while others may be reading. Overall, the complete **asyncFedDR** is presented in Algorithm 2.

In our analysis below, a transition of iteration from $k$ to $k+1$ is triggered whenever a user completes its update. Moreover, at Step 3, active user $i_k$ is chosen from a realization $(i_k, d^k)$ of a joint random vector $(\hat{i}_k, \hat{d}^k)$ at the $k$-th iteration. Here, we do not assume $i_k$ to be uniformly random or independent of the delay $d^k$. This allows Algorithm 2 to capture the variety of asynchronous implementations and architectures. Note that $\bar{x}^{k-d_{i_k}^k}$ at Step 4 is a delayed version of $\bar{x}^k$, which only exists on the server when user $i_k$ is reading. However, right after, $\bar{x}^k$ may be updated by another user.

**Illustrative example.** To better understand the update of **asyncFedDR**, Figure 1 depicts a simple scenario where there are 4 users ($C1$ - $C4$) asynchronously perform updates and with $g(\cdot) = 0$. At iteration $k = 4$, user $C4$ finishes its update so that the server performs updates. During this process, user $C1$ starts its update by receiving a global model $\bar{x}^{4-d_{i_4}^4}$ from server which is the average of $(\hat{x}_1^4, \hat{x}_2^4, \hat{x}_3^4, \hat{x}_4^4)$. At iteration $t = 7$, $C1$ finishes its update. Although $\hat{x}_1$ and $\hat{x}_4$ do not change during this time, i.e. $\hat{x}_1^6 = \hat{x}_1^4$ and $\hat{x}_4^6 = \hat{x}_4^4$, $\hat{x}_2$ and $\hat{x}_3$ have been updated at $k = 5, 6$ from user $C2$ and $C3$, respectively. Therefore, the global model $\bar{x}^k$ used to perform the update at $k = 7$ is actually aggregated from $(\hat{x}_1^6, \hat{x}_2^4, \hat{x}_3^5, \hat{x}_4^6)$ not $(\hat{x}_1^6, \hat{x}_2^6, \hat{x}_3^6, \hat{x}_4^6)$. In other words, each user receives a delay estimate $\bar{x}^{k-d^k}$ where $d^k = (d_1^k, \cdots, d_n^k)$ is a delay vector and $d_i^k = \max\{t \in [k] : i_t = i\}$, i.e. the

---
**Algorithm 2** (Asynchronous FedDR (**asyncFedDR**))
---
1: **Initialization:** Take $x^0 \in \mathrm{dom}(F)$ and choose $\eta > 0$ and $\alpha > 0$.
   Initialize the server with $\bar{x}^0 := x^0$ and $\tilde{x}^0 := 0$.
   Initialize each user $i \in [n]$ with $y_i^0 := x^0$, $x_i^0 := \mathrm{prox}_{\eta f_i}(y_i^0)$, and $\hat{x}_i^0 := 2x_i^0 - y_i^0$.
2: **For** $k := 0, \cdots, K$ **do**
3:   Select $i_k$ such that $(i_k, d^k)$ is a realization of $(\hat{i}_k, \hat{d}^k)$.
4:   [*Communication*] User $i_k$ receives $\bar{x}^{k-d_{i_k}^k}$, a delayed version of $\bar{x}^k$ with the delay $d_{i_k}^k$.
5:   [*Local update*] User $i_k$ updates

$$y_{i_k}^{k+1} := y_{i_k}^k + \alpha(\bar{x}^{k-d_{i_k}^k} - x_{i_k}^k), \quad x_{i_k}^{k+1} := \mathrm{prox}_{\eta f_{i_k}}(y_{i_k}^{k+1}), \text{ and } \hat{x}_{i_k}^{k+1} := 2x_{i_k}^{k+1} - y_{i_k}^{k+1}.$$

   Other users maintain $y_i^{k+1} := y_i^k$, $x_i^{k+1} := x_i^k$, and $\hat{x}_i^{k+1} := \hat{x}_i^k$ for $i \neq i_k$.
6:   [*Communication*] User $i_k$ sends $\Delta_{i_k}^k := \hat{x}_{i_k}^{k+1} - \hat{x}_{i_k}^k$ back to the server.
7:   [*Sever aggregation*] The server aggregates $\tilde{x}^{k+1} := \tilde{x}^k + \frac{1}{n}\Delta_{i_k}^k$.
8:   [*Sever update*] Then, the sever updates $\bar{x}^{k+1} := \mathrm{prox}_{\eta g}(\tilde{x}^{k+1})$.
9: **End For**
---

last time $\hat{x}_i$ gets updated up to iteration $k$. Note that when $d_i^k = 0$ for all $i$, Algorithm 2 reduces to its synchronous variant, i.e. a special variant of Algorithm 1 with $\mathcal{S}_k = \{i_k\}$.

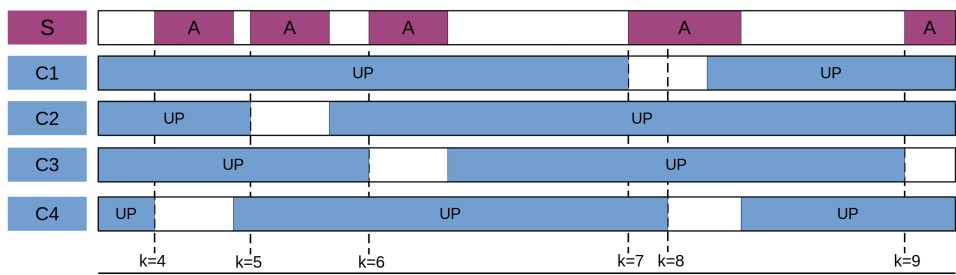

Figure 1: Asynchronous update with 4 users and without regularizer $g$. Here, "A" blocks represent server process and "UP" blocks represent user process; $C_1$-$C_4$ are communication rounds.

## 4.2   Convergence analysis

Since we treat the active user $i_k$ and the delay vector $d^k$ jointly at each iteration $k$ as a realization of a joint random vector $(\hat{i}_k, \hat{d}^k)$, we adopt the probabilistic model from [5] to analyze Algorithm 2. This new model allows us to cope with a more general class of asynchronous variants of our method.

**Probabilistic model.** Let $\xi^k := (i_k, d^k)$ be a realization of a random vector $\hat{\xi}^k := (\hat{i}_k, \hat{d}^k)$ containing the user index $\hat{i}_k \in [n]$ and the delay vector $\hat{d}^k = (\hat{d}_1^k, \cdots, \hat{d}_n^k) \in \mathcal{D} := \{0, 1, \cdots, \tau\}^n$ presented at the $k$-the iteration, respectively. We consider $k + 1$ random variables that form a random vector $\hat{\xi}^{0:k} := (\hat{\xi}^0, \cdots, \hat{\xi}^k)$. We also use $\xi^{0:k} = (\xi^0, \xi^1, \cdots, \xi^k)$ for $k + 1$ possible values of the random vector $\hat{\xi}^{0:k}$. Let $\Omega$ be the sample space of all sequences $\omega := \{(i_k, d^k)\}_{k \geq 0}$. We define a cylinder $\mathcal{C}_k(\xi^{0:k}) := \{\omega \in \Omega : (\omega_0, \cdots, \omega_k) = \xi^{0:k}\}$ and $\mathcal{C}_k$ is the set of all possible $\mathcal{C}_k(\xi^{0:k})$ when $\xi^t$, $t = 0, \cdots, k$ take all possible values, where $\omega_l$ is the $l$-th element of $\omega$. Let $\mathcal{F}_k := \sigma(\mathcal{C}_k)$ be the $\sigma$-algebra generated by $\mathcal{C}_k$ and $\mathcal{F} := \sigma(\cup_{k=0}^{\infty} \mathcal{C}_k)$. For each $\mathcal{C}_k(\xi^{0:k})$ we also equip with a probability $\mathbf{p}(\xi^{0:k}) := \mathbb{P}(\mathcal{C}_k(\xi^{0:k}))$. Then, $(\Omega, \mathcal{F}, \mathbb{P})$ forms a probability space. Assume that $\mathbf{p}(\xi^{0:k}) := \mathbb{P}(\hat{\xi}^{0:k} = \xi^{0:k}) > 0$. Our conditional probability is defined as $\mathbf{p}((i, d) \mid \xi^{0:k}) := \mathbb{P}(\mathcal{C}_{k+1}(\xi^{0:k+1}))/\mathbb{P}(\mathcal{C}_k(\xi^{0:k}))$, where $\mathbf{p}((i, d) \mid \xi^{0:k}) := 0$ if $\mathbf{p}(\xi^{0:k}) = 0$. We refer to Supp. Doc. B.2 for more details of our probabilistic model.

To analyze Algorithm 2, we impose Assumption 4.1 on the implementation below.

**Assumption 4.1.** For all $i \in [n]$ and $\omega \in \Omega$, there exists at least one $t \in \{0, 1, \cdots, T\}$ with $T > 0$, such that

$$\sum_{d \in \mathcal{D}} \mathbf{p}((i, d) \mid \xi^{0:k+t-1}) \geq \hat{\mathbf{p}} \quad \text{if } \mathbf{p}(\xi^{0:k}) > 0, \tag{7}$$

for a given $\hat{\mathbf{p}} > 0$ and any $k \geq 0$. Assume also that $d_i^k \leq \tau$ and $d_{i_k}^k = 0$ for all $k \geq 0$ and $i, i_k \in [n]$.

Assumption 4.1 implies that during an interval of $T$ iterations, every user has a non-negligible positive probability to be updated. Note that if the user $i_k$ is active, then it uses recent value with no delay, i.e., $d_{i_k}^k = 0$ as in Assumption 4.1. Moreover, the bounded delay assumption $d_i^k \leq \tau$ is standard to analyze convergence of asynchronous algorithms, see e.g., [5, 32, 34, 35, 44].

Suppose that we choose $0 < \alpha < \bar{\alpha}$ and $0 < \eta < \bar{\eta}$ in Algorithm 2, where $c := \frac{2\tau^2 - n}{n^2}$ is given, and $\bar{\alpha} > 0$ and $\bar{\eta} > 0$ are respectively computed by

$$\bar{\alpha} := \begin{cases} 1 & \text{if } 2\tau^2 \leq n, \\ \frac{2}{2+c} & \text{otherwise,} \end{cases} \quad \text{and} \quad \bar{\eta} := \begin{cases} \frac{\sqrt{16 - 8\alpha - 7\alpha^2} - \alpha}{2L(2+\alpha)} & \text{if } 2\tau^2 \leq n, \\ \frac{\sqrt{16 - 8\alpha - (7+4c+4c^2)\alpha^2} - \alpha}{2L[2 + (1+c)\alpha]} & \text{otherwise.} \end{cases} \tag{8}$$

Next, we introduce the following two constants:

$$\rho := \begin{cases} \frac{2(1-\alpha) - (2+\alpha)L^2\eta^2 - L\alpha\eta}{\alpha\eta n} & \text{if } 2\tau^2 \leq n, \\ \frac{n^2[2(1-\alpha) - (2+\alpha)L^2\eta^2 - L\alpha\eta] - \alpha(1+\eta^2 L^2)(2\tau^2 - n)}{\alpha\eta n^3} & \text{otherwise.} \end{cases} \tag{9}$$

$$D := \frac{8\alpha^2(1 + L^2\eta^2)(\tau^2 + 2Tn\hat{\mathbf{p}}) + 8n^2(1 + L^2\eta^2 + T\alpha^2\hat{\mathbf{p}})}{\hat{\mathbf{p}}\alpha^2 n^2}.$$

Then, both $\rho$ and $D$ are positive. We emphasize that though these formulas look complicated, they are computed explicitly without any tuning. Theorem 4.1 proves the convergence of Algorithm 2, whose analysis is in Supp. Doc. B.

**Theorem 4.1.** *Suppose that Assumption 2.1, 2.2, and 4.1 hold for* (1). *Let* $\bar{\alpha}$, $\bar{\eta}$, $\rho$, *and* $D$ *be given by* (8) *and* (9), *respectively. Let* $\{(x_i^k, y_i^k, \bar{x}^k)\}$ *be generated by Algorithm 2 with stepsizes* $\alpha \in (0, \bar{\alpha})$ *and* $\eta \in (0, \bar{\eta})$. *Then, the following bound holds:*

$$\frac{1}{K+1} \sum_{k=0}^{K} \mathbb{E}\left[ \|\mathcal{G}_\eta(\bar{x}^k)\|^2 \right] \leq \frac{\hat{C}\left[F(x^0) - F^\star\right]}{K+1}, \tag{10}$$

*where* $\hat{C} := \frac{2(1+\eta L)^2 D}{n\eta^2\rho} > 0$ *depending on* $n, L, \eta, \alpha, \tau, T,$ *and* $\hat{\mathbf{p}}$.

*Let* $\tilde{x}_K$ *be selected uniformly at random from* $\{\bar{x}^0, \cdots, \bar{x}^K\}$ *as the output of Algorithm 2. Then, after at most* $K := \mathcal{O}\left(\varepsilon^{-2}\right)$ *iterations,* $\tilde{x}^K$ *is an* $\varepsilon$-*stationary point of* (1) *as in Definition 2.2.*

*Remark* 4.1. From Theorem 4.1, we can see that **asyncFedDR** achieves the same worst-case communication complexity $\mathcal{O}\left(\varepsilon^{-2}\right)$ (up to a constant factor) as **FedDR**, but with smaller $\alpha$ and $\eta$.

## 5 Numerical Experiments

To evaluate the performance of **FedDR** and **asyncFedDR**, we conduct multiple experiments using both synthetic and real datasets. Since most existing methods are developed for non-composite problems, we also implement three other methods: **FedAvg**, **FedProx**, and **FedPD** to compare for this setting. We use training loss, training accuracy, and test accuracy as our performance metrics.

**Implementation.** To compare synchronous algorithms, we reuse the implementation of FedAvg and FedProx in [23] and implement FedDR and FedPD on top of it. To conduct the asynchronous examples, we implement our algorithms based on the asynchronous framework in [3]. All experiments are run on a Linux-based server with multiple nodes and configuration: 24-core 2.50GHz Intel processors, 30M cache, and 256GB RAM.

**Models and hyper-parameters selection.** Our models are neural networks, and their detail is given in Supp. Doc. C. As in [23], we use the same local solver (SGD) for all algorithms and run the local updates for 20 epochs. Parameters for each algorithm such as $\mu$ for FedProx, $\eta$ for FedPD, and $\alpha$ and $\eta$ for FedDR are tuned from a wide range of values. For each dataset, we pick the parameters that work best for each algorithm and plot their performance on the chosen parameters.

**Results on synthetic datasets.** We compare these algorithms using synthetic dataset in both iid and non-iid settings. We follow the data generation procedures described in [23, 38] to generate one iid dataset `synthetic-iid` and three non-iid datasets: `synthetic-`$(r, s)$ for $(r, s) =$

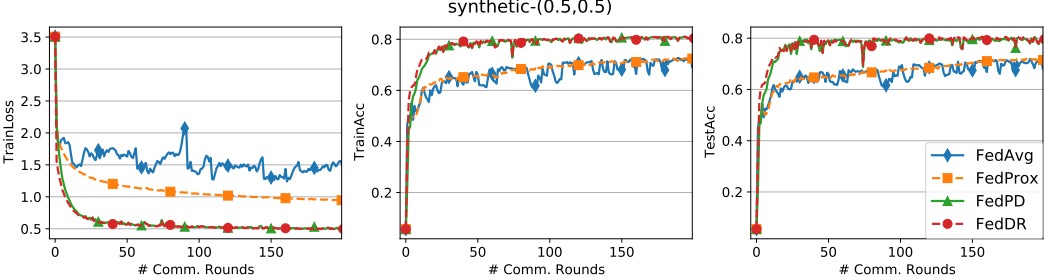

Figure 2: The performance of 4 algorithms on non-iid synthetic datasets without user sampling

$\{(0,0),(0.5,0.5),(1,1)\}$. We first compare these algorithms without using the user sampling scheme, i.e. all users perform update at each communication round, and for non-composite model of (1).

We report the performance of these algorithms on one non-iid dataset in Figure 2, but more results can be found in Sup. Doc. C. FedDR and FedPD are comparable in these datasets and they both outperform FedProx and FedAvg. FedProx works better than FedAvg which aligns with the results in [23]. However, when comparing on more datasets, our algorithm overall performs better than others.

Now we compare these algorithms where we sample 10 users out of 30 to perform update at each communication round for FedAvg, FedProx, and FedDR while we use all users for FedPD since FedPD only has convergence guarantee for this setting. In this test, the evaluation metric is plotted in terms of the number of bytes communicated between users and server at each communication round. Note that using user sampling scheme in this case can save one-third of communication cost each round. Figure 3 depicts the performance of 4 algorithms on one dataset, see also Sup. Doc. C.

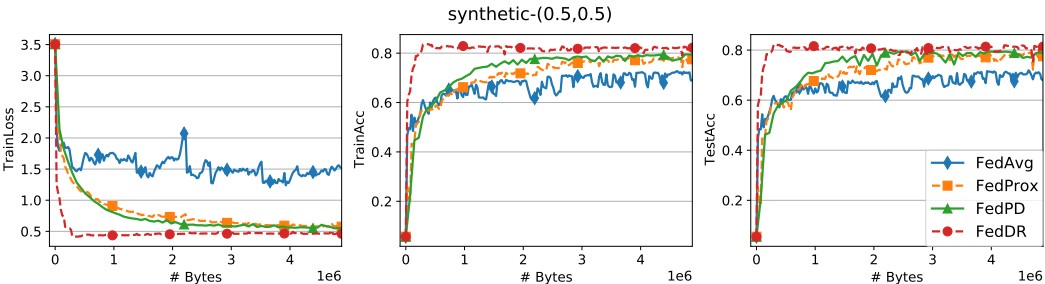

Figure 3: The performance of 4 algorithms with user sampling scheme on non-iid synthetic datasets.

From Figure 3, FedDR performs well compared to others. FedProx using user sampling scheme performs better and is slightly behind FedPD while FedDR, FedPD, and FedProx outperform FedAvg.

**Results on FEMNIST datasets.** FEMNIST [4] is an extended version of the MNIST dataset [19] where the data is partitioned by the writer of the digit/character. It has a total of 62 classes (10 digits, 26 upper-case and 26 lower-case letters) with over 800,000 samples. In this example, there are total of 200 users and we sample 50 users to perform update at each round of communication for FedAvg, FedProx, and FedDR while we use all users to perform update for FedPD. Fig. 4 depicts the performance of 4 algorithms in terms of communication cost. From Fig. 4, FedDR can achieve lower loss value and higher training accuracy than other algorithms while FedPD can reach the same test accuracy as ours at the end. Overall, FedDR seems working better than other algorithms in this test.

**Results with the $\ell_1$-norm regularizer.** We now consider the composite setting with $g(x) := 0.01 \|x\|_1$ to verify Algorithm 1 on different inexactness levels $\epsilon_{i,k}$ by varying the learning rate $(lr)$ and the number of local SGD epochs to approximately evaluate $\text{prox}_{\eta f_i}(y_i^k)$. We run Algorithm 1 on the FEMNIST dataset, and the results are shown in Figure 5.

We observe that Algorithm 1 works best when local learning rate is $0.003$ which aligns with [23] for the non-composite case. It also performs better when we decrease $\epsilon_{i,k}$ by increasing the number of epochs in evaluating $\text{prox}_{\eta f_i}$. This performance confirms our theoretical results in Supp. Doc. A.5.

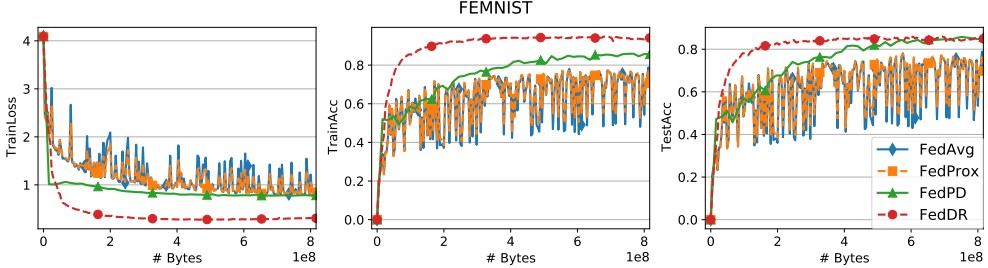

Figure 4: The performance of 4 algorithms on the FEMNIST dataset.

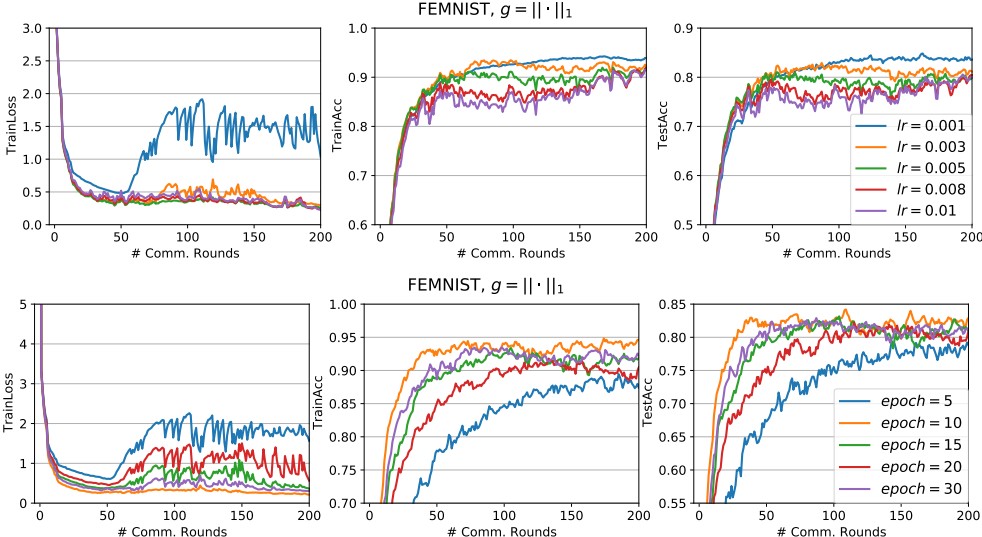

Figure 5: The performance of **FedDR** on `FEMNIST` dataset in composite setting.

**Results using asynchronous update.** To illustrate the advantage of asyncFedDR over FedDR, we conduct another example to train MNIST dataset using 20 users. Since we run these experiments on computing nodes with identical configurations, we simulate the case with computing power discrepancy between users by adding variable delay to each user's update process such that the difference between the fastest user may be up to twice as fast as the slowest one.

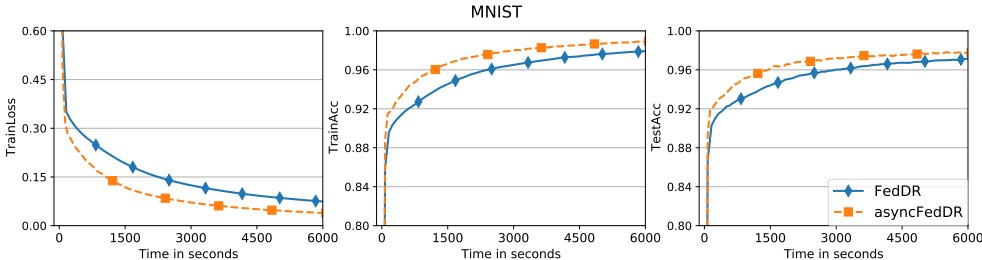

Figure 6: The performance of **FedDR** and **asyncFedDR** on the MNIST dataset.

The results of two variants are presented in Figure 6, see Supp. Doc. C for more examples. We can see that asyncFedDR can achieve better performance than FedDR in terms of training time which illustrate the advantage of asynchronous update in heterogeneous computing power.

## Acknowledgments and Disclosure of Funding

The work of Quoc Tran-Dinh is partially supported by the Office of Naval Research (ONR), grant No. N00014-20-1-2088. The authors would also like to thank all the anonymous reviewers and the ACs for their constructive comments to improve the paper.

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
