**Supplementary Document**

# FedDR – Randomized Douglas-Rachford Splitting Algorithms for Nonconvex Federated Composite Optimization



## A    The Analysis of Algorithm 1: Randomized Coordinate Variant — FedDR

In this Supplementary Document (Supp. Doc.), we first provide additional details in the derivation of Algorithm 1, **FedDR**. Then, we present the full proofs of the convergence results of Algorithm 1.

### A.1    Derivation of Algorithm 1

Our first step is to recast (1) into a constrained reformulation. Next, we apply the classical Douglas-Rachford (DR) splitting scheme to this reformulation. Finally, we randomize its updates to obtain a randomized block-coordinate DR variant.

**(a) Constrained reformulation.** With a little abuse of notation, we can equivalently write (1) into the following constrained minimization problem:

$$
\begin{cases}
\min\limits_{x_1,\cdots,x_n} & \left\{ F(\mathbf{x}) := f(\mathbf{x}) + g(\mathbf{x}) \equiv \dfrac{1}{n}\sum\limits_{i=1}^{n} f_i(x_i) + g(x_1) \right\} \\
\text{s.t.} & x_2 = x_1,\ x_3 = x_1,\ \cdots,\ x_n = x_1.
\end{cases}
\tag{11}
$$

where $\mathbf{x} := [x_1, x_2, \cdots, x_n]$ concatenates $n$ duplicated variables $x_1, x_2, \cdots, x_n$ of $x$ in (1) such that it forms a column vector in $\mathbb{R}^{np}$. Such duplications are characterized by $x_2 = x_1, x_3 = x_1, \cdots, x_n = x_1$, which define a linear subspace $\mathcal{L} := \{\mathbf{x} \in \mathbb{R}^{np} : x_2 = x_1,\ x_3 = x_1, \cdots, x_n = x_1\}$ in $\mathbb{R}^{np}$.

**(b) Unconstrained reformulation.** Let $\delta_{\mathcal{L}}$ be the indicator function of $\mathcal{L}$, i.e. $\delta_{\mathcal{L}}(\mathbf{x}) = 0$ if $\mathbf{x} \in \mathcal{L}$, and $\delta_{\mathcal{L}}(\mathbf{x}) = +\infty$, otherwise. Then, we can rewrite (11) into the following unconstrained setting:

$$
\min_{\mathbf{x} \in \mathbb{R}^{np}} \left\{ F(\mathbf{x}) := f(\mathbf{x}) + g(\mathbf{x}) + \delta_{\mathcal{L}}(\mathbf{x}) \equiv \frac{1}{n}\sum_{i=1}^{n} f_i(x_i) + g(x_1) + \delta_{\mathcal{L}}(\mathbf{x}) \right\}.
\tag{12}
$$

Clearly, (12) can be viewed as a composite nonconvex minimization problem of $f(\mathbf{x})$ and $g(\mathbf{x}) + \delta_{\mathcal{L}}(\mathbf{x})$. The first-order optimality condition of (12) can be written as

$$
0 \in \nabla f(\mathbf{x}^\star) + \partial g(\mathbf{x}^\star) + \partial \delta_{\mathcal{L}}(\mathbf{x}^\star),
\tag{13}
$$

where $\partial \delta_{\mathcal{L}}$ is the subdifferential of $\delta_{\mathcal{L}}$, which is the normal cone of $\mathcal{L}$ (or, equivalently, $\partial \delta_{\mathcal{L}}(\mathbf{x}) = \mathcal{L}^{\perp}$ if $\mathbf{x} \in \mathcal{L}$, the orthogonal subspace of $\mathcal{L}$, and $\partial \delta_{\mathcal{L}}(\mathbf{x}) = \emptyset$, otherwise), and $\partial g$ is the subdifferential of $g$. Note that since $f$ is nonconvex, (13) only provides a necessary condition for $\mathbf{x}^\star := [x_1^\star, \cdots, x_n^\star]$ to be a local minimizer. Any $\mathbf{x}^\star$ satisfying (13) is called a (first-order) stationary point of (12). In this case, we have $x_i^\star = x_1^\star$ for all $i \in [n]$. Hence, using (13), we have $0 \in \nabla f(\mathbf{x}^\star) + \partial g(\mathbf{x}^\star) + \mathcal{L}^{\perp}$. This condition is equivalent to $0 \in \frac{1}{n}\sum_{i=1}^{n} \nabla f_i(x_i^\star) + \partial g(x_1^\star)$. However, since $x_i^\star = x_1^\star$ for all $i \in [n]$, the last inclusion becomes $0 \in \frac{1}{n}\sum_{i=1}^{n} \nabla f_i(x_1^\star) + \partial g(x_1^\star)$. Equivalently, we have $x^\star := x_1^\star$ to be a stationary point of (1).

**(c) Full parallel DR variant.** Let us apply the DR splitting method to (13), which can be written explicitly as follows:

$$
\begin{cases}
\mathbf{y}^{k+1} & := \ \mathbf{x}^k + \alpha(\bar{\mathbf{x}}^k - \mathbf{x}^k), \\
\mathbf{x}^{k+1} & := \ \mathrm{prox}_{n\eta f}(\mathbf{y}^{k+1}), \\
\bar{\mathbf{x}}^{k+1} & := \ \mathrm{prox}_{n\eta(g+\delta_{\mathcal{L}})}(2\mathbf{x}^{k+1} - \mathbf{y}^{k+1}),
\end{cases}
\tag{14}
$$

where $\eta > 0$ is a given such that $n\eta$ is a step-size and $\alpha \in (0, 2]$ is a relaxation parameter [40]. If $\alpha = 1$, then we recover the classical Douglas-Rachford scheme [27] and if $\alpha = 2$, then we recover the Peaceman-Rachford splitting scheme [1]. Note that the classical DR scheme studied in [27] was developed to solve monotone inclusions, and in our context, convex problems. Recently, it has been extended to solve nonconvex optimization problems, see, e.g., [20, 40].

Let us further exploit the structure of $f$, $g$, and $\delta_{\mathcal{L}}$ in (12) to obtain a special parallel DR variant.

- First, since $f(\mathbf{x}) = \frac{1}{n} \sum_{i=1}^{n} f_i(x_i)$, we have

$$\min_{\mathbf{x}} \left\{ f(\mathbf{x}) + \frac{1}{2n\eta} \|\mathbf{x} - \mathbf{y}^{k+1}\|^2 \right\} = \min_{\mathbf{x}} \left\{ \frac{1}{n} \sum_{i=1}^{n} \left[ f_i(x_i) + \frac{1}{2\eta} \|x_i - y_i^{k+1}\|^2 \right] \right\}$$

$$= \frac{1}{n} \sum_{i=1}^{n} \min_{x_i} \left\{ f_i(x_i) + \frac{1}{2\eta} \|x_i - y_i^{k+1}\|^2 \right\}.$$

Hence, we can decompose the computation of $\mathbf{x}^{k+1} := \operatorname{prox}_{n\eta f}(\mathbf{y}^{k+1})$ from (14) as $x_i^{k+1} := \operatorname{prox}_{\eta f_i}(y_i^{k+1})$ for all $i \in [n]$.

- Next, we denote $\hat{\mathbf{x}}^{k+1} := 2\mathbf{x}^{k+1} - \mathbf{y}^{k+1}$, or equivalently, in component-wise $\hat{x}_i^{k+1} := 2x_i^{k+1} - y_i^{k+1}$ for all $i \in [n]$.
- Finally, the third line of (14) $\bar{\mathbf{x}}^{k+1} := \operatorname{prox}_{n\eta(g+\delta_{\mathcal{L}})}(\hat{\mathbf{x}}^{k+1})$ can be rewritten as

$$\bar{\mathbf{x}}^{k+1} := \operatorname{prox}_{n\eta(g+\delta_{\mathcal{L}})}(\hat{\mathbf{x}}^{k+1}) = \begin{cases} \underset{[x_1,\cdots,x_n]}{\arg\min} \left\{ g(x_1) + \frac{1}{2n\eta} \sum_{i=1}^{n} \|x_i - \hat{x}_i^{k+1}\|^2 \right\} \\ \text{s.t.} \quad x_i = x_1, \quad \text{for all } i = 2, \cdots, n. \end{cases} \quad (15)$$

Let us solve(15) explicitly. First, we define a Lagrange function associated with (15) as

$$\mathcal{L}(\mathbf{x}, \mathbf{z}) = g(x_1) + \frac{1}{2n\eta} \sum_{i=1}^{n} \|x_i - \hat{x}_i^{k+1}\|^2 + \sum_{i=1}^{n-1} z_i^{\top}(x_{i+1} - x_1),$$

where $z_i$ $(i = 1, \cdots, n-1)$ are the corresponding Lagrange multipliers. Hence, the KKT condition of (15) can be written as

$$\begin{cases} \partial g(\bar{x}_1^{k+1}) + \frac{1}{n\eta}(\bar{x}_1^{k+1} - \hat{x}_1^{k+1}) - \sum_{i=1}^{n-1} z_i = 0, \\ \frac{1}{n\eta}(\bar{x}_{i+1}^{k+1} - \hat{x}_{i+1}^{k+1}) + z_i = 0, \quad \text{for all } i = 1, \cdots, n-1, \\ \bar{x}_{i+1}^{k+1} = \bar{x}_1^{k+1}, \quad \text{for all } i = 1, \cdots, n-1. \end{cases}$$

Summing up the second line from $i = 1$ to $i = n - 1$ and combining the result with the last line of this KKT condition, we have

$$n\eta \sum_{i=1}^{n-1} z_i = \sum_{i=1}^{n-1} (\hat{x}_{i+1}^{k+1} - \bar{x}_{i+1}^{k+1}) = \sum_{i=2}^{n} \hat{x}_i^{k+1} - (n-1)\bar{x}_1^{k+1}.$$

Substituting this expression into the first line of the KKT condition, we get

$$\sum_{i=1}^{n} \hat{x}_i^{k+1} - (n-1)\bar{x}_1^{k+1} = \hat{x}_1^{k+1} + n\eta \sum_{i=1}^{n-1} z_i \in \bar{x}_1^{k+1} + n\eta\partial g(\bar{x}_1^{k+1}). \quad (16)$$

This condition is equivalent to $\sum_{i=1}^{n} \hat{x}_i^{k+1} \in n\bar{x}_1^{k+1} + n\eta\partial g(\bar{x}_1^{k+1})$. By introducing a new notation $\bar{x}^{k+1} := \bar{x}_1^{k+1}$, we eventually obtain from the last inclusion that

$$\bar{\mathbf{x}}^{k+1} := [\bar{x}^{k+1}, \cdots, \bar{x}^{k+1}] \in \mathbb{R}^{np}, \quad \text{where} \quad \bar{x}^{k+1} := \operatorname{prox}_{\eta g}\left(\frac{1}{n} \sum_{i=1}^{n} \hat{x}_i^{k+1}\right).$$

If we introduce a new variable $\tilde{x}^{k+1} := \frac{1}{n} \sum_{i=1}^{n} \hat{x}_i^{k+1}$, then $\bar{x}^{k+1} := \operatorname{prox}_{\eta g}(\tilde{x}^{k+1})$.

Putting the above steps together, we obtain the following parallel DR variant for solving (1):

$$\begin{cases} y_i^{k+1} := y_i^k + \alpha(\bar{x}^k - x_i^k), \quad \forall i \in [n] \\ x_i^{k+1} := \operatorname{prox}_{\eta f_i}(y_i^{k+1}), \quad \forall i \in [n] \\ \hat{x}_i^{k+1} := 2x_i^{k+1} - y_i^{k+1}, \quad \forall i \in [n] \\ \tilde{x}^{k+1} := \frac{1}{n} \sum_{i=1}^{n} \hat{x}_i^{k+1}, \\ \bar{x}^{k+1} := \operatorname{prox}_{\eta g}(\tilde{x}^{k+1}). \end{cases} \quad (17)$$

This variant can be implemented in parallel. It is also known as a special variant of Tseng's splitting method [1] in the convex case. This variant also covers FedSplit in [33] for FL as a special

case when $g = 0$, $f_i$ is convex for all $i \in [n]$, and $\alpha = 2$. In fact, `FedSplit` is a variant of the Peaceman-Rachford method, and is different from our algorithms due to $\alpha < 2$. If $g = 0$ (i.e., without regularizer), then the last line of (17) reduces to $\bar{x}^{k+1} = \tilde{x}^{k+1}$.

**(d) Inexact block-coordinate DR variant.** Instead of performing update for all users $i \in [n]$ as in (17), we propose a new block-coordinate DR variant, called `FedDR`, where only a subset of users $\mathcal{S}_k \subseteq [n]$ performs local update then send its local model to server for aggregation. For user $i \notin \mathcal{S}_k$, the local model is unchanged, i.e., for all $i \notin \mathcal{S}_k$: $y_i^{k+1} = y_i^k$, $x_i^{k+1} = x_i^k$, and $\hat{x}_i^{k+1} = \hat{x}_i^k$. Hence, no communication with the server is needed for these users. Furthermore, we assume that we can only approximate the proximal operator $\text{prox}_{\eta f_i}$ up to a given accuracy for all $i \in [n]$. In this case, we replace the exact proximal step $x_i^k := \text{prox}_{\eta f_i}(y_i^k)$ by its approximation $x_i^k :\approx \text{prox}_{\eta f_i}(y_i^k)$ up to a given accuracy $\epsilon_{i,k} \geq 0$ such that

$$\|x_i^k - \text{prox}_{\eta f_i}(y_i^k)\| \leq \epsilon_{i,k}. \tag{18}$$

Since $x_i^k$ is approximately computed from $\text{prox}_{\eta f_i}(y_i^k)$ as in (18), we have

$$x_i^k = z_i^k + e_i^k, \quad \text{where} \quad z_i^k := \text{prox}_{\eta f_i}(y_i^k) \quad \text{and} \quad \|e_i^k\| \leq \epsilon_{i,k}. \tag{19}$$

We will use this representation of $x_i^k$ and $x_i^{k+1}$ in our analysis in the sequel.

More specifically, the update of our inexact block-coordinate DR variant can be described as follows.

- **Initialization:** Given an initial vector $x^0 \in \text{dom}(F)$ and accuracies $\epsilon_{i,0} \geq 0$.
  Initialize the server with $\bar{x}^0 := x^0$.
  Initialize all users $i \in [n]$ with $y_i^0 := x^0$, $x_i^0 :\approx \text{prox}_{\eta f_i}(y_i^0)$, and $\hat{x}_i^0 := 2x_i^0 - y_i^0$.
- **The $k$-th iteration ($k \geq 0$):** Sample a proper subset $\mathcal{S}_k \subseteq [n]$ so that $\mathcal{S}_k$ presents as the subset of *active users*.
- (***Communication***) Each user $i \in \mathcal{S}_k$ receives $\bar{x}^k$ from the server.
- (***Local/user update***) For each user $i \in \mathcal{S}_k$, given $\epsilon_{i,k+1} \geq 0$, it updates

$$\begin{cases} y_i^{k+1} & := & y_i^k + \alpha(\bar{x}^k - x_i^k) \\ x_i^{k+1} & :\approx & \text{prox}_{\eta f_i}(y_i^{k+1}) \\ \hat{x}_i^{k+1} & := & 2x_i^{k+1} - y_i^{k+1}. \end{cases}$$

  Each user $i \notin \mathcal{S}_k$ does nothing, i.e.:

$$\begin{cases} y_i^{k+1} & := & y_i^k \\ x_i^{k+1} & := & x_i^k \\ \hat{x}_i^{k+1} & := & \hat{x}_i^k. \end{cases}$$

- (***Communication***) Each user $i \in \mathcal{S}_k$ sends only $\hat{x}_i^{k+1}$ to the server.
- (***Global/Server update***) The server aggregates $\tilde{x}^{k+1} := \frac{1}{n}\sum_{i=1}^n \hat{x}_i^{k+1}$, and then compute $\bar{x}^{k+1} := \text{prox}_{\eta g}(\tilde{x}^{k+1})$.

This scheme is exactly Algorithm 1. However, the global update on $\tilde{x}^{k+1}$ can be simplified as

$$\begin{aligned} \tilde{x}^{k+1} &:= \frac{1}{n}\sum_{i=1}^n \hat{x}_i^{k+1} = \frac{1}{n}\sum_{i \in \mathcal{S}_k} \hat{x}_i^{k+1} + \frac{1}{n}\sum_{i \notin \mathcal{S}_k} \hat{x}_i^k \\ &= \frac{1}{n}\sum_{i=1}^n \hat{x}_i^k + \frac{1}{n}\sum_{i \in \mathcal{S}_k}(\hat{x}_i^{k+1} - \hat{x}_i^k) \\ &= \tilde{x}^k + \frac{1}{n}\sum_{i \in \mathcal{S}_k} \Delta\hat{x}_i^k. \end{aligned}$$

This step is implemented in Algorithm 1.

To analyze convergence of Algorithm 1, we conceptually introduce $z_i^0$ and $z_i^{k+1}$ for $i \in [n]$ as follows:

$$z_i^0 := \text{prox}_{\eta f_i}(y_i^0), \quad z_i^{k+1} := \begin{cases} \text{prox}_{\eta f_i}(x_i^{k+1}) & \text{if} \quad i \in \mathcal{S}_k \\ z_i^k & \text{if} \quad i \notin \mathcal{S}_k, \end{cases} \quad \text{and} \quad x_i^k := z_i^k + e_i^k. \tag{20}$$

Here, $e_i^k$ is the vector of errors. Note that $z_i^0$ and $z_i^{k+1}$ do not exist in actual implementation of Algorithm 1, and we only have their approximations $x_i^0$ and $x_i^{k+1}$, respectively. For any $k \geq 0$, since

$x_i^{k+1} = x_i^k$ and $z_i^{k+1} = z_i^k$ for $i \notin \mathcal{S}_k$, we have $\|x_i^{k+1} - z_i^{k+1}\| = \|e_i^{k+1}\| = \|x_i^k - z_i^k\| = \|e_i^k\|$ for $i \notin \mathcal{S}_k$. To guarantee $\|e_i^{k+1}\| = \|e_i^k\|$ for $i \notin \mathcal{S}_k$, we must choose $\epsilon_{i,k+1} := \epsilon_{i,k}$ for $i \notin \mathcal{S}_k$.

Note that in Algorithm 1, we have not specified the choice of $\mathcal{S}_k$. The subset $\mathcal{S}_k$ is an iid realization of a random set-valued mapping $\hat{\mathcal{S}}$ from $[n]$ to $2^{[n]}$, the collection of all subsets of $[n]$. Moreover, $\hat{\mathcal{S}}$ is a proper sampling scheme in the sense that $\mathbf{p}_i := \mathbb{P}(i \in \hat{S}) > 0$ for all $i \in [n]$ as stated in Assumption 3.1. By specifying this probability distribution $\mathbf{p} := (\mathbf{p}_1, \cdots, \mathbf{p}_n)$, we obtain different sampling strategies ranging from uniform to non-uniform as discussed in [36]. Our analysis below holds for arbitrary sampling scheme that satisfies Assumption 3.1.

## A.2 Further details of comparison

We have compared our methods, Algorithm 1 and Algorithm 2, with various existing FL methods in the introduction (Section 1). Here, let us further elaborate this comparison in more detail. Due to the rapid development of FL in the last few years, it is impossible to review a majority of works in this field. Hence, we only select a few algorithms that we find most related to our work in this paper.

- **FedAvg:** FedAvg [29] has become a de facto standard federated learning algorithm in practice. However, it has several limitations as discussed in many papers, including [23]. It is also difficult to analyze convergence of FedAvg, especially in the nonconvex case and heterogeneity settings (both statistical and system heterogeneity). Moreover, FedAvg originally specifies SGD with a fixed number of epochs and a fixed learning rate as its local solver, making it less flexible in practice. Convergence analysis of FedAvg requires additional assumptions apart from the standard smoothness of $f_i$. Moreover, its extension to the composite setting, e.g., in [47] only focuses on the convex case, and requires a set of strong assumptions, including bounded heterogeneity. Since it was proposed, several attempts have been made to analyze convergence of FedAvg in both convex and nonconvex settings, see, e.g., [10, 11, 24, 26, 43].

- **FedProx:** FedProx proposed in [23], on the one hand, can be viewed as an extension of FedAvg, but on the other hand, can be cast into a quadratic penalty-type method for the constrained reformulation (11) of (1). Indeed, when $g = 0$, from (11), we can define a quadratic penalty function with a penalty parameter $\mu > 0$ as follows:

$$P_\mu(\mathbf{x}) := \frac{1}{n} \sum_{i=1}^n f_i(x_i) + \frac{\mu}{2n} \sum_{i=2}^n \|x_i - x_{n+1}\|^2.$$

First, we apply an alternating minimization strategy to minimize $P_\mu$ over $[x_1, \cdots, x_n]$ and then over $x_{n+1}$. Next, instead using the full minimization over all blocks $x_1, \cdots, x_n$, a block coordinate descent strategy is applied by selecting a subset of blocks $\mathcal{S}_k \subseteq [n]$ at random. Finally, we replace the exact minimization problem of each block $x_i$ by its inexact computation. This method exactly leads to FedProx in [23]. While FedProx can potentially handle a major heterogeneity challenge, it relies on a [local] dissimilarity assumption, which could be difficult to check. In addition, this assumption limits the application of FedProx.

- **Other methods: FedPD** proposed in [49] is exactly an augmented Lagrangian method applying to the constrained reformulation (11) of (1) when $g = 0$, combining with an alternating minimization strategy as in FedProx. However, FedPD requires all users to update their computation and flips a biased coin to decide if a global communication is carried out. This method essentially violates one crucial requirement of FL, which is known as system heterogeneity. Another FL method is **FedSplit** in [33], which also requires all users to participate into each communication round. This method also relies on Peaceman-Rachford splitting scheme [1] and is different from our algorithms. Its convergence analysis is only shown for convex problems in [33]. However, as shown in [33], this scheme can overcome the fundamental statistical heterogeneity challenge in FL.

In contrast to the above methods, our methods developed in this paper always converges under standard assumptions (i.e., only the $L$-smoothness and boundedness from below). The proposed methods can handle the majority of challenges in FL, including system and statistical heterogeneity. We also allow one to use any local solver to evaluate $\text{prox}_{\eta f_i}$ up to a given adaptive accuracy. Moreover, our methods can handle convex regularizers (in particular, convex constraints), and can be implemented in an asynchronous manner.

### A.3 Preparatory lemmas

We first present a useful lemma to characterize the relationship between $x_i^k$ and $y_i^k$ for all iteration $k$. Then, we prove a sure descent lemma to establish the main results in the main text.

**Lemma A.1.** *Let $\{(y_i^k, x_i^k, z_i^k)\}$ be generated by Algorithm 1 and (20) starting from $z_i^0 := \text{prox}_{\eta f_i}(y_i^0)$ for all $i \in [n]$ as in (20). Then, for all $i \in [n]$ and $k \geq 0$, we have*

$$y_i^k = z_i^k + \eta \nabla f_i(z_i^k), \quad and \quad \hat{x}_i^k = 2x_i^k - y_i^k. \tag{21}$$

*Proof.* We prove (21) by induction. For $k = 0$, due to the initialization step, Step 1 of Algorithm 1 and (20) with $z_i^0 := \text{prox}_{\eta f_i}(y_i^0)$, we have $y_i^0 = z_i^0 + \eta \nabla f_i(z_i^0)$ and $\hat{x}_i^0 = 2x_i^0 - y_i^0$ as in (21).

Suppose that (21) holds for all $k \geq 0$, i.e., $y_i^k = z_i^k + \eta \nabla f_i(z_i^k)$ and $\hat{x}_i^k = 2x_i^k - y_i^k$. We will show that (21) holds for $k+1$, i.e. $y_i^{k+1} = z_i^{k+1} + \eta \nabla f_i(z_i^{k+1})$ and $\hat{x}_i^{k+1} = 2x_i^{k+1} - y_i^{k+1}$ for all $i \in [n]$, respectively. We have two cases:

- For any user $i \in \mathcal{S}_k$, from the optimality condition of (20), we have

$$\nabla f_i(z_i^{k+1}) + \tfrac{1}{\eta}(z_i^{k+1} - y_i^{k+1}) = 0 \quad \Rightarrow \quad y_i^{k+1} = z_i^{k+1} + \eta \nabla f_i(z_i^{k+1}).$$

  Moreover, $\hat{x}_i^{k+1} = 2x_i^{k+1} - y_i^{k+1}$ due to Step 5 of Algoritihm 1.
- For any user $i \notin \mathcal{S}_k$, since $z_i^{k+1} := z_i^k$ due to (20), $x_i^{k+1} = x_i^k$, and $y_i^{k+1} = y_i^k$, we can also write $y_i^{k+1}$ as

$$y_i^{k+1} = y_i^k \stackrel{(*)}{=} z_i^k + \eta \nabla f_i(z_i^k) = z_i^{k+1} + \eta \nabla f_i(z_i^{k+1}).$$

  Here, $(*)$ follows from our induction assumption. Moreover, for $i \notin \mathcal{S}_k$, we maintain $\hat{x}_i^{k+1} = \hat{x}_i^k$ in Algoritihm 1. By our induction assumption, and $x_i^{k+1} = x_i^k$ and $y_i^{k+1} = y_i^k$, we have $\hat{x}_i^{k+1} = \hat{x}_i^k = 2x_i^k - y_i^k = 2x_i^{k+1} - y_i^{k+1}$.

In summary, both cases above imply that $y_i^{k+1} = z_i^{k+1} + \eta \nabla f_i(z_i^{k+1})$ and $\hat{x}_i^k = 2x_i^k - y_i^k$ hold for all $i \in [n]$, which proves (21). $\qquad\square$

Our next lemma is to bound $\|\bar{x}^k - x_i^k\|^2$ in terms of $\|x_i^{k+1} - x_i^k\|^2$.

**Lemma A.2.** *Let $\{(\bar{x}_i^k, z_i^k, x_i^k)\}$ be generated by Algorithm 1 and (20), and $\alpha > 0$. Then, for all $i \in \mathcal{S}_k$ and any $\gamma_1 > 0$, we have*

$$\|\bar{x}^k - x_i^k\|^2 \leq \tfrac{2(1+\eta^2 L^2)}{\alpha^2}\Big[(1+\gamma_1)\|x_i^{k+1} - x_i^k\|^2 + \tfrac{2(1+\gamma_1)}{\gamma_1}\big(\|e_i^{k+1}\|^2 + \|e_i^k\|^2\big)\Big]. \tag{22}$$

*In particular, if $e_i^k = e_i^{k+1} = 0$, then $\|\bar{x}^k - x_i^k\|^2 \leq \tfrac{2(1+\eta^2 L^2)}{\alpha^2}\|x_i^{k+1} - x_i^k\|^2$.*

*Proof.* From the update of $y_i^{k+1}$ and Lemma A.1, for $i \in \mathcal{S}_k$, we have

$$\bar{x}^k - x_i^k = \frac{1}{\alpha}(y_i^{k+1} - y_i^k) \stackrel{(21)}{=} \frac{1}{\alpha}(z_i^{k+1} - z_i^k) + \frac{\eta}{\alpha}(\nabla f_i(z_i^{k+1}) - \nabla f_i(z_i^k)).$$

Using this expression and $\|a + b\|^2 \leq 2\|a\|^2 + 2\|b\|^2$, we can bound $\|\bar{x}^k - x_i^k\|^2$ for all $i \in \mathcal{S}_k$ as

$$\begin{aligned}
\|\bar{x}^k - x_i^k\|^2 &= \|\tfrac{1}{\alpha}(z_i^k - z_i^{k+1}) + \tfrac{\eta}{\alpha}(\nabla f_i(z_i^k) - \nabla f_i(z_i^{k+1}))\|^2 \\
&\leq \tfrac{2}{\alpha^2}\|z_i^k - z_i^{k+1}\|^2 + \tfrac{2\eta^2}{\alpha^2}\|\nabla f_i(z_i^k) - \nabla f_i(z_i^{k+1})\|^2 \\
&\leq \tfrac{2}{\alpha^2}\|z_i^{k+1} - z_i^k\|^2 + \tfrac{2\eta^2 L^2}{\alpha^2}\|z_i^{k+1} - z_i^k\|^2 \quad \text{(by the $L$-smoothness of $f_i$)} \\
&= \tfrac{2(1+\eta^2 L^2)}{\alpha^2}\|x_i^{k+1} - x_i^k - e_i^{k+1} + e_i^k\|^2 \quad \text{(by (20))} \\
&\leq \tfrac{2(1+\eta^2 L^2)}{\alpha^2}\Big[(1+\gamma_1)\|x_i^{k+1} - x_i^k\|^2 + \tfrac{2(1+\gamma_1)}{\gamma_1}\big(\|e_i^{k+1}\|^2 + \|e_i^k\|^2\big)\Big].
\end{aligned}$$

Here, we have used Young's inequality twice in the last inequality. This proves (22). When $e_i^k = e_i^{k+1} = 0$, we can set $\gamma_1 = 0$ in the above estimate to obtain the last statement. $\qquad\square$

We still need to link the norm $\sum_{i=1}^n \|x_i^k - \bar{x}^k\|^2$ to the norm of gradient mapping $\|\mathcal{G}_\eta(\bar{x}^k)\|$.

**Lemma A.3.** *Let $\{(\bar{x}_i^k, x_i^k, z_i^k)\}$ be generated by Algorithm 1 and (20), and $\alpha > 0$ and $\mathcal{G}_\eta$ be defined by (4). Then, for any $\gamma_2 > 0$, we have*

$$\|\mathcal{G}_\eta(\bar{x}^k)\|^2 \le \frac{1}{n\eta^2}\left\{(1+\eta L)^2 \sum_{i=1}^n \left[(1+\gamma_2)\|x_i^k - \bar{x}^k\|^2 + \frac{(1+\gamma_2)}{\gamma_2}\|e_i^k\|^2\right]\right\}. \quad (23)$$

*In particular, if $e_i^k = 0$ for all $i \in [n]$, then we have $\|\mathcal{G}_\eta(\bar{x}^k)\|^2 \le \frac{(1+\eta L)^2}{n\eta^2}\sum_{i=1}^n \|x_i^k - \bar{x}^k\|^2$.*

*Proof.* From Step 7 of Algorithm 1 and (21), we have

$$\tilde{x}^k \overset{\text{Step 7}}{=} \frac{1}{n}\sum_{i=1}^n \hat{x}_i^k \overset{(21)}{=} \frac{1}{n}\sum_{i=1}^n (2x_i^k - y_i^k) \overset{(21)}{=} \frac{1}{n}\sum_{i=1}^n (2x_i^k - z_i^k - \eta\nabla f_i(z_i^k)). \quad (24)$$

From the definition (4) of $\mathcal{G}_\eta$ and the update of $\bar{x}^k$, we have

$$
\begin{aligned}
\eta\|\mathcal{G}_\eta(\bar{x}^k)\| &\overset{(4)}{=} \|\bar{x}^k - \text{prox}_{\eta g}(\bar{x}^k - \eta\nabla f(\bar{x}^k))\| \\
&= \|\text{prox}_{\eta g}(\tilde{x}^k) - \text{prox}_{\eta g}(\bar{x}^k - \eta\nabla f(\bar{x}^k))\| \\
&\le \|\tilde{x}^k - \bar{x}^k + \eta\nabla f(\bar{x}^k)\| \\
&\overset{(24)}{=} \frac{1}{n}\|\sum_{i=1}^n[(2x_i^k - z_i^k - \bar{x}^k) + \eta(\nabla f_i(\bar{x}^k) - \nabla f_i(z_i^k))]\|,
\end{aligned}
$$

where we have used the non-expansive property of $\text{prox}_g$ in the first inequality and $\nabla f(\bar{x}^k) = \frac{1}{n}\sum_{i=1}^n \nabla f_i(\bar{x}^k)$ in the last line.

Finally, using the $L$-smoothness of $f_i$, we can derive from the last inequality that

$$
\begin{aligned}
\eta^2\|\mathcal{G}_\eta(\bar{x}^k)\|^2 &\le \frac{1}{n^2}\left[\sum_{i=1}^n \left(\|2x_i^k - z_i^k - \bar{x}^k\| + \eta L\|z_i^k - \bar{x}^k\|\right)\right]^2 \\
&\le \frac{1}{n}\sum_{i=1}^n \left(\|2x_i^k - z_i^k - \bar{x}^k\| + \eta L\|z_i^k - \bar{x}^k\|\right)^2 \\
&\le \frac{1}{n}\sum_{i=1}^n \left[(1+\eta L)\|x_i^k - \bar{x}^k\| + (1+\eta L)\|e_i^k\|\right]^2 \\
&\le \frac{1}{n}(1+\eta L)^2 \sum_{i=1}^n \left[(1+\gamma_2)\|x_i^k - \bar{x}^k\|^2 + \frac{(1+\gamma_2)}{\gamma_2}\|e_i^k\|^2\right],
\end{aligned}
$$

which proves (23), where $\gamma_2 > 0$. Here, we have used Young's inequality in the second and the last inequalities, and $x_i^k = z_i^k + e_i^k$ from (20) in the third line. $\qquad\square$

To analyze convergence of Algorithm 1, we introduce the following Lyapunov function:

$$V_\eta^k(\bar{x}^k) := g(\bar{x}^k) + \frac{1}{n}\sum_{i=1}^n \left[f_i(x_i^k) + \langle\nabla f_i(x_i^k), \bar{x}^k - x_i^k\rangle + \frac{1}{2\eta}\|\bar{x}^k - x_i^k\|^2\right]. \quad (25)$$

First, we prove the following lemma.

**Lemma A.4.** *Suppose that Assumption 2.1, 2.2, and 3.1 hold. Let $\{(z_i^k, x_i^k, y_i^k, \hat{x}_i^k, \bar{x}^k)\}$ be generated by Algorithm 1 and (20). Let $V_\eta^k$ be defined by (25). Then, for any $\gamma_3 > 0$, we have*

$$
\begin{aligned}
V_\eta^{k+1}(\bar{x}^{k+1}) \le{}& g(\bar{x}^k) + \frac{1}{n}\sum_{i=1}^n \left[f_i(x_i^{k+1}) + \langle\nabla f_i(x_i^{k+1}), \bar{x}^k - x_i^{k+1}\rangle + \frac{1}{2\eta}\|\bar{x}^k - x_i^{k+1}\|^2\right] \\
&- \frac{(1-\gamma_3)}{2\eta}\|\bar{x}^{k+1} - \bar{x}^k\|^2 + \frac{(1+\eta^2 L^2)}{\gamma_3\eta}E_{k+1}^2,
\end{aligned}
\quad (26)
$$

*where $E_{k+1}^2 := \frac{1}{n}\sum_{i\notin\mathcal{S}_k}\|e_i^k\|^2 + \frac{1}{n}\sum_{i\in\mathcal{S}_k}\|e_i^{k+1}\|^2$. If $E_{k+1} = 0$, then we allow $\gamma_3 = 0$.*

*Proof.* First, from $\bar{x}^{k+1} = \text{prox}_{\eta g}(\tilde{x}^{k+1})$ at Step 7 of Algorithm 1, we have $\frac{1}{\eta}(\tilde{x}^{k+1} - \bar{x}^{k+1}) \in \partial g(\bar{x}^{k+1})$. Using this expression and the convexity of $g$, we obtain

$$g(\bar{x}^{k+1}) \le g(\bar{x}^k) + \frac{1}{\eta}\langle\tilde{x}^{k+1} - \bar{x}^k, \bar{x}^{k+1} - \bar{x}^k\rangle - \frac{1}{\eta}\|\bar{x}^{k+1} - \bar{x}^k\|^2. \quad (27)$$

Next, since $y_i^{k+1} = z_i^{k+1} + \eta \nabla f_i(z_i^{k+1})$ due to (21) and $x_i^{k+1} = z_i^{k+1} + e_i^{k+1}$ due to (20), we have

$$x_i^{k+1} + \eta \nabla f_i(x_i^{k+1}) \overset{(20)}{=} z_i^{k+1} + \eta \nabla f_i(z_i^{k+1}) + e_i^{k+1} + \eta(\nabla f_i(x_i^{k+1}) - \nabla f_i(z_i^{k+1}))$$
$$\overset{(21)}{=} y_i^{k+1} + e_i^{k+1} + \eta \xi_i^{k+1}, \tag{28}$$

where $\xi_i^{k+1} := \nabla f_i(x_i^{k+1}) - \nabla f_i(z_i^{k+1})$. Using this relation, we can derive

$$\begin{aligned}
\Delta_{k+1} &:= \tfrac{1}{n} \sum_{i=1}^n \left[ f_i(x_i^{k+1}) + \langle \nabla f_i(x_i^{k+1}), \bar{x}^{k+1} - x_i^{k+1} \rangle + \tfrac{1}{2\eta} \| \bar{x}^{k+1} - x_i^{k+1} \|^2 \right] \\
&= \tfrac{1}{n} \sum_{i=1}^n \left[ f_i(x_i^{k+1}) + \langle \nabla f_i(x_i^{k+1}), \bar{x}^k - x_i^{k+1} \rangle + \tfrac{1}{2\eta} \| \bar{x}^k - x_i^{k+1} \|^2 \right] \\
&\quad + \tfrac{1}{n\eta} \sum_{i=1}^n \langle \bar{x}^k - 2x_i^{k+1} + (x_i^{k+1} + \eta \nabla f_i(x_i^{k+1})), \bar{x}^{k+1} - \bar{x}^k \rangle + \tfrac{1}{2\eta} \| \bar{x}^{k+1} - \bar{x}^k \|^2 \\
&\overset{(28)}{=} \tfrac{1}{n} \sum_{i=1}^n \left[ f_i(x_i^{k+1}) + \langle \nabla f_i(x_i^{k+1}), \bar{x}^k - x_i^{k+1} \rangle + \tfrac{1}{2\eta} \| \bar{x}^k - x_i^{k+1} \|^2 \right] \\
&\quad + \tfrac{1}{n\eta} \sum_{i=1}^n \langle \bar{x}^k - 2x_i^{k+1} + y_i^{k+1}, \bar{x}^{k+1} - \bar{x}^k \rangle + \tfrac{1}{2\eta} \| \bar{x}^{k+1} - \bar{x}^k \|^2 \\
&\quad + \tfrac{1}{n\eta} \sum_{i=1}^n \langle e_i^{k+1} + \eta \xi_i^{k+1}, \bar{x}^{k+1} - \bar{x}^k \rangle \\
&\overset{\text{Step 7}}{=} \tfrac{1}{n} \sum_{i=1}^n \left[ f_i(x_i^{k+1}) + \langle \nabla f_i(x_i^{k+1}), \bar{x}^k - x_i^{k+1} \rangle + \tfrac{1}{2\eta} \| \bar{x}^k - x_i^{k+1} \|^2 \right] \\
&\quad + \tfrac{1}{\eta} \langle \bar{x}^k - \tilde{x}^{k+1}, \bar{x}^{k+1} - \bar{x}^k \rangle + \tfrac{1}{2\eta} \| \bar{x}^{k+1} - \bar{x}^k \|^2 \\
&\quad + \tfrac{1}{n\eta} \sum_{i=1}^n \langle e_i^{k+1} + \eta \xi_i^{k+1}, \bar{x}^{k+1} - \bar{x}^k \rangle.
\end{aligned}$$

Summing up this expression and (27), and using the definition of $V_\eta^k$ in (25), we get

$$\begin{aligned}
V_\eta^{k+1}(\bar{x}^{k+1}) &= \tfrac{1}{n} \sum_{i=1}^n \left[ f_i(x_i^{k+1}) + \langle \nabla f_i(x_i^{k+1}), \bar{x}^{k+1} - x_i^{k+1} \rangle + \tfrac{1}{2\eta} \| \bar{x}^{k+1} - x_i^{k+1} \|^2 \right] + g(\bar{x}^{k+1}) \\
&\leq g(\bar{x}^k) + \tfrac{1}{n} \sum_{i=1}^n \left[ f_i(x_i^{k+1}) + \langle \nabla f_i(x_i^{k+1}), \bar{x}^k - x_i^{k+1} \rangle + \tfrac{1}{2\eta} \| \bar{x}^k - x_i^{k+1} \|^2 \right] \\
&\quad - \tfrac{1}{2\eta} \| \bar{x}^{k+1} - \bar{x}^k \|^2 + \tfrac{1}{n\eta} \sum_{i=1}^n \langle e_i^{k+1} + \eta \xi_i^{k+1}, \bar{x}^{k+1} - \bar{x}^k \rangle.
\end{aligned}$$

By Young's inequality and $e_i^{k+1} = e_i^k$ for $i \notin \mathcal{S}_k$ due to (20), for any $\gamma_3 > 0$, we can estimate

$$\begin{aligned}
\mathcal{T}_{[1]} &:= \tfrac{1}{n\eta} \sum_{i=1}^n \langle e_i^{k+1} + \eta \xi_i^{k+1}, \bar{x}^{k+1} - \bar{x}^k \rangle \\
&\leq \tfrac{1}{2n\eta} \sum_{i=1}^n \left[ \tfrac{1}{\gamma_3} \| e_i^{k+1} + \eta \xi_i^{k+1} \|^2 + \gamma_3 \| \bar{x}^{k+1} - \bar{x}^k \|^2 \right] \\
&\leq \tfrac{\gamma_3}{2\eta} \| \bar{x}^{k+1} - \bar{x}^k \|^2 + \tfrac{1}{n\eta\gamma_3} \sum_{i=1}^n \| e_i^{k+1} \|^2 + \tfrac{\eta}{n\gamma_3} \sum_{i=1}^n \| \nabla f_i(x_i^{k+1}) - \nabla f_i(z_i^{k+1}) \|^2 \\
&\overset{(2)}{\leq} \tfrac{\gamma_3}{2\eta} \| \bar{x}^{k+1} - \bar{x}^k \|^2 + \tfrac{(1+\eta^2 L^2)}{n\eta\gamma_3} \left[ \sum_{i \in \mathcal{S}_k} \| e_i^{k+1} \|^2 + \sum_{i \notin \mathcal{S}_k} \| e_i^k \|^2 \right].
\end{aligned}$$

Substituting this inequality into the last estimate, we eventually obtain (26). However, if $E_{k+1}^2 = 0$, then we can deduce from the above inequality that $\gamma_3$ can be set to zero. $\qquad \square$

Now, we prove the following key result, which holds surely for any subset $\mathcal{S}_k$ of $[n]$.

**Lemma A.5** (Sure descent lemma)**.** *Suppose that Assumption 2.1, 2.2, and 3.1 hold. Let $\{(x_i^k, y_i^k, z_i^k, \hat{x}_i^k, \bar{x}^k)\}$ be generated by Algorithm 1 and (20), and $V_\eta^k(\cdot)$ be defined by (25). Then, the following estimate holds:*

$$\begin{aligned}
V_\eta^{k+1}(\bar{x}^{k+1}) &\leq V_\eta^k(\bar{x}^k) - \tfrac{[2 - \alpha(L\eta+1) - 2L^2\eta^2 - 4\alpha\gamma_4(1+L^2\eta^2)]}{2\alpha\eta n} \sum_{i \in \mathcal{S}_k} \| x_i^{k+1} - x_i^k \|^2 \\
&\quad - \tfrac{(1-\gamma_3)}{2\eta} \| \bar{x}^{k+1} - \bar{x}^k \|^2 + \tfrac{(1+\eta^2 L^2)}{\eta\gamma_3} E_{k+1}^2 \\
&\quad + \tfrac{2(1+\eta L)^2}{\gamma_4 \eta \alpha^2 n} \sum_{i \in \mathcal{S}_k} [\| e_i^k \|^2 + \| e_i^{k+1} \|^2],
\end{aligned} \tag{29}$$

*where $E_{k+1}^2 := \tfrac{1}{n} \sum_{i \notin \mathcal{S}_k} \| e_i^k \|^2 + \tfrac{1}{n} \sum_{i \in \mathcal{S}_k} \| e_i^{k+1} \|^2$, and $\gamma_3, \gamma_4 > 0$. In particular, if $E_{k+1}^2 = 0$, then we allow $\gamma_3 = 0$, and if $e_i^k = e_i^{k+1} = 0$ for all $i \in \mathcal{S}_k$, then we allow $\gamma_4 = 0$.*

*Proof.* First, using (26), we can further derive

$$V_\eta^{k+1}(\bar{x}^{k+1}) \overset{(26)}{\leq} \frac{1}{n}\sum_{i=1}^n \left[ f_i(x_i^{k+1}) + \langle \nabla f_i(x_i^{k+1}), \bar{x}^k - x_i^{k+1}\rangle + \frac{1}{2\eta}\|\bar{x}^k - x_i^{k+1}\|^2 \right]$$

$$+ g(\bar{x}^k) - \frac{(1-\gamma_3)}{2\eta}\|\bar{x}^{k+1} - \bar{x}^k\|^2 + \frac{(1+\eta^2 L^2)}{\eta\gamma_3}E_{k+1}^2$$

$$\overset{(*)}{=} \frac{1}{n}\sum_{i\in\mathcal{S}_k} f_i(x_i^{k+1}) + \frac{1}{n}\sum_{i\in\mathcal{S}_k} \langle \nabla f_i(x_i^{k+1}), x_i^k - x_i^{k+1}\rangle \tag{30}$$

$$+ \frac{1}{n}\sum_{i\in\mathcal{S}_k}\langle \nabla f_i(x_i^{k+1}), \bar{x}^k - x_i^k\rangle + \frac{1}{2\eta n}\sum_{i\in\mathcal{S}_k}\|\bar{x}^k - x_i^{k+1}\|^2$$

$$+ \frac{1}{n}\sum_{i\notin\mathcal{S}_k} f_i(x_i^k) + \frac{1}{n}\sum_{i\notin\mathcal{S}_k}\langle\nabla f_i(x_i^k), \bar{x}^k - x_i^k\rangle + \frac{1}{2\eta n}\sum_{i\notin\mathcal{S}_k}\|\bar{x}^k - x_i^k\|^2$$

$$+ g(\bar{x}^k) - \frac{(1-\gamma_3)}{2\eta}\|\bar{x}^{k+1} - \bar{x}^k\|^2 + \frac{(1+\eta^2 L^2)}{\eta\gamma_3}E_{k+1}^2,$$

where in (*) we have used the fact that only users in $\mathcal{S}_k$ perform update and added/subtracted $x_i^k$ in the term $\langle\nabla f_i(x_i^{k+1}), \bar{x}^k - x_i^{k+1}\rangle$.

On the other hand, from the $L$-smoothness of $f_i$, we have

$$f_i(x_i^{k+1}) + \langle\nabla f(x_i^{k+1}), x_i^k - x_i^{k+1}\rangle \leq f_i(x_i^k) + \frac{L}{2}\|x_i^{k+1} - x_i^k\|^2.$$

Substituting this inequality into (30), we can further bound it as

$$V_\eta^{k+1}(\bar{x}^{k+1}) \leq \frac{1}{n}\sum_{i\in\mathcal{S}_k} f_i(x_i^k) + \frac{L}{2n}\sum_{i\in\mathcal{S}_k}\|x_i^{k+1} - x_i^k\|^2 + \frac{1}{n}\sum_{i\in\mathcal{S}_k}\langle\nabla f_i(x_i^{k+1}), \bar{x}^k - x_i^k\rangle$$

$$+ \frac{1}{2\eta n}\sum_{i\in\mathcal{S}_k}\|\bar{x}^k - x_i^{k+1}\|^2 + \frac{1}{n}\sum_{i\notin\mathcal{S}_k} f_i(x_i^k) + \frac{1}{n}\sum_{i\notin\mathcal{S}_k}\langle\nabla f_i(x_i^k), \bar{x}^k - x_i^k\rangle$$

$$+ \frac{1}{2\eta n}\sum_{i\notin\mathcal{S}_k}\|\bar{x}^k - x_i^k\|^2 + g(\bar{x}^k)$$

$$- \frac{(1-\gamma_3)}{2\eta}\|\bar{x}^{k+1} - \bar{x}^k\|^2 + \frac{(1+\eta^2 L^2)}{\eta\gamma_3}E_{k+1}^2 \tag{31}$$

$$= \frac{1}{n}\sum_{i=1}^n f_i(x_i^k) + \frac{1}{n}\sum_{i=1}^n\langle\nabla f_i(x_i^k), \bar{x}^k - x_i^k\rangle + \frac{L}{2n}\sum_{i\in\mathcal{S}_k}\|x_i^{k+1} - x_i^k\|^2$$

$$+ \frac{1}{2\eta n}\sum_{i\in\mathcal{S}_k}\|\bar{x}^k - x_i^{k+1}\|^2 + \frac{1}{n}\sum_{i\in\mathcal{S}_k}\langle\nabla f_i(x_i^{k+1}) - \nabla f_i(x_i^k), \bar{x}^k - x_i^k\rangle$$

$$+ \frac{1}{2\eta n}\sum_{i\notin\mathcal{S}_k}\|\bar{x}^k - x_i^k\|^2 + g(\bar{x}^k)$$

$$- \frac{(1-\gamma_3)}{2\eta}\|\bar{x}^{k+1} - \bar{x}^k\|^2 + \frac{(1+\eta^2 L^2)}{\eta\gamma_3}E_{k+1}^2,$$

where we have added and subtracted $\frac{1}{n}\sum_{i\in\mathcal{S}_k}\langle\nabla f_i(x_i^k), \bar{x}^k - x_i^k\rangle$ to obtain the last equality.

Next, using the following elementary expression

$$\|\bar{x}^k - x_i^{k+1}\|^2 = \|\bar{x}^k - x_i^k\|^2 + 2\langle\bar{x}^k - x_i^k, x_i^k - x_i^{k+1}\rangle + \|x_i^k - x_i^{k+1}\|^2$$

into (31), we can further derive

$$V_\eta^{k+1}(\bar{x}^{k+1}) \leq g(\bar{x}^k) + \frac{1}{n}\sum_{i=1}^n\left[ f_i(x_i^k) + \langle\nabla f_i(x_i^k), \bar{x}^k - x_i^k\rangle + \frac{1}{2\eta}\|\bar{x}^k - x_i^k\|^2 \right]$$

$$+ \frac{1}{2\eta n}\sum_{i\in\mathcal{S}_k}\|x_i^{k+1} - x_i^k\|^2 + \frac{1}{\eta n}\sum_{i\in\mathcal{S}_k}\langle x_i^{k+1} - x_i^k, x_i^k - \bar{x}^k\rangle$$

$$+ \frac{1}{n}\sum_{i\in\mathcal{S}_k}\langle\nabla f_i(x_i^{k+1}) - \nabla f_i(x_i^k), \bar{x}^k - x_i^k\rangle + \frac{L}{2n}\sum_{i\in\mathcal{S}_k}\|x_i^{k+1} - x_i^k\|^2$$

$$- \frac{(1-\gamma_3)}{2\eta}\|\bar{x}^{k+1} - \bar{x}^k\|^2 + \frac{(1+\eta^2 L^2)}{\eta\gamma_3}E_{k+1}^2 \tag{32}$$

$$= V_\eta^k(\bar{x}^k) + \frac{1+\eta L}{2\eta n}\sum_{i\in\mathcal{S}_k}\|x_i^{k+1} - x_i^k\|^2 + \frac{1}{\eta n}\sum_{i\in\mathcal{S}_k}\langle x_i^{k+1} - x_i^k, x_i^k - \bar{x}^k\rangle$$

$$- \frac{(1-\gamma_3)}{2\eta}\|\bar{x}^{k+1} - \bar{x}^k\|^2 + \frac{(1+\eta^2 L^2)}{\eta\gamma_3}E_{k+1}^2$$

$$+ \frac{1}{n}\sum_{i\in\mathcal{S}_k}\langle\nabla f_i(x_i^{k+1}) - \nabla f_i(x_i^k), \bar{x}^k - x_i^k\rangle.$$

From the update of $y_i^{k+1}$, for $i \in \mathcal{S}_k$, and similar to the proof of (28), we have

$$x_i^k - \bar{x}^k = \frac{1}{\alpha}(y_i^k - y_i^{k+1})$$

$$\overset{(28)}{=} \frac{1}{\alpha}(z_i^k - z_i^{k+1}) + \frac{\eta}{\alpha}(\nabla f_i(z_i^k) - \nabla f_i(z_i^{k+1}))$$

$$= \frac{1}{\alpha}(x_i^k - x_i^{k+1}) + \frac{\eta}{\alpha}(\nabla f_i(x_i^k) - \nabla f_i(x_i^{k+1})) + \frac{1}{\alpha}[(e_i^{k+1} + \eta\xi_i^{k+1}) - (e_i^k + \eta\xi_i^k)]$$

$$= \frac{1}{\alpha}(x_i^k - x_i^{k+1}) + \frac{\eta}{\alpha}(\nabla f_i(x_i^k) - \nabla f_i(x_i^{k+1})) + s_i^k,$$

where $s_i^k := \frac{1}{\alpha}[e_i^{k+1} + \eta\xi_i^{k+1} - (e_i^k + \eta\xi_i^k))$ with $\xi_i^k := \nabla f_i(x_i^k) - \nabla f_i(z_i^k)$.

Consequently, using the last expression and the $L$-smoothness of $f_i$, we can further bound (32) as

$$
\begin{aligned}
V_\eta^{k+1}(\bar{x}^{k+1}) &\leq V_\eta^k(\bar{x}^k) + \frac{(1+\eta L)}{2\eta n}\sum_{i\in\mathcal{S}_k}\|x_i^{k+1} - x_i^k\|^2 - \frac{1}{\alpha\eta n}\sum_{i\in\mathcal{S}_k}\|x_i^{k+1} - x_i^k\|^2 \\
&\quad - \frac{1}{\alpha n}\sum_{i\in\mathcal{S}_k}\langle x_i^{k+1} - x_i^k, \nabla f_i(x_i^{k+1}) - \nabla f_i(x_i^k)\rangle + \frac{1}{\eta n}\sum_{i\in\mathcal{S}_k}\langle s_i^k, x_i^{k+1} - x_i^k\rangle \\
&\quad + \frac{1}{\alpha n}\sum_{i\in\mathcal{S}_k}\langle\nabla f_i(x_i^{k+1}) - \nabla f_i(x_i^k), x_i^{k+1} - x_i^k\rangle \\
&\quad + \frac{\eta}{\alpha n}\sum_{i\in\mathcal{S}_k}\|\nabla f_i(x_i^{k+1}) - \nabla f_i(x_i^k)\|^2 + \frac{1}{n}\sum_{i\in\mathcal{S}_k}\langle s_i^k, \nabla f_i(x_i^{k+1}) - \nabla f_i(x_i^k)\rangle \\
&\quad - \frac{(1-\gamma_3)}{2\eta}\|\bar{x}^{k+1} - \bar{x}^k\|^2 + \frac{(1+\eta^2 L^2)}{\eta\gamma_3}E_{k+1}^2 \\
&= V_\eta^k(\bar{x}^k) + \frac{\eta}{\alpha n}\sum_{i\in\mathcal{S}_k}\|\nabla f_i(x_i^{k+1}) - \nabla f_i(x_i^k)\|^2 + \frac{[\alpha(L\eta+1)-2]}{2\alpha\eta n}\sum_{i\in\mathcal{S}_k}\|x_i^{k+1} - x_i^k\|^2 \\
&\quad + \frac{1}{\eta n}\sum_{i\in\mathcal{S}_k}\langle s_i^k, (x_i^{k+1} - x_i^k) + \eta(\nabla f_i(x_i^{k+1}) - \nabla f_i(x_i^k))\rangle \\
&\quad - \frac{(1-\gamma_3)}{2\eta}\|\bar{x}^{k+1} - \bar{x}^k\|^2 + \frac{(1+\eta^2 L^2)}{\eta\gamma_3}E_{k+1}^2 \\
&\overset{(2)}{\leq} V_\eta^k(\bar{x}^k) + \frac{\eta L^2}{\alpha n}\sum_{i\in\mathcal{S}_k}\|x_i^{k+1} - x_i^k\|^2 + \frac{[\alpha(L\eta+1)-2]}{2\alpha\eta n}\sum_{i\in\mathcal{S}_k}\|x_i^{k+1} - x_i^k\|^2 \\
&\quad - \frac{(1-\gamma_3)}{2\eta}\|\bar{x}^{k+1} - \bar{x}^k\|^2 + \frac{(1+\eta^2 L^2)}{\eta\gamma_3}E_{k+1}^2 \\
&\quad + \frac{1}{n\eta}\sum_{i\in\mathcal{S}}\left[\frac{1}{\gamma_4}\|s_i^k\|^2 + 2\gamma_4\|x_i^k - x_i^{k+1}\|^2 + 2\gamma_4\eta^2\|\nabla f_i(x_i^k) - \nabla f_i(x_i^{k+1})\|^2\right] \\
&= V_\eta^k(\bar{x}^k) - \frac{[2-\alpha(L\eta+1)-2L^2\eta^2]}{2\alpha\eta n}\sum_{i\in\mathcal{S}_k}\|x_i^{k+1} - x_i^k\|^2 \\
&\quad + \frac{1}{n\gamma_4\eta}\sum_{i\in\mathcal{S}}\|s_i^k\|^2 + \frac{2\gamma_4(1+L^2\eta^2)}{n\eta}\sum_{i\in\mathcal{S}_k}\|x_i^{k+1} - x_i^k\|^2 \\
&\quad - \frac{(1-\gamma_3)}{2\eta}\|\bar{x}^{k+1} - \bar{x}^k\|^2 + \frac{(1+\eta^2 L^2)}{\eta\gamma_3}E_{k+1}^2.
\end{aligned}
$$

Finally, we bound $\|s_i^k\|^2$ as follows:

$$
\begin{aligned}
\|s_i^k\|^2 &= \frac{1}{\alpha^2}\|e_i^{k+1} + \eta\xi_i^{k+1} - (e_i^k + \eta\xi_i^k)\|^2 \\
&\leq \frac{1}{\alpha^2}\left[\|e_i^k\| + \|e_i^{k+1}\| + \eta\|\nabla f_i(x_i^k) - \nabla f_i(z_i^k)\| + \eta\|\nabla f_i(x_i^{k+1}) - \nabla f_i(z_i^{k+1})\|\right]^2 \\
&\leq \frac{2(1+\eta L)^2}{\alpha^2}(\|e_i^k\|^2 + \|e_i^{k+1}\|^2).
\end{aligned}
$$

Substituting this inequality into the last estimate, we obtain (29). The last statement follows from the last statement of Lemmas A.3 and A.4. $\qquad\square$

### A.4 The descent property of Algorithm 1

We prove a descent property of Algorithm 1, where $\text{prox}_{\eta f_i}$ is evaluated approximately.

**Lemma A.6.** *Suppose that Assumption 2.1, 2.2, and 3.1 hold. Let $V_\eta^k(\cdot)$ be defined by (25) and $\gamma_1, \gamma_2, \gamma_4 > 0$ be given. Let $\{(x_i^k, y_i^k, \hat{x}_i^k, \bar{x}^k)\}$ be generated by Algorithm 1 using*

$$
0 < \alpha < \frac{\min\{8, \sqrt{17+64\gamma_4} - 1\}}{4(1+4\gamma_4)} \quad \text{and} \quad 0 < \eta < \frac{\sqrt{(4-\alpha)^2 - 16\alpha^2\gamma_4(1+4\gamma_4)} - \alpha}{4L(1+2\alpha\gamma_4)}. \tag{33}
$$

*Then, $V_\eta^k$ is bounded from bellow by $F^\star$, i.e. $V_\eta^k \geq F^\star$ and the following estimate holds:*

$$
\frac{\beta}{2n}\sum_{i=1}^n\|\bar{x}^k - x_i^k\|^2 \leq V_\eta^k(\bar{x}^k) - \mathbb{E}\left[V_\eta^{k+1}(\bar{x}^{k+1}) \mid \mathcal{F}_{k-1}\right] + \frac{1}{n}\sum_{i=1}^n(\rho_1\epsilon_{i,k}^2 + \rho_2\epsilon_{i,k+1}^2), \tag{34}
$$

*where*

$$
\begin{cases}
\beta := \frac{\hat{p}\alpha[2-\alpha(L\eta+1)-2L^2\eta^2-4\gamma_4\alpha(1+L^2\eta^2)]}{2\eta(1+\gamma_1)(1+L^2\eta^2)} > 0, \\
\rho_2 := \frac{2(1+\eta L)^2}{\gamma_4\eta\alpha^2} + \frac{(1+\eta^2 L^2)}{\eta} + \frac{\alpha[2-\alpha(L\eta+1)-2L^2\eta^2-4\alpha\gamma_4(1+L^2\eta^2)]}{2\eta(1+L^2\eta^2)\gamma_1}, \\
\rho_1 := \rho_2 + \frac{(1+\eta^2 L^2)}{\eta}.
\end{cases} \tag{35}
$$

*Here, if $\epsilon_{i,k} = 0$ for all $i \in [n]$ and $k \geq 0$, then we allow $\gamma_1 = \gamma_2 = \gamma_4 = \rho_1 = \rho_2 = 0$.*

*Proof.* First, to guarantee a descent property in (29), we need to choose $\eta > 0$ and $\alpha > 0$ such that $2 - \alpha(L\eta + 1) - 2L^2\eta^2 - 4\gamma_4\alpha(1 + L^2\eta^2) > 0$. We first need $\alpha$ such that $0 < \alpha < \frac{2}{1+4\gamma_4}$, the condition for $\eta$ is

$$0 < \eta < \bar{\eta} := \frac{\sqrt{(4-\alpha)^2 + 16\alpha^2\gamma_4(1+4\gamma_4)} - \alpha}{4L(1+2\alpha\gamma_4)}.$$

To guarantee $\bar{\eta} > 0$, we need to choose $0 < \alpha < \frac{\sqrt{17+64\gamma_4}-1}{4(1+4\gamma_4)}$. Combining both conditions on $\alpha$, we obtain the first condition for $\alpha$ in (33).

Now, to show the boundedness of $V_\eta^k(\bar{x}^k)$ from below, we have

$$
\begin{aligned}
V_\eta^k(\bar{x}^k) &= g(\bar{x}^k) + \frac{1}{n}\sum_{i=1}^n \left[ f_i(x_i^k) + \langle \nabla f_i(x_i^k), \bar{x}^k - x_i^k \rangle + \frac{1}{2\eta}\|\bar{x}^k - x_i^k\|^2 \right] \\
&\geq g(\bar{x}^k) + \frac{1}{n}\sum_{i=1}^n \left[ f_i(\bar{x}^k) - \frac{L}{2}\|\bar{x}^k - x_i^k\|^2 + \frac{1}{2\eta}\|\bar{x}^k - x_i^k\|^2 \right] \quad \text{(the $L$-smoothness of $f_i$)} \\
&\geq f(\bar{x}^k) + g(\bar{x}^k) + \left(\frac{1}{\eta} - L\right)\frac{1}{2n}\sum_{i=1}^n \|\bar{x}^k - x_i^k\|^2 \\
&\geq F^\star \quad \text{(since $\eta \leq \frac{1}{L}$ and Assumption 2.1).}
\end{aligned}
\tag{36}
$$

Next, from (22), we have

$$\frac{\alpha^2}{2(1+L^2\eta^2)(1+\gamma_1)}\sum_{i\in\mathcal{S}_k}\|\bar{x}^k - x_i^k\|^2 \leq \sum_{i\in\mathcal{S}_k}\left[\|x_i^{k+1} - x_i^k\|^2 + \frac{\alpha^2}{(1+L^2\eta^2)\gamma_1}\left(\|e_i^{k+1}\|^2 + \|e_i^k\|^2\right)\right].$$

Moreover, from Assumption 3.1, for a nonnegative random variable $W_i^k$ with $i \in \mathcal{S}_k$, by taking expectation of this random variable w.r.t. $\mathcal{S}_k$ conditioned on $\mathcal{F}_{k-1}$, we have

$$\mathbb{E}\left[\sum_{i\in\mathcal{S}_k} W_i^k \mid \mathcal{F}_{k-1}\right] = \sum_{\mathcal{S}}\mathbb{P}(\mathcal{S}_k = \mathcal{S})\sum_{i\in\mathcal{S}}W_i^k = \sum_{i=1}^n\sum_{\mathcal{S}:i\in\mathcal{S}}\mathbb{P}(\mathcal{S})W_i^k \overset{\text{Ass.(3.1)}}{=} \sum_{i=1}^n \mathbf{p}_i W_i^k.$$

Using this relation with $W_i^k := \|x_i^k - \bar{x}^k\|^2$, $W_i^k := \|e_i^k\|^2$, and $W_i^k := \|e_i^{k+1}\|^2$, and then combining the results with the last inequality, we can derive that

$$
\begin{aligned}
\mathbb{E}\left[\sum_{i\in\mathcal{S}_k}\|x_i^{k+1} - x_i^k\|^2 \mid \mathcal{F}_{k-1}\right] &\geq \frac{\alpha^2}{2(1+L^2\eta^2)(1+\gamma_1)}\sum_{i=1}^n \mathbf{p}_i\|\bar{x}^k - x_i^k\|^2 \\
&\quad - \frac{\alpha^2}{(1+L^2\eta^2)\gamma_1}\sum_{i=1}^n \mathbf{p}_i\left(\|e_i^{k+1}\|^2 + \|e_i^k\|^2\right) \\
&\geq \frac{\hat{\mathbf{p}}\alpha^2}{2(1+L^2\eta^2)(1+\gamma_1)}\sum_{i=1}^n\|\bar{x}^k - x_i^k\|^2 \\
&\quad - \frac{\alpha^2}{(1+L^2\eta^2)\gamma_1}\sum_{i=1}^n\left(\|e_i^{k+1}\|^2 + \|e_i^k\|^2\right),
\end{aligned}
\tag{37}
$$

where we have used $\hat{\mathbf{p}} := \min_{i\in[n]}\mathbf{p}_i > 0$ in Assumption 3.1 and $\mathbf{p}_i \leq 1$ for all $i \in [n]$.

Taking expectation both sides of (29) w.r.t. $\mathcal{S}_k$ conditioned on $\mathcal{F}_{k-1}$, and letting $\gamma_3 := 1$, we get

$$
\begin{aligned}
\mathbb{E}\left[V_\eta^{k+1}(\bar{x}^{k+1}) \mid \mathcal{F}_{k-1}\right] &\leq V_\eta^k(\bar{x}^k) + \frac{(1+\eta^2 L^2)}{\eta n}\sum_{i=1}^n\left[(1+\mathbf{p}_i)\|e_i^k\|^2 + \mathbf{p}_i\|e_i^{k+1}\|^2\right] \\
&\quad + \frac{2(1+\eta L)^2}{\gamma_4\eta\alpha^2 n}\sum_{i=1}^n \mathbf{p}_i\left[\|e_i^k\|^2 + \|e_i^{k+1}\|^2\right] \\
&\quad - \frac{[2-\alpha(L\eta+1) - 2L^2\eta^2 - 4\alpha\gamma_4(1+L^2\eta^2)]}{2\eta\alpha n}\mathbb{E}\left[\sum_{i\in\mathcal{S}_k}\|x_i^{k+1} - x_i^k\|^2 \mid \mathcal{F}_{k-1}\right].
\end{aligned}
\tag{38}
$$

Here, we have used $E_{k+1}^2 \leq \frac{1}{n}\sum_{i=1}^n\|e_i^k\|^2 + \frac{1}{n}\sum_{i\in\mathcal{S}_k}\left[\|e_i^k\|^2 + \|e_i^{k+1}\|^2\right]$ and the fact that $\mathbb{E}\left[\sum_{i\in\mathcal{S}_k}\left[\|e_i^k\|^2 + \|e_i^{k+1}\|^2\right] \mid \mathcal{F}_{k-1}\right] = \sum_{i=1}^n \mathbf{p}_i\left[\|e_i^k\|^2 + \|e_i^{k+1}\|^2\right]$. Combining (37) and (38) we obtain

$$
\begin{aligned}
\mathbb{E}\left[V_\eta^{k+1}(\bar{x}^{k+1}) \mid \mathcal{F}_{k-1}\right] &\leq V_\eta^k(\bar{x}^k) + \frac{(1+\eta^2 L^2)}{\eta(n+1)}\sum_{i=1}^n\|e_i^k\|^2 \\
&\quad + \left[\frac{2(1+\eta L)^2}{\gamma_4\eta\alpha^2 n} + \frac{(1+\eta^2 L^2)}{\eta n}\right]\sum_{i=1}^n \mathbf{p}_i\left[\|e_i^k\|^2 + \|e_i^{k+1}\|^2\right] \\
&\quad + \frac{\alpha[2-\alpha(L\eta+1) - 2L^2\eta^2 - 4\alpha\gamma_4(1+L^2\eta^2)]}{2\eta(1+L^2\eta^2)\gamma_1 n}\sum_{i=1}^n\left[\|e_i^k\|^2 + \|e_i^{k+1}\|^2\right] \\
&\quad - \frac{\hat{\mathbf{p}}\alpha[2-\alpha(L\eta+1) - 2L^2\eta^2 - 4\alpha\gamma_4(1+L^2\eta^2)]}{4\eta(1+L^2\eta^2)(1+\gamma_1)n}\sum_{i=1}^n\|\bar{x}^k - x_i^k\|^2.
\end{aligned}
$$

Rearranging terms in the last inequality and using $\mathbf{p}_i \leq 1$ and $\|e_i^k\|^2 \leq \epsilon_{i,k}^2$ for all $i \in [n]$ and $k \geq 0$ from (19), we obtain (34). Note that if $\epsilon_{i,k} = 0$ for all $i \in [n]$ and $k \geq 0$, then we allow to set $\gamma_1 = \gamma_2 = \gamma_4 = \rho_1 = \rho_2 = 0$ as a consequence of the last statement in Lemma A.2, Lemma A.3, and Lemma A.5. $\qquad\square$

## A.5 Convergence rate and communication complexity of Algorithm 1 – The inexact variant

***The proof of Theorem 3.1.*** First, from (34), we have

$$\frac{(1+\eta L)^2(1+\gamma_2)}{n\eta^2}\sum_{i=1}^n\|x_i^k-\bar{x}^k\|^2 \le \frac{2(1+\eta L)^2(1+\gamma_2)}{\eta^2\beta}\left[V_\eta^k(\bar{x}^k)-\mathbb{E}\left[V_\eta^{k+1}(\bar{x}^{k+1})\mid\mathcal{F}_{k-1}\right]\right],$$

$$+\frac{2(1+\eta L)^2(1+\gamma_2)}{n\eta^2\beta}\sum_{i=1}^n(\rho_1\epsilon_{i,k}^2+\rho_2\epsilon_{i,k+1}^2). \tag{39}$$

Substituting these estimates into (23) of Lemma A.3, we have

$$\|\mathcal{G}_\eta(\bar{x}^k)\|^2 \le \frac{2(1+\eta L)^2(1+\gamma_2)}{\eta^2\beta}\left[V_\eta^k(\bar{x}^k)-\mathbb{E}\left[V_\eta^{k+1}(\bar{x}^{k+1})\mid\mathcal{F}_{k-1}\right]\right]$$

$$+\frac{2(1+\eta L)^2(1+\gamma_2)}{n\eta^2\beta}\sum_{i=1}^n(\rho_1\epsilon_{i,k}^2+\rho_2\epsilon_{i,k+1}^2)+\frac{(1+\eta L)^2(1+\gamma_2)}{n\eta^2\gamma_2}\sum_{i=1}^n\epsilon_{i,k}^2.$$

Let us introduce three constants

$$C_1 := \frac{2(1+\eta L)^2(1+\gamma_2)}{\eta^2\beta}, \quad C_2 := \rho_1 C_1, \quad \text{and} \quad C_3 := \rho_2 C_1+\frac{(1+\eta L)^2(1+\gamma_2)}{\eta^2\gamma_2}.$$

Now, taking the total expectation of the last estimate w.r.t. $\mathcal{F}_k$ and using the definition of $C_i$ $(i=1,2,3)$, we have

$$\mathbb{E}\left[\|\mathcal{G}_\eta(\bar{x}^k)\|^2\right] \le C_1\left(\mathbb{E}\left[V_\eta^k(\bar{x}^k)\right]-\mathbb{E}\left[V_\eta^{k+1}(\bar{x}^{k+1})\right]\right)+\frac{C_2}{n}\sum_{i=1}^n\epsilon_{i,k}^2+\frac{C_3}{n}\sum_{i=1}^n\epsilon_{i,k+1}^2.$$

Summing up this inequality from $k:=0$ to $k:=K$, and multiplying the result by $\frac{1}{K+1}$, we get

$$\frac{1}{K+1}\sum_{k=0}^K\mathbb{E}\left[\|\mathcal{G}_\eta(\bar{x}^k)\|^2\right] \le C_1\left(\mathbb{E}\left[V_\eta^0(\bar{x}^0)\right]-\mathbb{E}\left[V_\eta^{K+1}(\bar{x}^{K+1})\right]\right)$$

$$+\frac{1}{n(K+1)}\sum_{k=0}^K\sum_{i=1}^n\left(C_2\epsilon_{i,k}^2+C_3\epsilon_{i,k+1}^2\right).$$

Furthermore, from the initial condition $x_i^0 := x^0$ and $\bar{x}^0 := x^0$, we have $V_\eta^0(\bar{x}^0) = g(x^0)+\frac{1}{n}\sum_{i=1}^n f_i(x^0) = F(x^0)$. In addition, $\mathbb{E}\left[V_\eta^{K+1}(\bar{x}^{K+1})\right] \ge F^\star$ due to (36). Consequently, the last estimate becomes

$$\frac{1}{K+1}\sum_{k=0}^K\mathbb{E}\left[\|\mathcal{G}_\eta(\bar{x}^k)\|^2\right] \le \frac{C_1}{K+1}\left[F(x^0)-F^\star\right]+\frac{1}{n(K+1)}\sum_{k=0}^K\sum_{i=1}^n\left(C_2\epsilon_{i,k}^2+C_3\epsilon_{i,k+1}^2\right),$$

which proves (5).

Finally, let $\tilde{x}^K$ be selected uniformly at random from $\{\bar{x}^0,\cdots,\bar{x}^K\}$ as the output of Algorithm 1. Then, from (5) and $\frac{1}{n}\sum_{i=1}^n\sum_{k=0}^{K+1}\epsilon_{i,k}^2 \le M$ for all $K \ge 0$, we have

$$\mathbb{E}\left[\|\mathcal{G}_\eta(\tilde{x}^K)\|^2\right] = \frac{1}{K+1}\sum_{k=0}^K\mathbb{E}\left[\|\mathcal{G}_\eta(\bar{x}^k)\|^2\right] \le \frac{C_1\left[F(x^0)-F^\star\right]+(C_2+C_3)M}{K+1}.$$

Consequently, to guarantee $\mathbb{E}\left[\|\mathcal{G}_\eta(\tilde{x}^K)\|^2\right] \le \varepsilon^2$, from the last estimate we need to choose $K$ such that $\frac{C_1[F(x^0)-F^\star]+(C_2+C_3)M}{K+1} \le \varepsilon^2$. This condition leads to

$$K+1 \ge \frac{C_1[F(x^0)-F^\star]+(C_2+C_3)M}{\varepsilon^2}.$$

Hence, we can take $K := \left\lfloor\frac{C_1[F(x^0)-F^\star]+(C_2+C_3)M}{\varepsilon^2}\right\rfloor \equiv \mathcal{O}\left(\frac{1}{\varepsilon^2}\right)$ as its lower bound. $\square$

## A.6 Convergence of Algorithm 1 when $\mathbf{p}_i = \frac{1}{n}, i \in [n]$ – The exact variant

***The proof of Corollary 3.1.*** Under the exact variant, we can verify that the choice $\alpha = 1$ and $\eta = \frac{1}{3L}$ satisfies (33). As a result, using $\hat{\mathbf{p}} = \frac{1}{n}$, from (35) we can exactly calculate $\beta = \frac{3L}{5n}$, while $\rho_1 = \rho_2 = 0$. Consequently, (39) leads to

$$\frac{(1+\eta L)^2}{n\eta^2}\sum_{i=1}^n\|x_i^k-\bar{x}^k\|^2 \le \frac{2(1+\eta L)^2}{\eta^2\beta}\left[V_\eta^k(\bar{x}^k)-\mathbb{E}\left[V_\eta^{k+1}(\bar{x}^{k+1})\mid\mathcal{F}_{k-1}\right]\right].$$

Alternatively, using Lemma A.3, we have

$$\|\mathcal{G}_\eta(\bar{x}^k)\|^2 \leq \frac{(1+\eta L)^2}{n\eta^2} \sum_{i=1}^{n} \|x_i^k - \bar{x}^k\|^2.$$

Combining the last two inequalities, we obtain

$$
\begin{aligned}
\|\mathcal{G}_\eta(\bar{x}^k)\|^2 &\leq \frac{2(1+\eta L)^2}{\eta^2\beta} \left[ V_\eta^k(\bar{x}^k) - \mathbb{E}\left[ V_\eta^{k+1}(\bar{x}^{k+1}) \mid \mathcal{F}_{k-1} \right] \right] \\
&= \frac{160Ln}{3} \left[ V_\eta^k(\bar{x}^k) - \mathbb{E}\left[ V_\eta^{k+1}(\bar{x}^{k+1}) \mid \mathcal{F}_{k-1} \right] \right].
\end{aligned}
$$

Now, taking the total expectation of the last estimate w.r.t. $\mathcal{F}_k$, we have

$$\mathbb{E}\left[ \|\mathcal{G}_\eta(\bar{x}^k)\|^2 \right] \leq \frac{160Ln}{3} \left( \mathbb{E}\left[ V_\eta^k(\bar{x}^k) \right] - \mathbb{E}\left[ V_\eta^{k+1}(\bar{x}^{k+1}) \right] \right).$$

Summing this inequality from $k = 0$ to $k = K$, and then multiplying the result by $\frac{1}{K+1}$, we obtain

$$\frac{1}{K+1} \sum_{k=0}^{K} \mathbb{E}\left[ \|\mathcal{G}_\eta(\bar{x}^k)\|^2 \right] \leq \frac{160Ln}{3(K+1)} \left( \mathbb{E}\left[ V_\eta^k(\bar{x}^0) \right] - \mathbb{E}\left[ V_\eta^{k+1}(\bar{x}^{K+1}) \right] \right). \tag{40}$$

Recall that from the initial condition $x_i^0 := x^0$ and $\bar{x}^0 := x^0$, we have $V_\eta^0(\bar{x}^0) = g(x^0) + \frac{1}{n} \sum_{i=1}^{n} f_i(x^0) = F(x^0)$. In addition, $\mathbb{E}\left[ V_\eta^{K+1}(\bar{x}^{K+1}) \right] \geq F^\star$ due to (36). As a result, (40) can be further upper bounded as

$$\frac{1}{K+1} \sum_{k=0}^{K} \mathbb{E}\left[ \|\mathcal{G}_\eta(\bar{x}^k)\|^2 \right] \leq \frac{160Ln}{3(K+1)} \left( F(x^0) - F^\star \right),$$

which exactly proves (6).

Finally, if $\tilde{x}^K$ is selected uniformly at random from $\{\bar{x}^0, \cdots, \bar{x}^K\}$ as the output of Algorithm 1, then we have

$$\mathbb{E}\left[ \|\mathcal{G}_\eta(\tilde{x}^K)\|^2 \right] = \frac{1}{K+1} \sum_{k=0}^{K} \mathbb{E}\left[ \|\mathcal{G}_\eta(\bar{x}^k)\|^2 \right] \leq \frac{160Ln}{3(K+1)} \left( F(x^0) - F^\star \right).$$

Consequently, to guarantee $\mathbb{E}\left[ \|\mathcal{G}_\eta(\tilde{x}^K)\|^2 \right] \leq \varepsilon^2$, from the last estimate we need to choose $K$ such that $\frac{160Ln}{3(K+1)} \left( F(x^0) - F^\star \right) \leq \varepsilon^2$. This condition leads to

$$K + 1 \geq \frac{160Ln[F(x^0) - F^\star]}{3\varepsilon^2}.$$

Hence, we can take $K := \left\lfloor \frac{160Ln[F(x^0)-F^\star]}{3\varepsilon^2} \right\rfloor \equiv \mathcal{O}\left( \frac{1}{\varepsilon^2} \right)$ as its lower bound. $\qquad\square$

### A.7 Convergence of Algorithm 1 under relative accuracies

As suggested by a reviewer, we provide here an analysis of Algorithm 1, when relative accuracies are used to evaluate $\mathrm{prox}_{\eta f_i}$. Such a strategy has been widely used in the literature, including [28, 37]. Let us adopt this concept from [28, Definition 3.3] to our context as follows:

**Definition A.1.** For any $i \in \mathcal{S}_k$, given $x_i^k$ and $y_i^{k+1}$, we say that $x_i^{k+1}$ approximates $\mathrm{prox}_{\eta f_i}(y_i^{k+1})$ up to a **bounded relative error** if there is a constant $\theta_i > 0$ (independent of $k$) such that

$$\|x_i^{k+1} - \mathrm{prox}_{\eta f_i}(y_i^{k+1})\|^2 \leq \varepsilon_{i,k+1}^2 := \theta_i \|x_i^{k+1} - x_i^k\|^2 \tag{41}$$

The following theorem states convergence of Algorithm 1 under the bounded relative error (41).

**Theorem A.1.** *Suppose that Assumptions 2.1, 2.2, and 3.1 hold, and the bounded relative error condition* (41) *in Definition A.1 holds with $\theta_i := \hat{\theta}\mathbf{p}_i$ for a fixed constant $\hat{\theta} > 0$. Let $\{(x_i^k, y_i^k, \hat{x}_i^k, \bar{x}^k)\}$ be generated by Algorithm 1 using a relaxation stepsize $\alpha = 1$ and $x_i^0 := \mathrm{prox}_{\eta f_i}(y_i^0)$ for $i \in [n]$. If $\gamma_4$ and $\hat{\theta}$ are chosen such that $1 - 4\gamma_4 - 8\hat{C}\hat{\theta} > 0$ and $\eta$ is chosen by*

$$0 < \eta < \bar{\eta} := \frac{\sqrt{1+8(1+2\gamma_4)(1-4\gamma_4-8\hat{C}\hat{\theta})}-1}{4L(1+2\gamma_4)}, \tag{42}$$

*where* $\hat{C} := \max\left\{1 + \eta^2 L^2, \frac{2(1+\eta L)^2}{\gamma_4}\right\}$, *then the following bound holds*

$$\frac{1}{K+1}\sum_{k=0}^{K}\mathbb{E}\left[\|\mathcal{G}_\eta(\bar{x}^k)\|^2\right] \leq \frac{\widetilde{C}\left[F(x^0) - F^\star\right]}{(K+1)}, \tag{43}$$

*where* $\widetilde{C} > 0$ *is computed by*

$$\widetilde{C} := \frac{\hat{\mathbf{p}}^2\eta[1 - L\eta - 2L^2\eta^2 - 4\gamma_4(1 + L^2\eta^2) - 8\hat{C}\hat{\theta}]}{4\left[4(1 + L^2\eta^2 + 2\hat{\theta}) + \hat{\mathbf{p}}\hat{\theta}\right](1 + \eta L)^2}. \tag{44}$$

*The remaining conclusions of this theorem are similar to Theorem 3.1, and we omit them here.*

*Proof.* Firstly, starting from (29), using $\alpha = 1$, choosing $\gamma_3 = 1$, and noting that $E_{k+1}^2 := \frac{1}{n}\sum_{i\notin\mathcal{S}_k}\|e_i^k\|^2 + \frac{1}{n}\sum_{i\in\mathcal{S}_k}\|e_i^{k+1}\|^2$, we have

$$V_\eta^{k+1}(\bar{x}^{k+1}) \leq V_\eta^k(\bar{x}^k) - \frac{[1-L\eta-2L^2\eta^2-4\gamma_4(1+L^2\eta^2)]}{2\eta n}\sum_{i\in\mathcal{S}_k}\|x_i^{k+1} - x_i^k\|^2$$
$$+ \frac{(1+\eta^2 L^2)}{\eta n}\left(\sum_{i\notin\mathcal{S}_k}\|e_i^k\|^2 + \sum_{i\in\mathcal{S}_k}\|e_i^{k+1}\|^2\right)$$
$$+ \frac{2(1+\eta L)^2}{\gamma_4\eta n}\sum_{i\in\mathcal{S}_k}[\|e_i^k\|^2 + \|e_i^{k+1}\|^2].$$

If we define $\hat{C} := \max\left\{1 + \eta^2 L^2, \frac{2(1+\eta L)^2}{\gamma_4}\right\}$, then we can further upper bound this estimate as

$$V_\eta^{k+1}(\bar{x}^{k+1}) \leq V_\eta^k(\bar{x}^k) - \frac{[1-L\eta-2L^2\eta^2-4\gamma_4(1+L^2\eta^2)]}{2\eta n}\sum_{i\in\mathcal{S}_k}\|x_i^{k+1} - x_i^k\|^2$$
$$+ \frac{\hat{C}}{n\eta}\left(\sum_{i\notin\mathcal{S}_k}\|e_i^k\|^2 + \sum_{i\in\mathcal{S}_k}\|e_i^{k+1}\|^2\right) + \frac{\hat{C}}{n\eta}\sum_{i\in\mathcal{S}_k}[\|e_i^k\|^2 + \|e_i^{k+1}\|^2]$$
$$= V_\eta^k(\bar{x}^k) - \frac{[1-L\eta-2L^2\eta^2-4\gamma_4(1+L^2\eta^2)]}{2\eta n}\sum_{i\in\mathcal{S}_k}\|x_i^{k+1} - x_i^k\|^2$$
$$+ \frac{\hat{C}}{n\eta}\left(\sum_{i=1}^n\|e_i^k\|^2 + 2\sum_{i\in\mathcal{S}_k}\|e_i^{k+1}\|^2\right)$$
$$\leq V_\eta^k(\bar{x}^k) - \frac{[1-L\eta-2L^2\eta^2-4\gamma_4(1+L^2\eta^2)]}{2\eta n}\sum_{i\in\mathcal{S}_k}\|x_i^{k+1} - x_i^k\|^2$$
$$+ \frac{\hat{C}}{n\eta}\left(\sum_{i=1}^n\|e_i^k\|^2 + 2\sum_{i=1}^n\|e_i^{k+1}\|^2\right)$$
$$\leq V_\eta^k(\bar{x}^k) - \frac{[1-L\eta-2L^2\eta^2-4\gamma_4(1+L^2\eta^2)]}{2\eta n}\sum_{i\in\mathcal{S}_k}\|x_i^{k+1} - x_i^k\|^2$$
$$+ \frac{2\hat{C}}{n\eta}\sum_{i=1}^n\left(\|e_i^k\|^2 + \|e_i^{k+1}\|^2\right).$$

Rearranging terms and noting that $\mathbb{E}\left[\sum_{i\in\mathcal{S}_k}\|x_i^{k+1} - x_i^k\|^2 \mid \mathcal{F}_{k-1}\right] = \sum_{i=1}^n \mathbf{p}_i\|x_i^{k+1} - x_i^k\|^2$, we obtain from the last estimate that

$$\frac{[1-L\eta-2L^2\eta^2-4\gamma_4(1+L^2\eta^2)]}{2\eta n}\sum_{i=1}^n\mathbf{p}_i\|x_i^{k+1} - x_i^k\|^2 \leq V_\eta^k(\bar{x}^k) - V_\eta^{k+1}(\bar{x}^{k+1})$$
$$+ \frac{2\hat{C}}{n\eta}\sum_{i=1}^n\left(\|e_i^k\|^2 + \|e_i^{k+1}\|^2\right).$$

Now, taking the total expectation of the last inequality w.r.t. $\mathcal{F}_k$, we have

$$\frac{[1-L\eta-2L^2\eta^2-4\gamma_4(1+L^2\eta^2)]}{2\eta n}\sum_{i=1}^n\mathbf{p}_i\mathbb{E}\left[\|x_i^{k+1} - x_i^k\|^2\right]$$
$$\leq \mathbb{E}\left[V_\eta^k(\bar{x}^k)\right] - \mathbb{E}\left[V_\eta^{k+1}(\bar{x}^{k+1})\right] + \frac{2\hat{C}}{n\eta}\sum_{i=1}^n\mathbb{E}\left[\|e_i^k\|^2 + \|e_i^{k+1}\|^2\right].$$

Summing this inequality from $k = 0$ to $k = K$, we get

$$\frac{[1-L\eta-2L^2\eta^2-4\gamma_4(1+L^2\eta^2)]}{2\eta n}\sum_{k=0}^K\sum_{i=1}^n\mathbf{p}_i\mathbb{E}\left[\|x_i^{k+1} - x_i^k\|^2\right] \leq \mathbb{E}\left[V_\eta^0(\bar{x}^0)\right] - \mathbb{E}\left[V_\eta^{K+1}(\bar{x}^{K+1})\right]$$
$$+ \frac{2\hat{C}}{n\eta}\sum_{k=0}^K\sum_{i=1}^n\mathbb{E}\left[\|e_i^k\|^2 + \|e_i^{k+1}\|^2\right].$$

If we choose $\varepsilon_{i,0} = 0$ for $i \in [n]$, then the last estimate reduces to

$$\frac{[1-L\eta-2L^2\eta^2-4\gamma_4(1+L^2\eta^2)]}{2\eta n}\sum_{k=0}^K\sum_{i=1}^n\mathbf{p}_i\mathbb{E}\left[\|x_i^{k+1} - x_i^k\|^2\right] \leq \mathbb{E}\left[V_\eta^0(\bar{x}^0)\right] - \mathbb{E}\left[V_\eta^{K+1}(\bar{x}^{K+1})\right]$$
$$+ \frac{4\hat{C}}{n\eta}\sum_{k=0}^K\sum_{i=1}^n\mathbb{E}\left[\|e_i^{k+1}\|^2\right]. \tag{45}$$

From (41) in Definition A.1, we have $\|e_i^{k+1}\|^2 = \|x_i^{k+1} - \text{prox}_{\eta f_i}(y_i^{k+1})\|^2 \leq \varepsilon_{i,k+1}^2 := \theta_i\|x_i^{k+1} - x_i^k\|^2$. Using this condition in (45), we have

$$\frac{[1-L\eta-2L^2\eta^2-4\gamma_4(1+L^2\eta^2)]}{2\eta n} \sum_{k=0}^{K}\sum_{i=1}^{n}\mathbf{p}_i\mathbb{E}\left[\|x_i^{k+1}-x_i^k\|^2\right] \leq \mathbb{E}\left[V_\eta^0(\bar{x}^0)\right] - \mathbb{E}\left[V_\eta^{K+1}(\bar{x}^{K+1})\right]$$
$$+ \frac{4\hat{C}}{n\eta}\sum_{k=0}^{K}\sum_{i=1}^{n}\theta_i\mathbb{E}\left[\|x_i^{k+1}-x_i^k\|^2\right].$$

Now, we can choose $\theta_i$ such that $\theta_i = \hat{\theta}\mathbf{p}_i$ for given $\hat{\theta} > 0$. Plugging this choice of $\theta_i$ into the last estimate, we have

$$\frac{[1-L\eta-2L^2\eta^2-4\gamma_4(1+L^2\eta^2)]}{2\eta n}\sum_{k=0}^{K}\sum_{i=1}^{n}\mathbf{p}_i\mathbb{E}\left[\|x_i^{k+1}-x_i^k\|^2\right] \leq \mathbb{E}\left[V_\eta^0(\bar{x}^0)\right] - \mathbb{E}\left[V_\eta^{K+1}(\bar{x}^{K+1})\right]$$
$$+ \frac{4\hat{C}\hat{\theta}}{n\eta}\sum_{k=0}^{K}\sum_{i=1}^{n}\mathbf{p}_i\mathbb{E}\left[\|x_i^{k+1}-x_i^k\|^2\right].$$

Rearranging terms in the above estimate, we arrive at

$$\frac{[1-L\eta-2L^2\eta^2-4\gamma_4(1+L^2\eta^2)-8\hat{C}\hat{\theta}]}{2\eta n}\sum_{k=0}^{K}\sum_{i=1}^{n}\mathbf{p}_i\mathbb{E}\left[\|x_i^{k+1}-x_i^k\|^2\right] \leq \mathbb{E}\left[V_\eta^0(\bar{x}^0)\right] - \mathbb{E}\left[V_\eta^{K+1}(\bar{x}^{K+1})\right].$$

From the initial condition $x_i^0 := x^0$ and $\bar{x}^0 := x^0$, we have $V_\eta^0(\bar{x}^0) = g(x^0) + \frac{1}{n}\sum_{i=1}^{n}f_i(x^0) = F(x^0)$. In addition, $\mathbb{E}\left[V_\eta^{K+1}(\bar{x}^{K+1})\right] \geq F^\star$ due to (36). Using these conditions, the last estimate can be further upper bounded by

$$\frac{\hat{\mathbf{p}}[1 - L\eta - 2L^2\eta^2 - 4\gamma_4(1 + L^2\eta^2) - 8\hat{C}\hat{\theta}]}{2\eta n}\sum_{k=0}^{K}\sum_{i=1}^{n}\mathbb{E}\left[\|x_i^{k+1}-x_i^k\|^2\right] \leq F(x^0) - F^\star, \quad (46)$$

where we have used $\mathbf{p}_i \geq \hat{\mathbf{p}}$ for all $i \in [n]$.

Now, we need to choose $\eta$ and $\hat{\theta}$ such that $1 - L\eta - 2L^2\eta^2 - 4\gamma_4(1 + L^2\eta^2) - 8\hat{C}\hat{\theta} > 0$. First, we need to choose $\gamma_4 > 0$ and $\hat{\theta} > 0$ such that $1 - 4\gamma_4 - 8\hat{C}\hat{\theta} > 0$. Then, the condition for $\eta$ is

$$0 < \eta < \bar{\eta} := \frac{\sqrt{1+8(1+2\gamma_4)(1-4\gamma_4-8\hat{C}\hat{\theta})}-1}{4L(1+2\gamma_4)}.$$

Next, we connect the term $\|x_i^{k+1} - x_i^k\|^2$ with $\|\mathcal{G}_\eta(\bar{x}^k)\|^2$ as follows. From (22) with $\alpha = 1$ and $\gamma_1 = 1$, we have

$$\frac{1}{4(1+L^2\eta^2)}\sum_{i\in\mathcal{S}_k}\|\bar{x}^k-x_i^k\|^2 \leq \sum_{i\in\mathcal{S}_k}\left[\|x_i^{k+1}-x_i^k\|^2 + \frac{1}{(1+L^2\eta^2)}\left(\|e_i^{k+1}\|^2 + \|e_i^k\|^2\right)\right].$$

Taking expecatation w.r.t. $\mathcal{S}_k$ given $\mathcal{F}_{k-1}$, and then taking full expectation, we obtain

$$\frac{1}{4(1+L^2\eta^2)}\sum_{i=1}^{n}\mathbf{p}_i\mathbb{E}\left[\|\bar{x}^k-x_i^k\|^2\right] \leq \sum_{i=1}^{n}\mathbf{p}_i\mathbb{E}\left[\|x_i^{k+1}-x_i^k\|^2\right]$$
$$+ \frac{1}{(1+L^2\eta^2)}\sum_{i=1}^{n}\mathbf{p}_i\mathbb{E}\left[\|e_i^{k+1}\|^2 + \|e_i^k\|^2\right]$$
$$\leq \sum_{i=1}^{n}\mathbf{p}_i\mathbb{E}\left[\|x_i^{k+1}-x_i^k\|^2\right]$$
$$+ \frac{1}{(1+L^2\eta^2)}\sum_{i=1}^{n}\mathbb{E}\left[\|e_i^{k+1}\|^2 + \|e_i^k\|^2\right].$$

Summing this inequality from $k = 0$ to $k = K$, we get

$$\frac{1}{4(1+L^2\eta^2)}\sum_{k=0}^{K}\sum_{i=1}^{n}\mathbf{p}_i\mathbb{E}\left[\|\bar{x}^k-x_i^k\|^2\right] \leq \sum_{k=0}^{K}\sum_{i=1}^{n}\mathbf{p}_i\mathbb{E}\left[\|x_i^{k+1}-x_i^k\|^2\right]$$
$$+ \frac{1}{(1+L^2\eta^2)}\sum_{k=0}^{K}\sum_{i=1}^{n}\mathbb{E}\left[\|e_i^{k+1}\|^2 + \|e_i^k\|^2\right].$$

Using the condition that $\epsilon_{i,0} = 0$, similar to (45), we have

$$
\begin{aligned}
\frac{1}{4(1+L^2\eta^2)} \sum_{k=0}^{K} \sum_{i=1}^{n} \mathbf{p}_i \mathbb{E}\left[\|\bar{x}^k - x_i^k\|^2\right] &\leq \sum_{k=0}^{K} \sum_{i=1}^{n} \mathbf{p}_i \mathbb{E}\left[\|x_i^{k+1} - x_i^k\|^2\right] \\
&\quad + \frac{2}{(1+L^2\eta^2)} \sum_{k=0}^{K} \sum_{i=1}^{n} \mathbb{E}\left[\|e_i^{k+1}\|^2\right] \\
&\leq \sum_{k=0}^{K} \sum_{i=1}^{n} \mathbf{p}_i \mathbb{E}\left[\|x_i^{k+1} - x_i^k\|^2\right] \\
&\quad + \frac{2}{(1+L^2\eta^2)} \sum_{k=0}^{K} \sum_{i=1}^{n} \theta_i \mathbb{E}\left[\|x_i^{k+1} - x_i^k\|^2\right] \\
&\leq \frac{1+L^2\eta^2+2\hat{\theta}}{(1+L^2\eta^2)} \sum_{k=0}^{K} \sum_{i=1}^{n} \mathbf{p}_i \mathbb{E}\left[\|x_i^{k+1} - x_i^k\|^2\right].
\end{aligned}
$$

In fact, we can further bound this estimate as

$$
\frac{\hat{\mathbf{p}}}{4(1+L^2\eta^2)} \sum_{k=0}^{K} \sum_{i=1}^{n} \mathbb{E}\left[\|\bar{x}^k - x_i^k\|^2\right] \leq \frac{1+L^2\eta^2+2\hat{\theta}}{(1+L^2\eta^2)} \sum_{k=0}^{K} \sum_{i=1}^{n} \mathbb{E}\left[\|x_i^{k+1} - x_i^k\|^2\right],
$$

where we have used $\hat{\mathbf{p}} \leq \mathbf{p}_i \leq 1$. Next, multiply both sides of this inequality by $\frac{8(1+L^2\eta^2)(1+\eta L)^2}{\hat{\mathbf{p}}\eta^2 n}$, we obtain

$$
\frac{2(1+\eta L)^2}{n\eta^2} \sum_{k=0}^{K} \sum_{i=1}^{n} \mathbb{E}\left[\|\bar{x}^k - x_i^k\|^2\right] \leq \frac{8(1+L^2\eta^2+2\hat{\theta})(1+\eta L)^2}{\hat{\mathbf{p}}\eta^2 n} \sum_{k=0}^{K} \sum_{i=1}^{n} \mathbb{E}\left[\|x_i^{k+1} - x_i^k\|^2\right]. \tag{47}
$$

Furthermore, from (23), choosing $\gamma_2 = 1$ and summing the result from $k = 0$ to $k = K$, we get

$$
\begin{aligned}
\sum_{k=0}^{K} \mathbb{E}\left[\|\mathcal{G}_\eta(\bar{x}^k)\|^2\right] &\leq \frac{2(1+\eta L)^2}{n\eta^2} \sum_{k=0}^{K} \sum_{i=1}^{n} \mathbb{E}\left[\|x_i^k - \bar{x}^k\|^2\right] \\
&\quad + \frac{2(1+\eta L)^2}{n\eta^2} \sum_{k=0}^{K} \sum_{i=1}^{n} \mathbb{E}\left[\|e_i^k\|^2\right] \\
&\leq \frac{2(1+\eta L)^2}{n\eta^2} \sum_{k=0}^{K} \sum_{i=1}^{n} \mathbb{E}\left[\|x_i^k - \bar{x}^k\|^2\right] \\
&\quad + \frac{2(1+\eta L)^2}{n\eta^2} \sum_{k=0}^{K} \sum_{i=1}^{n} \theta_i \mathbb{E}\left[\|x_i^{k+1} - x_i^k\|^2\right] \\
&= \frac{2(1+\eta L)^2}{n\eta^2} \sum_{k=0}^{K} \sum_{i=1}^{n} \mathbb{E}\left[\|x_i^k - \bar{x}^k\|^2\right] \\
&\quad + \frac{2(1+\eta L)^2 \hat{\theta}}{n\eta^2} \sum_{k=0}^{K} \sum_{i=1}^{n} \mathbf{p}_i \mathbb{E}\left[\|x_i^{k+1} - x_i^k\|^2\right]
\end{aligned} \tag{48}
$$

where the last equality comes from the fact that $\theta_i = \hat{\theta}\mathbf{p}_i$.

Now, plugging (47) into (48) and using $\mathbf{p}_i \leq 1$, we can get

$$
\begin{aligned}
\sum_{k=0}^{K} \mathbb{E}\left[\|\mathcal{G}_\eta(\bar{x}^k)\|^2\right] &\leq \left[\frac{8[1+L^2\eta^2+2\hat{\theta}](1+\eta L)^2}{\hat{\mathbf{p}}\eta^2 n} + \frac{2(1+\eta L)^2\hat{\theta}}{n\eta^2}\right] \sum_{k=0}^{K} \sum_{i=1}^{n} \mathbb{E}\left[\|x_i^{k+1} - x_i^k\|^2\right] \\
&= \frac{2[4(1+L^2\eta^2+2\hat{\theta})+\hat{\mathbf{p}}\hat{\theta}](1+\eta L)^2}{\hat{\mathbf{p}}n\eta^2} \sum_{k=0}^{K} \sum_{i=1}^{n} \mathbb{E}\left[\|x_i^{k+1} - x_i^k\|^2\right].
\end{aligned} \tag{49}
$$

From the definition of $\widetilde{C}$ in (44), we can verify that

$$
\frac{\hat{\mathbf{p}}[1 - L\eta - 2L^2\eta^2 - 4\gamma_4(1+L^2\eta^2) - 8\hat{C}\hat{\theta}]}{2\eta n \widetilde{C}} = \frac{2\left[4(1+L^2\eta^2+2\hat{\theta}) + \hat{\mathbf{p}}\hat{\theta}\right](1+\eta L)^2}{\hat{\mathbf{p}}n\eta^2}.
$$

Next, multiplying both sides of (46) by $\frac{1}{\widetilde{C}}$, and then using (49), we obtain

$$
\begin{aligned}
\sum_{k=0}^{K} \mathbb{E}\left[\|\mathcal{G}_\eta(\bar{x}^k)\|^2\right] &\leq \frac{2[4(1+L^2\eta^2+\hat{\theta})+\hat{\mathbf{p}}\hat{\theta}](1+\eta L)^2}{\hat{\mathbf{p}}n\eta^2} \sum_{k=0}^{K} \sum_{i=1}^{n} \mathbb{E}\left[\|x_i^{k+1} - x_i^k\|^2\right] \\
&\overset{(46)}{\leq} \widetilde{C}\left[F(x^0) - F^\star\right].
\end{aligned}
$$

Finally, multiplying both sides of this inequality by $\frac{1}{K+1}$, we obtain (43). $\qquad\square$

## B  Analysis of Algorithm 2: The Asynchronous Variant — asyncFedDR

This section provides the full proof of Lemma B.2 and Theorem 4.1 in the main text. However, let us first discuss an asynchronous implementation of Algorithm 2 and present the full description of our probabilistic models based on [5] used in Section 4.

## B.1   Asynchronous implementation: Dual-memory approach

Let us provide more details on the implementation of our asynchronous algorithm. When a user finishes its local update, the updated model (or model difference) is sent to the server for a proximal aggregation step. When the server is performing a proximal aggregation step, other users might need to read from the global model. To allow concurrent read/write operations, one easy method is to have two models stored on the server, denoted as model 1 and model 2. At any given time, one model is on "read" state (it is supposed to be read from) and the other will be on "write" state (it will be written on when the server finishes aggregation). Suppose model 1 is on a "read" state and model 2 is on a "write" state, then all users can read from model 1. When the server completes the proximal aggregation, model 2 becomes the latest model and it will change to a "read" state while model 1 is on a "write" state. This implementation detail is also discussed in [34], which is termed by a *dual-memory approach*.

## B.2   Probabilistic model

Let $\xi^k := (i_k, d^k)$ be a realization of a joint random vector $\hat{\xi}^k := (\hat{i}_k, \hat{d}^k)$ of the user index $\hat{i}_k \in [n]$ and the delay vector $\hat{d}^k = (\hat{d}_1^k, \cdots, \hat{d}_n^k) \in \mathcal{D} := \{0, 1, \cdots, \tau\}^n$ presented at the current iteration $k$. We consider $k + 1$ random vectors $\hat{\xi}^l$ ($0 \le l \le k$) that form a concatenate random vector $\hat{\xi}^{0:k} := (\hat{\xi}^0, \cdots, \hat{\xi}^k)$. We also use $\xi^{0:k} = (\xi^0, \xi^1, \cdots, \xi^k)$ for $k + 1$ possible values of the random vector $\hat{\xi}^{0:k}$. Let $\Omega$ be the sample space of all sequences $\omega := \{(i_k, d^k)\}_{k \ge 0} \equiv \{\xi^k\}_{k \ge 0}$. We define a cylinder $\mathcal{C}_k(\xi^{0:k}) := \{\omega \in \Omega : (\omega_0, \cdots, \omega_k) = \xi^{0:k}\}$ as a subset in $\Omega$ and $\mathcal{C}_k$ is the set of all possible subsets $\mathcal{C}_k(\xi^{0:k})$ when $\xi^t$, $t = 0, \cdots, k$, take all possible values, where $\omega_l$ is the $l$-th coordinate of $\omega$. Note that $\{\mathcal{C}_k\}_{k \ge 0}$ forms a partition of $\Omega$ and measurable. Let $\mathcal{F}_k := \sigma(\mathcal{C}_k)$ be the $\sigma$-algebra generated by $\mathcal{C}_k$ and $\mathcal{F} := \sigma(\cup_{k=0}^\infty \mathcal{C}_k)$. Clearly, $\{\mathcal{F}_k\}_{k \ge 0}$ forms a filtration such that $\mathcal{F}_k \subseteq \mathcal{F}_{k+1} \subseteq \cdots \subseteq \mathcal{F}$ for $k \ge 0$ that is sufficient to cope with the evolution of Algorithm 2.

For each $\mathcal{C}_k(\xi^{0:k})$ we also equip with a probability $\mathbf{p}(\xi^{0:k}) := \mathbb{P}(\mathcal{C}_k(\xi^{0:k}))$. Then, $(\Omega, \mathcal{F}, \mathbb{P})$ forms a probability space. Our conditional probability is defined as $\mathbf{p}((i, d) \mid \xi^{0:k}) := \mathbb{P}(\mathcal{C}_{k+1}(\xi^{0:k+1}))/\mathbb{P}(\mathcal{C}_k(\xi^{0:k}))$, where we set $\mathbf{p}((i, d) \mid \xi^{0:k}) := 0$ if $\mathbf{p}(\xi^{0:k}) = 0$. We do not need to know these probabilities in advance. They are determined based on the particular system such as hardware architecture, software implementation, asynchrony, and our strategy for selecting active user.

Now, if $X$ is a random variable defined on $\Omega$, then as shown in [5], we have

$$\mathbb{E}[X \mid \mathcal{F}_k] = \sum_{(i,d) \in [n] \times \mathcal{D}} \mathbf{p}((i, d) \mid \xi^{0:k}) X(\xi^{0:k}, (i, d)). \tag{50}$$

Note from Assumption 4.1 that

$$\mathbf{p}(i \mid \xi^{0:k}) := \sum_{d \in \mathcal{D}} \mathbf{p}((i, d) \mid \xi^{0:k}) \ge \hat{\mathbf{p}}. \tag{51}$$

Our probability model described above allows us to handle a variety class of asynchronous algorithms derived from the DR splitting scheme. Here, we do not make independent assumption between the active user $\hat{i}_k$ and the delay vector $\hat{d}^k$.

## B.3   Preparatory lemmas

For the asynchronous algorithm, Algorithm 2, the following facts hold.

- For $x_i^k$ and $y_i^k$ updated by Algorithm 2, since $\mathcal{S}_k = \{i_k\}$ and the update of $y_i^k$ and $x_i^k$ remain the same as in Algorithm 1 when the error $e_i^k = 0$, the relation (21) remains true, i.e. $y_i^k = x_i^k + \eta \nabla f_i(x_i^k)$ and $\hat{x}_i^k = 2x_i^k - y_i^k$ for all $i \in [n]$ and $k \ge 0$.
- Let $\bar{\mathbf{x}}^{k-d^k} := [\bar{x}^{k-d_1^k}, \bar{x}^{k-d_2^k}, \cdots, \bar{x}^{k-d_n^k}]$ be a delayed copy of the vector $\bar{\mathbf{x}}^k := [\bar{x}^k, \cdots, \bar{x}^k] \in \mathbb{R}^{np}$. Since at each iteration $k$, there is only one block $i_k$ being updated, as shown in [5, 34], for all $i \in [n]$, we can write

$$\bar{x}^{k-d_i^k} = \bar{x}^k + \sum_{l \in J_i^k} (\bar{x}^l - \bar{x}^{l+1}), \tag{52}$$

where $J_i^k := \{k - d_i^k, k - d_i^k + 1, \cdots, k - 1\} \subseteq \{k - \tau, \cdots, k - 1\}$.

These facts will be repeatedly used in the sequel.

Now, let us first prove the following lemma to provide a key estimate for establishing Lemma B.2.

**Lemma B.1** (Sure descent)**.** *Suppose that Assumptions 2.1, 2.2, and 4.1 hold for* (1)*. Let* $\{(x_i^k, y_i^k, \hat{x}_i^k, \tilde{x}^k, \bar{x}^k)\}$ *be generated by Algorithm 2 and* $V_\eta^k(\cdot)$ *be defined as in* (25)*. Then, for all* $k \geq 0$*, the following estimate holds:*

$$V_\eta^{k+1}(\bar{x}^{k+1}) + \tfrac{\tau}{n\eta}\sum_{l=k+1-\tau}^k (l - k + \tau)\|\bar{x}^{l+1} - \bar{x}^l\|^2 \leq V_\eta^k(\bar{x}^k) - \tfrac{\rho}{2}\|x_{i_k}^{k+1} - x_{i_k}^k\|^2$$
$$+ \tfrac{\tau}{n\eta}\sum_{l=k-\tau}^{k-1}(l - (k-1) + \tau)\|\bar{x}^{l+1} - \bar{x}^l\|^2, \tag{53}$$

*where*

$$\rho := \begin{cases} \dfrac{2(1-\alpha)-(2+\alpha)L^2\eta^2 - L\alpha\eta}{\alpha\eta n} & \text{if} \quad 2\tau^2 \leq n, \\[2mm] \dfrac{n^2[2(1-\alpha)-(2+\alpha)L^2\eta^2 - L\alpha\eta] - \alpha(1+\eta^2 L^2)(2\tau^2 - n)}{\alpha\eta n^3} & \text{otherwise.} \end{cases}$$

*Proof.* Let $V_\eta^k$ be defined by (25). For $(x_i^k, \hat{x}_i^k, y_i^k)$ updated as in Algorithm 2, the results of Lemma A.1 still hold true. Hence, (26) still holds for Algorithm 2 with $\gamma_3 = 0$ and $E_{k+1}^2 = 0$, i.e.:

$$V_\eta^{k+1}(\bar{x}^{k+1}) \leq g(\bar{x}^k) + \tfrac{1}{n}\sum_{i=1}^n \left[f_i(x_i^{k+1}) + \langle \nabla f_i(x_i^{k+1}), \bar{x}^k - x_i^{k+1}\rangle + \tfrac{1}{2\eta}\|\bar{x}^k - x_i^{k+1}\|^2\right]$$
$$- \tfrac{1}{2\eta}\|\bar{x}^{k+1} - \bar{x}^k\|^2.$$

Using this inequality, the update of $x_{i_k}^{k+1}$ for $i = i_k$, and $x_i^{k+1} = x_i^k$ for $i \neq i_k$, we can expand

$$V_\eta^{k+1}(\bar{x}^{k+1}) \leq g(\bar{x}^k) + \tfrac{1}{n}\sum_{i\neq i_k}\left[f_i(x_i^k) + \langle\nabla f_i(x_i^k), \bar{x}^k - x_i^k\rangle + \tfrac{1}{2\eta}\|\bar{x}^k - x_i^k\|^2\right]$$
$$+ \tfrac{1}{n}\left[f_{i_k}(x_{i_k}^{k+1}) + \langle\nabla f_{i_k}(x_{i_k}^{k+1}), x_{i_k}^k - x_{i_k}^{k+1}\rangle\right] + \tfrac{1}{n}\langle\nabla f_{i_k}(x_{i_k}^{k+1}), \bar{x}^k - x_{i_k}^k\rangle \tag{54}$$
$$+ \tfrac{1}{2\eta n}\|\bar{x}^k - x_{i_k}^k + x_{i_k}^k - x_{i_k}^{k+1}\|^2 - \tfrac{1}{2\eta}\|\bar{x}^{k+1} - \bar{x}^k\|^2.$$

Now, by the $L$-smoothness of $f_{i_k}$, we have

$$f_{i_k}(x_{i_k}^{k+1}) + \langle\nabla f_{i_k}(x_{i_k}^{k+1}), x_{i_k}^k - x_{i_k}^{k+1}\rangle \leq f_{i_k}(x_{i_k}^k) + \tfrac{L}{2}\|x_{i_k}^k - x_{i_k}^{k+1}\|^2.$$

Plugging this inequality into (54) and expanding the third last term of (54), we obtain

$$V_\eta^{k+1}(\bar{x}^{k+1}) \leq g(\bar{x}^k) + \tfrac{1}{n}\sum_{i\neq i_k}\left[f_i(x_i^k) + \langle\nabla f_i(x_i^k), \bar{x}^k - x_i^k\rangle + \tfrac{1}{2\eta}\|\bar{x}^k - x_i^k\|^2\right]$$
$$+ \tfrac{1}{n}f_{i_k}(x_{i_k}^k) + \tfrac{L}{2n}\|x_{i_k}^{k+1} - x_{i_k}^k\|^2 + \tfrac{1}{n}\langle\nabla f_{i_k}(x_{i_k}^{k+1}), \bar{x}^k - x_{i_k}^k\rangle$$
$$+ \tfrac{1}{2\eta n}\|\bar{x}^k - x_{i_k}^k\|^2 + \tfrac{1}{2\eta n}\|x_{i_k}^{k+1} - x_{i_k}^k\|^2 + \tfrac{1}{\eta n}\langle x_{i_k}^{k+1} - x_{i_k}^k, x_{i_k}^k - \bar{x}^k\rangle$$
$$- \tfrac{1}{2\eta}\|\bar{x}^{k+1} - \bar{x}^k\|^2$$
$$= g(\bar{x}^k) + \tfrac{1}{n}\sum_{i=1}^n\left[f_i(x_i^k) + \langle\nabla f_i(x_i^k), \bar{x}^k - x_i^k\rangle + \tfrac{1}{2\eta}\|\bar{x}^k - x_i^k\|^2\right]$$
$$+ \tfrac{(1+\eta L)}{2n\eta}\|x_{i_k}^{k+1} - x_{i_k}^k\|^2 + \tfrac{1}{n}\langle\nabla f_{i_k}(x_{i_k}^{k+1}) - \nabla f_{i_k}(x_{i_k}^k), \bar{x}^k - x_{i_k}^k\rangle \tag{55}$$
$$+ \tfrac{1}{\eta n}\langle x_{i_k}^{k+1} - x_{i_k}^k, x_{i_k}^k - \bar{x}^k\rangle - \tfrac{1}{2\eta}\|\bar{x}^{k+1} - \bar{x}^k\|^2$$
$$\overset{(25)}{=} V_\eta^k(\bar{x}^k) + \tfrac{1}{n}\langle\nabla f_{i_k}(x_{i_k}^{k+1}) - \nabla f_{i_k}(x_{i_k}^k), \bar{x}^{k-d_{i_k}^k} - x_{i_k}^k\rangle$$
$$+ \tfrac{1}{n}\langle\nabla f_{i_k}(x_{i_k}^{k+1}) - \nabla f_{i_k}(x_{i_k}^k), \bar{x}^k - \bar{x}^{k-d_{i_k}^k}\rangle + \tfrac{(1+L\eta)}{2\eta n}\|x_{i_k}^{k+1} - x_{i_k}^k\|^2$$
$$+ \tfrac{1}{\eta n}\langle x_{i_k}^{k+1} - x_{i_k}^k, x_{i_k}^k - \bar{x}^{k-d_{i_k}^k}\rangle + \tfrac{1}{\eta n}\langle x_{i_k}^{k+1} - x_{i_k}^k, \bar{x}^{k-d_{i_k}^k} - \bar{x}^k\rangle$$
$$- \tfrac{1}{2\eta}\|\bar{x}^{k+1} - \bar{x}^k\|^2.$$

From $y_{i_k}^{k+1} := y_{i_k}^k + \alpha(\bar{x}^{k-d_{i_k}^k} - x_{i_k}^k)$ at Step 5 of Algorithm 2 and the relation (21), we have

$$\bar{x}^{k-d_{i_k}^k} - x_{i_k}^k = \tfrac{1}{\alpha}(y_{i_k}^{k+1} - y_{i_k}^k) \overset{(21)}{=} \tfrac{1}{\alpha}(x_{i_k}^{k+1} - x_{i_k}^k) + \tfrac{\eta}{\alpha}(\nabla f_{i_k}(x_{i_k}^{k+1}) - \nabla f_{i_k}(x_{i_k}^k)). \tag{56}$$

This relation leads to

$$\frac{1}{n}\langle \nabla f_{i_k}(x_{i_k}^{k+1}) - \nabla f_{i_k}(x_{i_k}^k), \bar{x}^{k-d_{i_k}^k} - x_{i_k}^k \rangle = \frac{1}{\alpha n}\langle \nabla f_{i_k}(x_{i_k}^{k+1}) - \nabla f_{i_k}(x_{i_k}^k), x_{i_k}^{k+1} - x_{i_k}^k \rangle$$
$$+ \frac{\eta}{\alpha n}\|\nabla f_{i_k}(x_{i_k}^{k+1}) - \nabla f_{i_k}(x_{i_k}^k)\|^2, \quad (57)$$

and

$$\frac{1}{\eta n}\langle x_{i_k}^{k+1} - x_{i_k}^k, x_{i_k}^k - \bar{x}^{k-d_{i_k}^k} \rangle = -\frac{1}{\alpha n}\langle \nabla f_{i_k}(x_{i_k}^{k+1}) - \nabla f_{i_k}(x_{i_k}^k), x_{i_k}^{k+1} - x_{i_k}^k \rangle$$
$$- \frac{1}{\eta \alpha n}\|x_{i_k}^{k+1} - x_{i_k}^k\|^2. \quad (58)$$

Substituting (57) and (58) into (55), we obtain

$$V_\eta^{k+1}(\bar{x}^{k+1}) \leq V_\eta^k(\bar{x}^k) + \frac{(1+L\eta)}{2\eta n}\|x_{i_k}^{k+1} - x_{i_k}^k\|^2 + \frac{\eta}{\alpha n}\|\nabla f_{i_k}(x_{i_k}^{k+1}) - \nabla f_{i_k}(x_{i_k}^k)\|^2$$
$$+ \frac{1}{n}\langle \nabla f_{i_k}(x_{i_k}^{k+1}) - \nabla f_{i_k}(x_{i_k}^k), \bar{x}^k - \bar{x}^{k-d_{i_k}^k}\rangle + \frac{1}{\eta n}\langle x_{i_k}^{k+1} - x_{i_k}^k, \bar{x}^{k-d_{i_k}^k} - \bar{x}^k\rangle$$
$$- \frac{1}{\eta \alpha n}\|x_{i_k}^{k+1} - x_{i_k}^k\|^2 - \frac{1}{2\eta}\|\bar{x}^{k+1} - \bar{x}^k\|^2$$
$$\overset{(2)}{\leq} V_\eta^k(\bar{x}^k) + \frac{\alpha(L\eta+1)-2}{2\eta\alpha n}\|x_{i_k}^{k+1} - x_{i_k}^k\|^2 + \frac{\eta L^2}{\alpha n}\|x_{i_k}^{k+1} - x_{i_k}^k\|^2$$
$$+ \frac{1}{n}\langle \nabla f_{i_k}(x_{i_k}^{k+1}) - \nabla f_{i_k}(x_{i_k}^k), \bar{x}^k - \bar{x}^{k-d_{i_k}^k}\rangle + \frac{1}{\eta n}\langle x_{i_k}^{k+1} - x_{i_k}^k, \bar{x}^{k-d_{i_k}^k} - \bar{x}^k\rangle$$
$$- \frac{1}{2\eta}\|\bar{x}^{k+1} - \bar{x}^k\|^2.$$

Next, using Young's inequality twice in the above estimate, we can further expand

$$V_\eta^{k+1}(\bar{x}^{k+1}) \leq V_\eta^k(\bar{x}^k) + \frac{\alpha(L\eta+1)+2L^2\eta^2-2}{2\eta\alpha n}\|x_{i_k}^{k+1} - x_{i_k}^k\|^2 + \frac{\eta}{2n}\|\nabla f_{i_k}(x_{i_k}^{k+1}) - \nabla f_{i_k}(x_{i_k}^k)\|^2$$
$$+ \frac{1}{2\eta n}\|\bar{x}^k - \bar{x}^{k-d_{i_k}^k}\|^2 + \frac{1}{2\eta n}\|x_{i_k}^{k+1} - x_{i_k}^k\|^2 + \frac{1}{2\eta n}\|\bar{x}^k - \bar{x}^{k-d_{i_k}^k}\|^2$$
$$- \frac{1}{2\eta}\|\bar{x}^{k+1} - \bar{x}^k\|^2$$
$$\overset{(2)}{\leq} V_\eta^k(\bar{x}^k) + \frac{[\alpha(L\eta+2)+2L^2\eta^2-2]}{2\alpha\eta n}\|x_{i_k}^{k+1} - x_{i_k}^k\|^2 + \frac{L^2\eta}{2n}\|x_{i_k}^{k+1} - x_{i_k}^k\|^2 \qquad (59)$$
$$+ \frac{1}{\eta n}\|\bar{x}^k - \bar{x}^{k-d_{i_k}^k}\|^2 - \frac{1}{2\eta}\|\bar{x}^{k+1} - \bar{x}^k\|^2$$
$$= V_\eta^k(\bar{x}^k) + \frac{[\alpha(L^2\eta^2+L\eta+2)+2L^2\eta^2-2]}{2\alpha\eta n}\|x_{i_k}^{k+1} - x_{i_k}^k\|^2 - \frac{1}{2\eta}\|\bar{x}^{k+1} - \bar{x}^k\|^2$$
$$+ \frac{1}{n\eta}\|\bar{x}^{k-d_{i_k}^k} - \bar{x}^k\|^2.$$

Using (52), we can bound $\|\bar{x}^{k-d_{i_k}^k} - \bar{x}^k\|^2$ as follows:

$$\|\bar{x}^{k-d_{i_k}^k} - \bar{x}^k\|^2 \overset{(52)}{=} \left\|\sum_{l\in J_{i_k}^k}(\bar{x}^l - \bar{x}^{l+1})\right\|^2$$
$$\leq d_{i_k}^k \sum_{l=k-d_{i_k}^k}^{k-1}\|\bar{x}^{l+1} - \bar{x}^l\|^2 \quad \text{(Young's inequality and the definition of } J_{i_k}^k\text{)}$$
$$\leq \tau \sum_{l=k-\tau}^{k-1}\|\bar{x}^{l+1} - \bar{x}^l\|^2 \quad \text{(since } d_{i_k}^k \leq \tau \text{ in Assumption 4.1)} \qquad (60)$$
$$= \tau\left[\sum_{l=k-\tau}^{k-1}[l-(k-\tau)+1]\|\bar{x}^{l+1} - \bar{x}^l\|^2 - \sum_{l=k-\tau+1}^{k}(l-(k-\tau))\|\bar{x}^{l+1} - \bar{x}^l\|^2\right]$$
$$+ \tau^2\|\bar{x}^{k+1} - \bar{x}^k\|^2.$$

Now, we consider two cases as follows.

**Case 1:** If $n \geq 2\tau^2$, then by plugging (60) into (59), we finally arrive at

$$V_\eta^{k+1}(\bar{x}^{k+1}) + \frac{\tau}{n\eta}\sum_{l=k-\tau+1}^{k}[l-(k-\tau)]\|\bar{x}^{l+1} - \bar{x}^l\|^2 \leq V_\eta^k(\bar{x}^k)$$
$$+ \frac{\tau}{n\eta}\sum_{l=k-\tau}^{k-1}[l-(k-\tau)+1]\|\bar{x}^{l+1} - \bar{x}^l\|^2$$
$$- \frac{[2(1-\alpha)-(2+\alpha)\eta^2L^2-\alpha\eta L]}{2\alpha\eta n}\|x_{i_k}^{k+1} - x_{i_k}^k\|^2 - \frac{(n-2\tau^2)}{2n\eta}\|\bar{x}^{k+1} - \bar{x}^k\|^2.$$

Rearranging the last estimate, we finally arrive at (53).

**Case 2:** if $2\tau^2 > n$, then using (21), we can show that

$$
\begin{aligned}
\|\bar{x}^{k+1} - \bar{x}^k\|^2 = \left\|\text{prox}_{\eta g}\big(\tilde{x}^{k+1}\big) - \text{prox}_{\eta g}\big(\tilde{x}^k\big)\right\|^2 &\leq \|\tilde{x}^{k+1} - \tilde{x}^k\|^2 \\
&= \|\tfrac{1}{n}\sum_{i=1}^n (\hat{x}_i^{k+1} - \hat{x}_i^k)\|^2 \\
&= \tfrac{1}{n^2}\|\hat{x}_{i_k}^{k+1} - \hat{x}_{i_k}^k\|^2 \quad \text{(since only block } i_k \text{ is updated)} \\
&\overset{(21)}{=} \tfrac{1}{n^2}\|(x_{i_k}^{k+1} - x_{i_k}^k) - \eta(\nabla f_{i_k}(x_{i_k}^{k+1}) - \nabla f_{i_k}(x_{i_k}^k))\|^2 \\
&\leq \tfrac{2}{n^2}\|x_{i_k}^{k+1} - x_{i_k}^k\|^2 + \tfrac{2\eta^2}{n^2}\|\nabla f_{i_k}(x_{i_k}^{k+1}) - \nabla f_{i_k}(x_{i_k}^k)\|^2 \\
&\leq \tfrac{2(1+\eta^2 L^2)}{n^2}\|x_{i_k}^{k+1} - x_{i_k}^k\|^2.
\end{aligned}
\tag{61}
$$

Substituting this inequality into the previous one, we can get

$$
\begin{aligned}
V_\eta^{k+1}(\bar{x}^{k+1}) + \tfrac{\tau}{n\eta}\sum_{l=k-\tau+1}^{k}[l-(k-\tau)]\|\bar{x}^{l+1} - \bar{x}^l\|^2 &\leq V_\eta^k(\bar{x}^k) \\
&\quad + \tfrac{\tau}{n\eta}\sum_{l=k-\tau}^{k-1}[l-(k-\tau)+1]\|\bar{x}^{l+1} - \bar{x}^l\|^2 \\
&\quad - \left[\tfrac{2(1-\alpha)-(2+\alpha)\eta^2 L^2 - \alpha\eta L}{2\alpha\eta n} - \tfrac{(1+\eta^2 L^2)(2\tau^2 - n)}{2n^3\eta}\right]\|x_{i_k}^{k+1} - x_{i_k}^k\|^2.
\end{aligned}
$$

Simplifying the coefficients of this estimate, we finally arrive at (53). $\qquad\square$

To analyze Algorithm 2, we need the following key lemma.

**Lemma B.2** (Sure descent lemma). *Suppose that Assumptions 2.1, 2.2, and 4.1 hold. Let $\left\{(x_i^k, y_i^k, \hat{x}_i^k, \tilde{x}^k, \bar{x}^k)\right\}$ be generated by Algorithm 2 and $V_\eta^k$ be defined as in (25). Let*

$$
\widetilde{V}_\eta^k(\bar{x}^k) := V_\eta^k(\bar{x}^k) + \tfrac{1}{n\eta}\sum_{l=k-\tau}^{k-1}[l-(k-\tau)+1]\|\bar{x}^{l+1} - \bar{x}^l\|^2.
\tag{62}
$$

*Suppose that we choose $0 < \alpha < \bar{\alpha}$ and $0 < \eta < \bar{\eta}$, where $c := \frac{2\tau^2 - n}{n^2}$,*

$$
\bar{\alpha} := \begin{cases} 1 & \text{if } 2\tau^2 \leq n, \\ \tfrac{2}{2+c} & \text{otherwise,} \end{cases}
$$

$$
\text{and} \quad \bar{\eta} := \begin{cases} \tfrac{\sqrt{16 - 8\alpha - 7\alpha^2} - \alpha}{2L(2+\alpha)} & \text{if } 2\tau^2 \leq n, \\ \tfrac{\sqrt{16 - 8\alpha - (7+4c+4c^2)\alpha^2} - \alpha}{2L[2+(1+c)\alpha]} & \text{otherwise.} \end{cases}
\tag{63}
$$

*Then, the following statement holds:*

$$
\tfrac{\rho}{2}\|x_{i_k}^{k+1} - x_{i_k}^k\|^2 \leq \widetilde{V}_\eta^k(\bar{x}^k) - \widetilde{V}_\eta^{k+1}(\bar{x}^{k+1}),
\tag{64}
$$

*where*

$$
\rho := \begin{cases} \tfrac{2(1-\alpha)-(2+\alpha)L^2\eta^2 - L\alpha\eta}{\alpha\eta n} & \text{if} \quad 2\tau^2 \leq n, \\ \tfrac{n^2[2(1-\alpha)-(2+\alpha)L^2\eta^2 - L\alpha\eta] - \alpha(1+\eta^2 L^2)(2\tau^2 - n)}{\alpha\eta n^3} & \text{otherwise.} \end{cases}
$$

*Moreover, $\rho$ is positive.*

*Proof.* If we define $\widetilde{V}_\eta^k$ as in (62) of Lemma B.2, i.e.:

$$
\widetilde{V}_\eta^k(\bar{x}^k) := V_\eta^k(\bar{x}^k) + \tfrac{\tau}{\eta n^2}\sum_{l=k-\tau}^{k-1}[l-(k-\tau)+1]\|\bar{x}^{l+1} - \bar{x}^l\|^2,
$$

then from (53), we have

$$
\widetilde{V}_\eta^{k+1}(\bar{x}^{k+1}) \leq \widetilde{V}^k(\bar{x}^k) - \tfrac{\rho}{2}\|x_{i_k}^{k+1} - x_{i_k}^k\|^2,
$$

which is equivalent to (64).

Now, we find conditions of $\alpha$ and $\eta$ such that $\rho$ and $\theta$ are positive. We consider two cases as follows.

**Case 1:** If $2\tau^2 \leq n$, then

$$\rho := \frac{2(1-\alpha)-(2+\alpha)L^2\eta^2-L\alpha\eta}{\alpha\eta n}.$$

Let us choose $0 < \alpha < 1$. To guarantee $\rho > 0$, we require $2(1-\alpha) > (2+\alpha)L^2\eta^2 + L\alpha\eta$. In this case, we need to choose $0 < \eta < \frac{\sqrt{L^2\alpha^2+8(1-\alpha)(2+\alpha)L^2}-L\alpha}{2L^2(2+\alpha)} = \frac{\sqrt{16-8\alpha-7\alpha^2}-\alpha}{2L(2+\alpha)}$. These are the choices in (63) when $2\tau^2 \leq n$.

**Case 2:** If $2\tau^2 > n$, then

$$\rho := \frac{n^2[2(1-\alpha)-(2+\alpha)L^2\eta^2-L\alpha\eta]-\alpha(1+\eta^2L^2)(2\tau^2-n)}{\alpha\eta n^3}.$$

Let $c := \frac{2\tau^2-n}{n^2} > 0$. In order to guarantee that $\rho > 0$, we need to choose $0 < \alpha < 1$ and $\eta > 0$ such that

$$2 - 2\alpha - \frac{\alpha(2\tau^2-n)}{n^2} > \left[2 + \alpha + \frac{\alpha(2\tau^2-n)}{n^2}\right]L^2\eta^2 + L\alpha\eta,$$

$$\text{and} \quad 0 < \alpha < \frac{2n^2}{2n^2+(2\tau^2-n)} = \frac{2}{2+c}.$$

Using the definition of $c$, the first condition becomes $2 - 2\alpha - c\alpha > L\alpha\eta + (2+\alpha+c\alpha)L^2\eta^2$. First, we need to impose $2 - 2\alpha - c\alpha > 0$, leading to $0 < \alpha < \frac{2}{2+c}$. Next, we solve the above inequality w.r.t. $\eta > 0$ to get

$$0 < \eta < \bar{\eta} := \frac{\sqrt{16-8\alpha-(7+4c+4c^2)\alpha^2}-\alpha}{2L[2+(1+c)\alpha]}.$$

These are the choices in (63) when $2\tau^2 > n$. To guarantee $\bar{\eta} > 0$, we need to choose $\alpha < \frac{4}{1+\sqrt{1+4(2+c+c^2)}}$. Combining four conditions of $\alpha$, we get $0 < \alpha < \frac{2}{2+c}$. Finally, we conclude that under the choice of $\alpha$ and $\eta$ as in (63), we have $\rho > 0$ and $\theta > 0$. $\qquad\square$

Next lemma bounds the term $\sum_{i=1}^{n} \mathbb{E}\left[\|\bar{x}^k - x_i^k\|^2\right]$ in order to bound $\mathbb{E}\left[\|\mathcal{G}_\eta(\bar{x}^k)\|^2\right]$.

**Lemma B.3.** *Suppose that Assumptions 2.1, 2.2, and 4.1 hold. Let $\left\{(x_i^k, y_i^k, \hat{x}_i^k, \bar{x}^k)\right\}$ be generated by Algorithm 2. Then, we have*

$$\sum_{i=1}^{n} \mathbb{E}\left[\|\bar{x}^k - x_i^k\|^2\right] \leq D \sum_{t=k-\tau}^{k+T} \mathbb{E}\left[\|x_{i_t}^{t+1} - x_{i_t}^t\|^2\right], \tag{65}$$

*where $D := \frac{8\alpha^2(1+L^2\eta^2)(\tau^2+2Tn\hat{\mathbf{p}}) + 8n^2(1+L^2\eta^2+T\alpha^2\hat{\mathbf{p}})}{\hat{\mathbf{p}}\alpha^2 n^2}$.*

*Proof.* Let $t_k(i) := \min\left\{t \in \{0, \cdots, T\} : \mathbf{p}(i \mid \xi^{0:k+t-1}) \geq \hat{\mathbf{p}}\right\}$. In fact, $t_k(i)$ is the first time in the iteration window $[k, k+T]$, user $i$ is active, i.e. gets updated. For any $\gamma \in (0,1)$, we have

$$\sum_{t=k}^{k+T} \mathbb{E}\left[\|\bar{x}^t - x_{\hat{i}_t}^t\|^2 \mid \mathcal{F}_{t-1}\right](\omega) = \sum_{t=k}^{k+T} \sum_{i=1}^{n} \mathbf{p}(i \mid \xi^{0:t-1})\|\bar{x}^t - x_i^t\|^2$$

$$\overset{(51)}{\geq} \sum_{i=1}^{n} \hat{\mathbf{p}}\|\bar{x}^{k+t_k(i)} - x_i^{k+t_k(i)}\|^2$$

$$\overset{(*)}{\geq} \hat{\mathbf{p}} \sum_{i=1}^{n} \left[\|\bar{x}^k - x_i^k\| - \|\bar{x}^{k+t_k(i)} - x_i^{k+t_k(i)} - (\bar{x}^k - x_i^k)\|\right]^2$$

$$\geq -2\hat{\mathbf{p}} \sum_{i=1}^{n} \|\bar{x}^k - x_i^k\|\|\bar{x}^{k+t_k(i)} - x_i^{k+t_k(i)} - (\bar{x}^k - x_i^k)\|$$

$$\quad + \hat{\mathbf{p}} \sum_{i=1}^{n} \|\bar{x}^k - x_i^k\|^2$$

$$\overset{(**)}{\geq} \hat{\mathbf{p}} \sum_{i=1}^{n} \left[\|\bar{x}^k - x_i^k\|^2 - \frac{1}{2}\|\bar{x}^k - x_i^k\|^2\right]$$

$$\quad - 4\hat{\mathbf{p}} \sum_{i=1}^{n} \|\bar{x}^{k+t_k(i)} - \bar{x}^k\|^2 - 4\hat{\mathbf{p}} \sum_{i=1}^{n} \|x_i^{k+t_k(i)} - x_i^k\|^2,$$

where (*) comes from the reverse triangle inequality $\|a - b\|^2 \geq (\|a\| - \|b\|)^2$ and (**) is due to $4\|v\|^2 + 4\|s\|^2 + \frac{1}{2}\|u\|^2 \geq 2\|u\|\|v + s\|$. Note that the conditional expectation above is only taken w.r.t. $\hat{i}_k$, which is $\sigma(d^k, \mathcal{F}_{k-1})$-measurable. For simplicity of notation, we drop $(\omega)$ in the sequel.

Rearranging the last inequality, we obtain

$$\frac{\hat{\mathbf{p}}}{2}\sum_{i=1}^{n}\|\bar{x}^k - x_i^k\|^2 \leq \sum_{t=k}^{k+T}\mathbb{E}\left[\|\bar{x}^t - x_{\hat{i}_t}^t\|^2 \mid \mathcal{F}_{t-1}\right] + 4\hat{\mathbf{p}}\sum_{i=1}^{n}\|\bar{x}^{k+t_k(i)} - \bar{x}^k\|^2 \tag{66}$$
$$+ 4\hat{\mathbf{p}}\sum_{i=1}^{n}\|x_i^{k+t_k(i)} - x_i^k\|^2.$$

Next, we bound the term $\sum_{i=1}^{n}\|\bar{x}^{k+t_k(i)} - \bar{x}^k\|^2$ as follows:

$$
\begin{aligned}
\sum_{i=1}^{n}\|\bar{x}^{k+t_k(i)} - \bar{x}^k\|^2 &= \sum_{i=1}^{n}\|\sum_{t=k}^{k+t_k(i)-1}(\bar{x}^{t+1} - \bar{x}^t)\|^2 \\
&\leq \sum_{i=1}^{n}t_k(i)\sum_{t=k}^{k+t_k(i)-1}\|\bar{x}^{t+1} - \bar{x}^t\|^2 \qquad \text{(Young's inequality)} \\
&\leq T\sum_{i=1}^{n}\sum_{t=k}^{k+t_k(i)-1}\|\bar{x}^{t+1} - \bar{x}^t\|^2 \qquad \text{(since } t_k(i) \leq T) \\
&= nT\sum_{t=k}^{k+T}\|\bar{x}^{t+1} - \bar{x}^t\|^2 \\
&\overset{(61)}{\leq} \frac{2T(1+\eta^2 L^2)}{n}\sum_{t=k}^{k+T}\|x_{i_t}^{t+1} - x_{i_t}^t\|^2.
\end{aligned}
\tag{67}
$$

We can also bound $\sum_{i=1}^{n}\|x_i^{k+t_k(i)} - x_i^k\|^2$ as follows:

$$
\begin{aligned}
\sum_{i=1}^{n}\|x_i^{k+t_k(i)} - x_i^k\|^2 &= \sum_{i=1}^{n}\|\sum_{t=k}^{k+t_k(i)-1}(x_i^{t+1} - x_i^t)\|^2 \\
&\leq \sum_{i=1}^{n}t_k(i)\sum_{t=k}^{k+t_k(i)-1}\|x_i^{t+1} - x_i^t\|^2 \qquad \text{(Young's inequality)} \\
&\leq T\sum_{t=k}^{k+T-1}\sum_{i=1}^{n}\|x_i^{t+1} - x_i^t\|^2 \qquad \text{(since } t_k(i) \leq T) \\
&= T\sum_{t=k}^{k+T}\|x_{i_t}^{t+1} - x_{i_t}^t\|^2 \quad \text{(since only user } i_t \text{ is updated at iteration } t).
\end{aligned}
\tag{68}
$$

Let us bound the first term on the right-hand side of (66) as follows:

$$
\begin{aligned}
\sum_{t=k}^{k+T}\mathbb{E}\left[\|\bar{x}^t - x_{\hat{i}_t}^t\|^2 \mid \mathcal{F}_{t-1}\right] &\leq 2\sum_{t=k}^{k+T}\mathbb{E}\left[\|\bar{x}^{t-d_{i_t}^t} - x_{\hat{i}_t}^t\|^2 \mid \mathcal{F}_{t-1}\right] \\
&\quad + 2\sum_{t=k}^{k+T}\mathbb{E}\left[\|\bar{x}^t - \bar{x}^{t-d_{i_t}^t}\|^2 \mid \mathcal{F}_{t-1}\right].
\end{aligned}
\tag{69}
$$

However, similar to the proof of (60) and (61), we can show that

$$
\begin{aligned}
\sum_{t=k}^{k+T}\|\bar{x}^t - \bar{x}^{t-d_{i_t}^t}\|^2 &\overset{(60)}{\leq} \tau\sum_{t=k}^{k+T}\sum_{l=t-\tau}^{t-1}\|\bar{x}^{l+1} - \bar{x}^l\|^2 \\
&\leq \tau^2\sum_{t=k-\tau}^{k+T}\|\bar{x}^{t+1} - \bar{x}^t\|^2 \\
&\overset{(61)}{\leq} \sum_{t=k-\tau}^{k+T}\frac{2\tau^2(1+\eta^2 L^2)}{n^2}\|x_{i_t}^{t+1} - x_{i_t}^t\|^2.
\end{aligned}
\tag{70}
$$

On the other hand, by using (56), we have

$$
\begin{aligned}
\|\bar{x}^{t-d_{i_t}^t} - x_{\hat{i}_t}^t\|^2 &= \|y_{\hat{i}_t}^{t+1} - y_{\hat{i}_t}^t\|^2 \qquad \text{(by the update of } y_{\hat{i}_k}^k \text{ in Algorithm 2)} \\
&\overset{(56)}{=} \|\frac{1}{\alpha}(x_{\hat{i}_t}^{t+1} - x_{\hat{i}_t}^t) + \frac{\eta}{\alpha}(\nabla f_{\hat{i}_t}(x_{\hat{i}_t}^{t+1}) - \nabla f_{\hat{i}_t}(x_{\hat{i}_t}^t))\|^2 \\
&\leq \frac{2}{\alpha^2}\|x_{\hat{i}_t}^{t+1} - x_{\hat{i}_t}^t\|^2 + \frac{2\eta^2}{\alpha^2}\|\nabla f_{\hat{i}_t}(x_{\hat{i}_t}^{t+1}) - \nabla f_{\hat{i}_t}(x_{\hat{i}_t}^t)\|^2 \\
&\leq \frac{2(1+\eta^2 L^2)}{\alpha^2}\|x_{\hat{i}_t}^{t+1} - x_{\hat{i}_t}^t\|^2.
\end{aligned}
\tag{71}
$$

Therefore, plugging (70) and (71) into (69), we have

$$
\begin{aligned}
\sum_{t=k}^{k+T}\mathbb{E}\left[\|\bar{x}^t - x_{\hat{i}_t}^t\|^2 \mid \mathcal{F}_{t-1}\right] &\leq \frac{4\tau^2(1+\eta^2 L^2)}{n^2}\sum_{t=k-\tau}^{k+T}\mathbb{E}\left[\|x_{\hat{i}_t}^{t+1} - x_{\hat{i}_t}^t\|^2 \mid \mathcal{F}_{t-1}\right] \\
&\quad + \frac{4(1+\eta^2 L^2)}{\alpha^2}\sum_{t=k-\tau}^{k+T}\mathbb{E}\left[\|x_{\hat{i}_t}^{t+1} - x_{\hat{i}_t}^t\|^2 \mid \mathcal{F}_{t-1}\right] \\
&= \frac{4(1+\eta^2 L^2)[\tau^2\alpha^2 + n^2]}{n^2\alpha^2}\sum_{t=k-\tau}^{k+T}\mathbb{E}\left[\|x_{\hat{i}_t}^{t+1} - x_{\hat{i}_t}^t\|^2 \mid \mathcal{F}_{t-1}\right].
\end{aligned}
\tag{72}
$$

Substituting (67), (68), and (72) into (66), we obtain

$$
\begin{aligned}
\frac{\hat{\mathbf{p}}}{2}\sum_{i=1}^{n}\|\bar{x}^k - x_i^k\|^2 &\leq \frac{4(1+\eta^2 L^2)[\tau^2\alpha^2 + n^2]}{n^2\alpha^2}\sum_{t=k-\tau}^{k+T}\mathbb{E}\left[\|x_{\hat{i}_t}^{t+1} - x_{\hat{i}_t}^t\|^2 \mid \mathcal{F}_{t-1}\right] \\
&\quad + \frac{8\hat{\mathbf{p}}T(1+\eta^2 L^2)}{n}\sum_{t=k}^{k+T}\|x_{i_t}^{t+1} - x_{i_t}^t\|^2 + 4\hat{\mathbf{p}}T\sum_{t=k}^{k+T}\|x_{i_t}^{t+1} - x_{i_t}^t\|^2 \\
&\leq \frac{4(1+\eta^2 L^2)[\tau^2\alpha^2 + n^2]}{n^2\alpha^2}\sum_{t=k-\tau}^{k+T}\mathbb{E}\left[\|x_{\hat{i}_t}^{t+1} - x_{\hat{i}_t}^t\|^2 \mid \mathcal{F}_{t-1}\right] \\
&\quad + \frac{4\hat{\mathbf{p}}T[2(1+\eta^2 L^2)+n]}{n}\sum_{t=k-\tau}^{k+T}\|x_{i_t}^{t+1} - x_{i_t}^t\|^2.
\end{aligned}
$$

Finally, taking full expectation both sides of the last inequality w.r.t. $\sigma(d^k, \mathcal{F}_{k-1})$, and multiplying the result by $\frac{2}{\hat{\mathbf{p}}}$, we arrive at

$$\sum_{i=1}^{n} \mathbb{E}\left[\|\bar{x}^k - x_i^k\|^2\right] \leq D \sum_{t=k-\tau}^{k+T} \mathbb{E}\left[\|x_{i_t}^{t+1} - x_{i_t}^t\|^2\right],$$

where $D := \frac{8(1+\eta^2 L^2)(\tau^2\alpha^2+n^2)}{\hat{\mathbf{p}}n^2\alpha^2} + \frac{8T[2(1+\eta^2 L^2)+n]}{n}$. This inequality is exactly (65). $\square$

## B.4 The proof of Theorem 4.1: Convergence of Algorithm 2

By Assumption 4.1, for each $T$ iterations, the probability of each user $i$ getting updated is at least $\hat{\mathbf{p}} > 0$. Hence, from (64) of Lemma B.2, we sum up from $t := k - \tau$ to $t := k + T$, and have

$$\frac{\rho}{2} \sum_{t=k-\tau}^{k+T} \|x_{i_t}^{t+1} - x_{i_t}^t\|^2 \leq \sum_{t=k-\tau}^{k+T} \left[\widetilde{V}_\eta^t(\bar{x}^t) - \widetilde{V}_\eta^{t+1}(\bar{x}^{t+1})\right],$$

where $\rho > 0$ is given in Lemma B.2. Now, take full expectation both sides of this inequality w.r.t. $\mathcal{F}_k$, we obtain

$$\frac{\rho}{2} \sum_{t=k-\tau}^{k+T} \mathbb{E}\left[\|x_{i_t}^{t+1} - x_{i_t}^t\|^2\right] \leq \sum_{t=k-\tau}^{k+T} \left[\mathbb{E}\left[\widetilde{V}_\eta^t(\bar{x}^t)\right] - \mathbb{E}\left[\widetilde{V}_\eta^{t+1}(\bar{x}^{t+1})\right]\right]. \tag{73}$$

Next, using (23) from Lemma A.3 with $\gamma_2 = 0$, we have

$$\|\mathcal{G}_\eta(\bar{x}^k)\|^2 \leq \frac{(1+\eta L)^2}{n\eta^2} \sum_{i=1}^{n} \|x_i^k - \bar{x}^k\|^2.$$

Taking full expectation both sides of this inequality, and then combining the result and (65), we obtain

$$\mathbb{E}\left[\|\mathcal{G}_\eta(\bar{x}^k)\|^2\right] \leq \frac{(1+\eta L)^2 D}{n\eta^2} \sum_{t=k-\tau}^{k+T} \mathbb{E}\left[\|x_{i_t}^{t+1} - x_{i_t}^t\|^2\right],$$

where $D$ is given in Lemma B.3.

Combining the last inequality and (73), we arrive at

$$\mathbb{E}\left[\|\mathcal{G}_\eta(\bar{x}^k)\|^2\right] \leq \frac{2(1+\eta L)^2 D}{n\eta^2\rho} \sum_{t=k-\tau}^{k+T} \left(\mathbb{E}\left[\widetilde{V}_\eta^t(\bar{x}^t)\right] - \mathbb{E}\left[\widetilde{V}_\eta^{t+1}(\bar{x}^{t+1})\right]\right).$$

Averaging this inequality from $k := 0$ to $k := K$, we get

$$\begin{aligned}
\frac{1}{K+1} \sum_{k=0}^{K} \mathbb{E}\left[\|\mathcal{G}_\eta(\bar{x}^k)\|^2\right] &\leq \frac{\hat{C}}{K+1} \sum_{k=0}^{K} \sum_{t=k-\tau}^{k+T} \left[\mathbb{E}\left[\widetilde{V}_\eta^t(\bar{x}^t)\right] - \mathbb{E}\left[\widetilde{V}_\eta^{t+1}(\bar{x}^{t+1})\right]\right] \\
&\leq \frac{\hat{C}}{K+1} \left[\widetilde{V}_\eta^0(\bar{x}^0) - \mathbb{E}\left[\widetilde{V}_\eta^{K+T+1}(\bar{x}^{K+T+1})\right]\right],
\end{aligned} \tag{74}$$

where $\hat{C} := \frac{2(1+\eta L)^2 D}{n\rho\eta^2}$. Here, we have used the monotonicity of $\{\mathbb{E}\left[\widetilde{V}_\eta^k(\bar{x}^k)\right]\}_{k\geq 0}$ and $\mathbb{E}\left[\widetilde{V}_\eta^0(\bar{x}^0)\right] = \widetilde{V}_\eta^0(\bar{x}^0)$ in the last equality.

Now, recall from the definition of $\widetilde{V}_\eta^k(\cdot)$ and $V_\eta^k(\cdot)$ that

$$\widetilde{V}_\eta^0(\bar{x}^0) = V_\eta^0(\bar{x}^0) = F(x^0) \qquad \text{and} \qquad \mathbb{E}\left[\widetilde{V}_\eta^k(\bar{x}^k)\right] \geq \mathbb{E}\left[V_\eta^k(\bar{x}^k)\right] \overset{(36)}{\geq} F^\star.$$

Substituting these relations into (74), we eventually get

$$\frac{1}{K+1} \sum_{k=0}^{K} \mathbb{E}\left[\|\mathcal{G}_\eta(\bar{x}^k)\|^2\right] \leq \frac{\hat{C}}{(K+1)} \left[F(x^0) - F^\star\right],$$

which is exactly (10). Using the definition of $\rho$, $\theta$, and $D$ into $\hat{C}$, we obtain its simplified formula as in Theorem 4.1. The remaining conclusion of the theorem is a direct consequence of (10). $\square$

# C   Implementation Details and Additional Numerical Examples

In this section, we provide more details on the set up of numerical experiments and present additional numerical results to illustrate the performance of our algorithms compared to others.

## C.1   Details on numerical experiments

**Parameter selection.**   We use the learning rate for local solver (SGD) as reported in [23] to approximately evaluate $\text{prox}_{\eta f_i}(y_i^k)$ at each user $i \in [n]$. The learning rates are 0.01 for all synthetic datasets, 0.01 for MNIST, and 0.003 for FEMNIST. We also perform a grid-search over multiple values to select the parameter and stepsizes for FedProx, FedPD and FedDR. In particular, we choose $\mu \in [0.001, 1]$ for FedProx, $\eta \in [1, 1000]$ for FedPD, and $\eta \in [1, 1000]$, $\alpha \in [0, 1.99]$ for FedDR. All algorithms perform local SGD updates with 20 epochs to approximately evaluate $\text{prox}_{\eta f_i}(y_i^k)$ before sending the results to server for [proximal] aggregation.

**Training models.**   For all datasets, we use fully-connected neural network as training models. For all synthetic datasets, we use a neural network of size $60 \times 32 \times 10$ where we use the format input size $\times$ hiddden layer $\times$ output size. For MNIST, we use a network of size $784 \times 128 \times 10$. For FEMNIST used in the main text, we reuse the dataset from [23] and a $784 \times 128 \times 26$ model.

**Composite examples.**   We test our algorithm under composite setting where we set $g(x) = 0.01 \|x\|_1$. In the first test, we choose $\eta = 500$, $\alpha = 1.95$ and select the local learning rate (*lr*) for SGD to approximately evaluate $\text{prox}_{\eta f_i}(y_i^k)$ from the set $\{0.0025, 0.005, 0.0075, 0.01, 0.025\}$ for `synthetic-(0,0)` and $\{0.001, 0.003, 0.005, 0.008, 0.01\}$ for `FEMNIST`. Next, we fix the local learning rate at 0.01 for `synthetic-(0,0)` and 0.003 for `FEMNIST` then adjust the number of local epochs in the set $\{5, 10, 15, 20, 30\}$ to evaluate $\text{prox}_{\eta f_i}(y_i^k)$. Finally, we test our algorithm when changing the total number of users participating at each communication round $|\mathcal{S}_k|$. For `synthetic-(0,0)` dataset, we set $|\mathcal{S}_k| \in \{5, 10, 15, 20, 25\}$. For `FEMNIST` dataset, we set $|\mathcal{S}_k| \in \{10, 25, 50, 75, 100\}$.

**Asynchronous example.**   To make the sample size larger for each user, we generate the FEMNIST dataset using Leaf [4]. In the new dataset, there are actually 62 classes instead of 26 classes as used in [23]. Therefore, we denote this dataset as **FEMNIST - 62 classes**. In this new dataset, each user has sample size ranging from 97 to 356. We implement the communication between server and user using the distributed package in `Pytorch` [1] as in [3]. There are 21 threads created, one acts as server and 20 others are users. To simulate the case when users have different computing power, we add a certain amount of delay at the end of each user's local update such that the total update time varies between all users. For **FEMNIST - 62 classes** dataset, the model is a fully-connected neural network of the size $784 \times 128 \times 62$.

## C.2   Additional numerical results

We first present two experiments on iid and non-iid datasets without using user sampling scheme as shown in Figure 7. That is all users participate into the system at each communication round.

From Figure 7, FedAvg appears to perform best while the other three algorithms are comparable in the iid setting. Similar behavior is also observed in [23]. For the non-iid datasets along with Figure 2, we observe that the more non-iid the dataset is, the more unstable these algorithms behave. In the `synthetic-(1,1)` dataset, FedDR appears to be the best followed by FedPD. FedProx also performs much better than FedAvg in this test.

Figure 8 depicts the performance of 4 algorithms in terms of communication cost on the `synthetic-(1,1)` dataset. We still observe that FedDR works well while FedProx and FedPD are comparable but still better than FedAvg.

More results of experiments on the composite setting are presented in Figure 9. We observe that the learning rate (*lr*) of SGD needs to be tuned for each dataset and the local iteration should be selected carefully to trade-off between local computation cost and inexactness of the evaluation of $\text{prox}_{\eta f_i}(y_i^k)$.

---

[1] See `https://pytorch.org/tutorials/beginner/dist_overview.html` for more details.

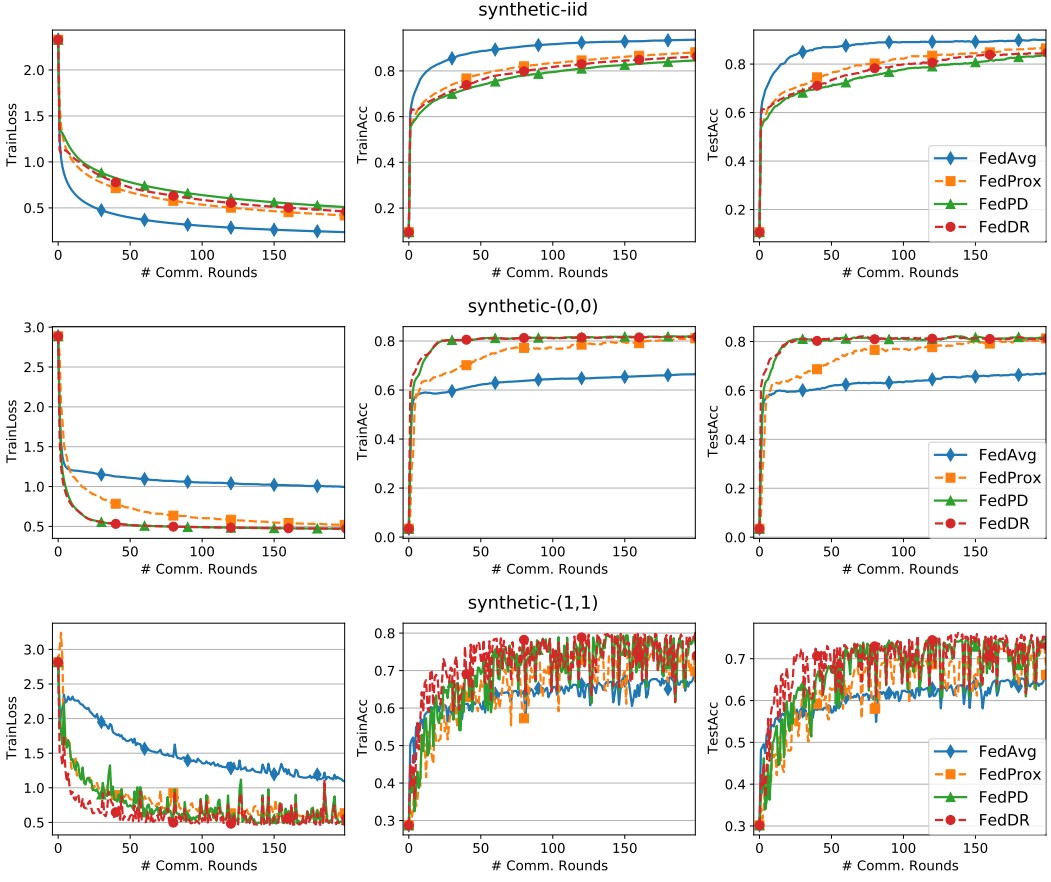

Figure 7: The performance of 4 algorithms on iid and non-iid synthetic datasets without user sampling scheme. The first row is for one iid dataset, and the last two rows are for non-iid datasets.

We also vary the number of users sampled at each communication round. The results are depicted in Figure 10 for two datasets. We observe that the performance when we sample smaller number of user per round is not as good as larger ones in terms of communication rounds. However, this might not be a fair comparison since fewer clients also require less communication cost. Therefore, we plot these results in terms of number of bytes communicated. The results are depicted in Figure 11. From Figure 11, FedDR performs very similarly under different choices of $\mathcal{S}_k$.

We also compare FedDR and asyncFedDR using the FEMNIST dataset. The results are depicted in Figure 12. We can see that asyncFedDR is advantageous over FedDR to achieve lower loss value and higher accuracies.

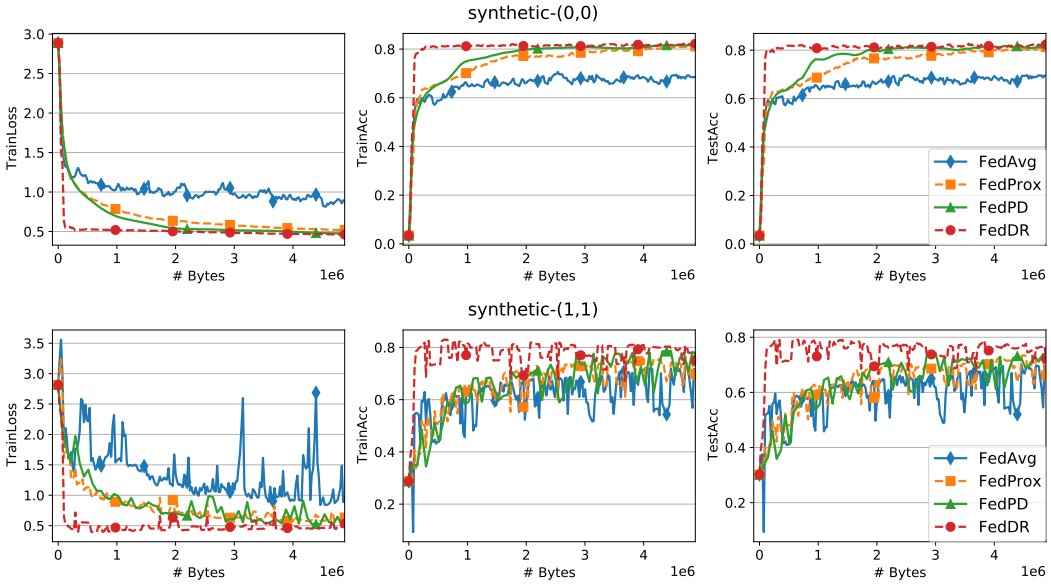

Figure 8: The performance of 4 algorithms without user sampling scheme on non-iid datasets in terms of communication effort.

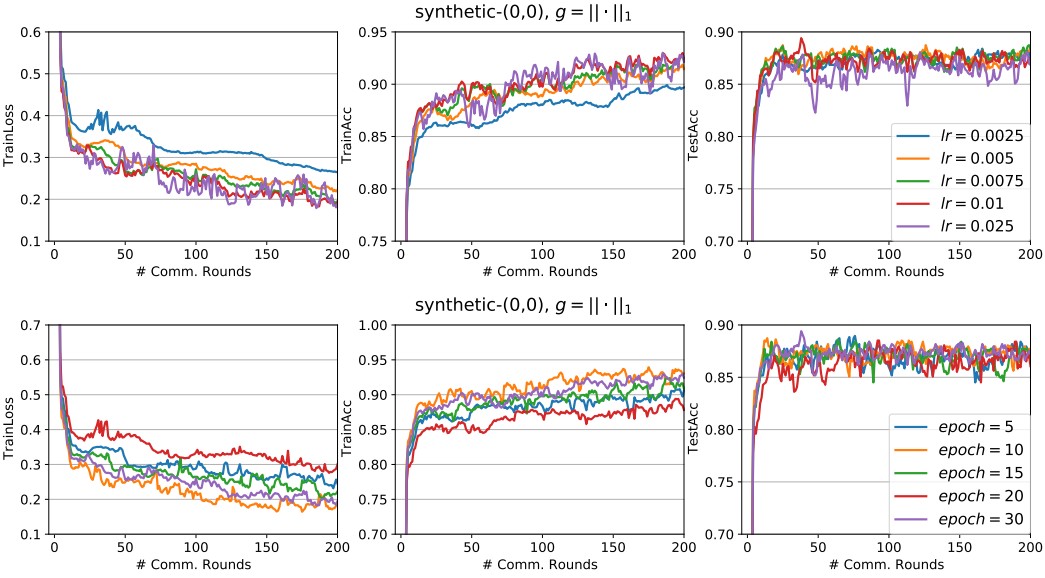

Figure 9: The performance of **FedDR** on synthetic dataset in composite setting.

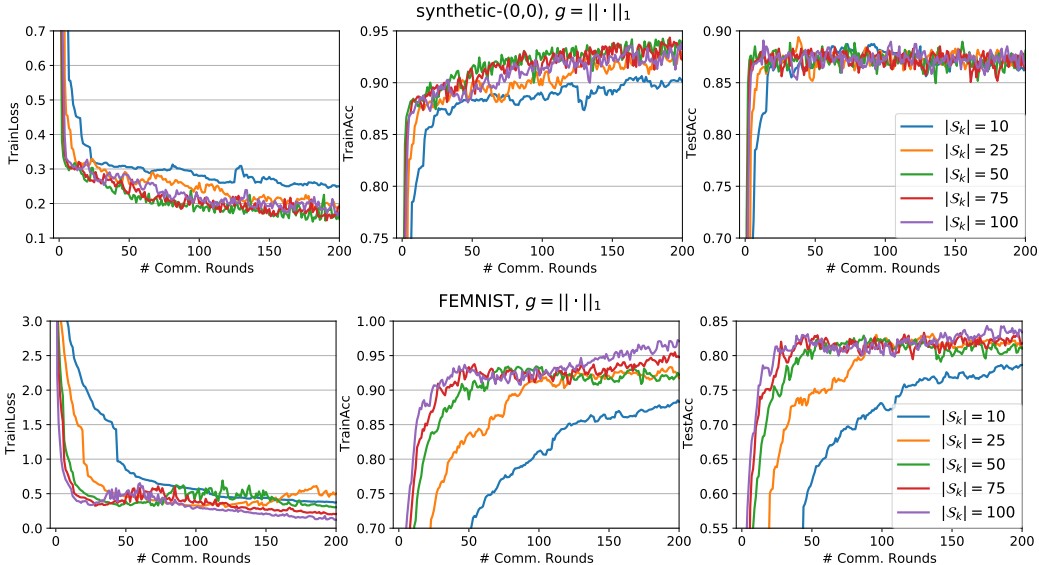

Figure 10: The performance of **FedDR** in composite setting in terms of communication rounds.

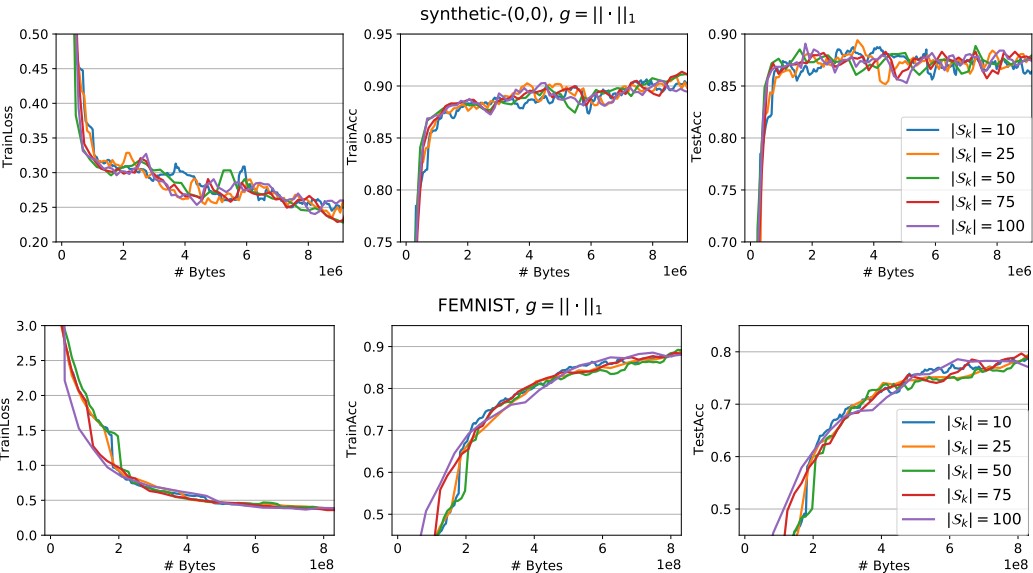

Figure 11: The performance of **FedDR** in composite setting in terms of number of bytes.

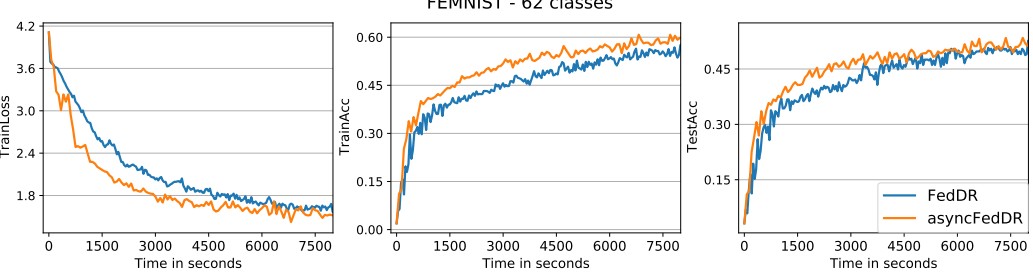

Figure 12: The performance of FedDR and asyncFedDR on **FEMNIST - 62 classes** dataset.