# OpenReview forum: "FedDR – Randomized Douglas-Rachford Splitting Algorithms for Nonconvex Federated Composite Optimization"
_NeurIPS.cc/2021/Conference — NeurIPS 2021 Poster_

### Official Review · Reviewer_TPm9 · 2021-07-13

**Rating:** 6
**Confidence:** 3

**Summary:**

This paper proposes FedDR, which extends FedSplit by allowing non-convex objective, partial participation and convex composite regularizers. This work also studies the asynchronous variant of FedDR. Detailed proofs and experiments are provided.

**Limitations And Societal Impact:**

See the main review section for limitations that needs improvement.

**Main Review:**

I would like to express my appreciation for your submission to NeurIPS 2021. This paper is overall well-written and I enjoyed reading it. This paper proposes FedDR which differs from existing works FedSplit, FedPD by allowing non-convex objective, partial participation and convex composite regularizers.  My main concerns are as follows:

1. The algorithm FedDR requires an inner solver to compute the proximal operator of f. Although the authors kindly provide an inexact variant that allows for inexact approximation of prox_f, the additional inner computation is still not factored into the overall complexity. Therefore, it is hard to calibrate the complexities of FedDR with other first-order canonical algorithms such as FedAvg. (I acknowledge the same complain could apply to other prior works along this line such as FedPD and FedSplit.)

2. This paper is mostly compared with the FedSplit and FedPD literature. However, as noted on line 128, the composite federated optimization has been explored by [45, Federated Composite Optimization] for the convex case. Though I agree with the author’s claim that FedDR is different from the algorithms in [45], this paper misses necessary comparisons with [45] either theoretically or empirically. For example, how does FedDR compare with [45] (theoretically and empirically) when the objective f is convex?

3. The theorem 3.1 is somewhat concerning as the constant C still depends on the hyperparameter $\alpha$ and $\eta$. Is it possible to state a cleaner bound with specific variables $\alpha, \eta$ so that the dependency can be eliminated?


**Time Spent Reviewing:**

3

---

> ### Author Response · Authors · 2021-08-08
> **Response to Reviewer TPm9's comments**
>
> Thank you so much for your constructive comments. Below is our detailed response to each comment.
>
> 1. Reviewer's summary:
> This paper proposes FedDR, which extends FedSplit by allowing non-convex objective, partial participation and convex composite regularizers. This work also studies the asynchronous variant of FedDR. Detailed proofs and experiments are provided.
>
> * Response:
> We are sorry for a small mixed up.
> Since Reviewer 1 has indicated that FedSplit is an instance of the Peaceman-Rachford splitting method, rather than the original Douglas-Rachford splitting scheme  (see also Section 3.2 of [32]), we now realize that our method is not an extension of FedSplit from the convex to the nonconvex case.
> In addition, since our algorithm relies on the Douglas-Rachford method with a relaxation parameter $\alpha$, and since $\alpha < 2$, we cannot recover the Peaceman-Rachford scheme, and therefore, FedSplit.
>
> 2. Reviewer's main comment 1:
> I would like to express my appreciation for your submission to NeurIPS 2021. This paper is overall well-written and I enjoyed reading it. This paper proposes FedDR which differs from existing works FedSplit, FedPD by allowing non-convex objective, partial participation and convex composite regularizers.
>
> * Response:
> Thank you very much for your appreciation of our work. We really appreciate your positive feedback.
>
> 3. Reviewer's main concern 1:
> The algorithm FedDR requires an inner solver to compute the proximal operator of f. Although the authors kindly provide an inexact variant that allows for inexact approximation of prox_f, the additional inner computation is still not factored into the overall complexity. Therefore, it is hard to calibrate the complexities of FedDR with other first-order canonical algorithms such as FedAvg. (I acknowledge the same complain could apply to other prior works along this line such as FedPD and FedSplit.)
>
> * Response:
> Thank you for indicating this point. Indeed, our initial submission did not focus on the inner loop since it requires solving a strongly convex subproblem on each user. Many methods can be applied to solve this subproblem depending on the form and structures of $f_i$. The original FedAvg suggests applying an SGD scheme to the inner loop. However, this may not be a good choice for all users, since some users may have closed-form solutions due to the simplicity of $f_i$, while others may have a more complex $f_i$. Hence, we leave this subsolver unspecified.
>
>     Our complexity presented in Theorems 3.1 and 3.2 is only communication complexity and it is independent of the choice of subsolver. However, based on your concern, we will specify one special case of our method in the next version when a specific subsolver is applied to evaluate each $\mathrm{prox}_{\eta f_i}$. In this case, we can estimate the overall complexity of the algorithm.
>
> 4. Reviewer's main concern 2:
> This paper is mostly compared with the FedSplit and FedPD literature. However, as noted on line 128, the composite federated optimization has been explored by [45, Federated Composite Optimization] for the convex case. Though I agree with the author’s claim that FedDR is different from the algorithms in [45], this paper misses necessary comparisons with [45] either theoretically or empirically. For example, how does FedDR compare with [45] (theoretically and empirically) when the objective f is convex?
>
> * Response:
> Thank you for your concern. Though we do not know how to theoretically compare our methods with [45] since [45] considers the convex case, and ours is on the nonconvex one, we will do some numerical comparison with [45] in the convex case as suggested.
>
> 5. Reviewer's main concern 3:
> The theorem 3.1 is somewhat concerning as the constant $C$ still depends on the hyperparameter $\alpha$ and $\eta$. Is it possible to state a cleaner bound with specific variables $\alpha, \eta$ so that the dependency can be eliminated?
>
> * Response:
> We will add a remark or corollary on a specific choice of $\alpha$ and $\eta$ as suggested. For example, we can choose $\alpha = 1$ to obtain the classical DR variant and then fix $\eta = \frac{1}{3L}$.

---

> > ### Comment · Reviewer_TPm9 · 2021-08-20
> > **Thanks for the response.**
> >
> > Thanks for the response. I have read the response and decided to keep my original score.

---

### Official Review · Reviewer_A2nV · 2021-07-16

**Rating:** 7
**Confidence:** 4

**Summary:**


This paper proposes FedDR and asyncFedDR for FL, which combines DRS and the randomized BCD strategy. Detailed convergence guarantees are also provided, which seems clear and correct to me. The authors also provide numerical results to support their theoretical claims.

**Limitations And Societal Impact:**

Yes

**Main Review:**


Major concerns:

1. My first concern is regarding the inexactness evaluations of \prox_{\eta f_i}.

   (1) First, in theorems 3.1 and 4.1, it would be much better if that fact that the error tolerances \eps_{i, k} should be square summable (which is stated in the proof in the appendix).

   (2) Secondly, in order to have square summable error tolerances, more and more inner loops (e.g., SGD or other solvers) are needed to solve these subproblems. This would make the total stochastic gradient complexity to be quite high. Note that what's presented in theorems 3.1 and 4.1 is the outer iteration complexity.

   (3) A possible remedy is to apply warm-start on each subproblem and only solve it for a fixed number of iterations. Then, there will be a "bounded relative error", which will not hurt the overall 1/K convergence rate.

This idea has been applied in the following analysis of inexact preconditioned PDHG (or equivalently, ADMM). More specifically, please refer to Theorem 4 of this paper:

Liu, Yanli, Yunbei Xu, and Wotao Yin. "Acceleration of Primal–Dual Methods by Preconditioning and Simple Subproblem Procedures." Journal of Scientific Computing 86.2 (2021): 1-34.

Since the analysis in the above paper also depends on a Lyapunov function, which looks quite similar to this work (Lemma A.5 and Lemma B.1). I think that this may help improve the overall stochastic gradient complexity of this work.

(4) In the experiments, it should also be mentioned how many inner loops of SGD are applied in FedDR, asyncFedDR, as well as other algorithms. And a comparison of stochastic gradient complexity should be provided, in addition to the communication complexity presented.

Minor concerns:
1. The analysis also looks similar to those of ARock ([33] in this paper), where the bounded delay is also assumed and a Lyapunov type analysis is done, would you please provide some comparisons?
2. On line 63, [nonsmooth] seems to be a typo.
3. On line 79, early should be earliest.

**Time Spent Reviewing:**

2

---

> ### Author Response · Authors · 2021-08-08
> **Response to Reviewer A2nV's comments**
>
> Thank you so much for your constructive comments. Below is our detailed response to each comment.
>
> 1. Reviewer's main concern 1:
> My first concern is regarding the inexactness evaluations of $\mathrm{prox}_{\eta{f_i}}$.
>
>      First, in theorems 3.1 and 4.1, it would be much better if that fact that the error tolerances $\epsilon_{i, k}$ should be square summable (which is stated in the proof in the appendix).
>
> * Response:
> Thank you for your suggestion. We will bring the result for the inexact case to the main paper and state the condition for  tolerances as suggested.
>
> 2. Reviewer's main concern 2:
> Secondly, in order to have square summable error tolerances, more and more inner loops (e.g., SGD or other solvers) are needed to solve these subproblems. This would make the total stochastic gradient complexity to be quite high. Note that what's presented in theorems 3.1 and 4.1 is the outer iteration complexity.
>
> * Response:
> We agree with the reviewer that to achieve $\mathcal{O}(\varepsilon^{-2})$ complexity bound, we need to choose the inner tolerances as stated in Theorem A.1. Clearly, we can choose $\epsilon_{i,k} = \frac{C\epsilon}{(k+1)^s}$ for some $C > 0$ and $s > ½$. In this case, we can still guarantee the conditions of Theorem A.1. but have the flexibility to choose $C$ and $s$. Other choices of $\epsilon_{i,k}$ may work, but the complexity bound could be worse than $\mathcal{O}(\varepsilon^2)$. Therefore, trading-off the inner accuracies $\epsilon_{i,k}$ and $\epsilon$ is an interesting question to consider.
> However, we highlight that our $\mathcal{O}(\varepsilon^{-2})$ complexity is the communication complexity.
>
> 3. Reviewer's main concern 3:
> A possible remedy is to apply warm-start on each subproblem and only solve it for a fixed number of iterations. Then, there will be a "bounded relative error", which will not hurt the overall 1/K convergence rate.
> This idea has been applied in the following analysis of inexact preconditioned PDHG (or equivalently, ADMM). More specifically, please refer to Theorem 4 of this paper: Liu, Yanli, Yunbei Xu, and Wotao Yin. "Acceleration of Primal–Dual Methods by Preconditioning and Simple Subproblem Procedures." Journal of Scientific Computing 86.2 (2021): 1-34.
> Since the analysis in the above paper also depends on a Lyapunov function, which looks quite similar to this work (Lemma A.5 and Lemma B.1). I think that this may help improve the overall stochastic gradient complexity of this work.
>
> * Response:
> We completely agree with the reviewer.  One can apply a warm-start strategy to quickly guarantee the condition on the tolerance of Theorem A.1. Our initial submission did not focus on the inner loop, which requires  solving strongly convex problems; we have not presented a specific method to evaluate $\mathrm{prox}_{\eta f_i}$.
>
>      Following the reviewer suggestion, we will cite this paper and investigate how to incorporate the idea of Liu et al paper to obtain a corollary on the overall complexity when a warm-start strategy is used together with a specified method for evaluating $\mathrm{prox}_{\eta f_i}$.
>
> 4. Reviewer's main concern 4:
> In the experiments, it should also be mentioned how many inner loops of SGD are applied in FedDR, asyncFedDR, as well as other algorithms. And a comparison of stochastic gradient complexity should be provided, in addition to the communication complexity presented.
>
> * Response:
> Thank you for your concern.
> We will add this information to the experiments as suggested.
>
> 5. Reviewer's minor concern 1:
> The analysis also looks similar to those of ARock ([33] in this paper), where the bounded delay is also assumed and a Lyapunov type analysis is done, would you please provide some comparisons?
>
> * Response: For Algorithm 2, our Lyapunov function has a tail part related to the delay as in ARock [33] (or a much earlier work by Tseng (1991)), but our analysis is fundamentally different from ARock. ARock is for the convex case (or monotone operator), and its analysis relies on standard techniques widely used in the literature (e.g., [1]) combined with a new Lyapunov function to handle the delay and a standard randomized coordinate strategy. Similar works include [7, 8].
>
>     Compared to the convex case, including ARock, we face the following difficulties when dealing with nonconvex problems:
>
>     (a). The choice of metric to form a Lyapunov function. Unlike using $\Vert x^k - x^{*}\Vert^2$ or the objective residual $F(x^k) - F^{\star}$ as in convex optimization methods, we use a so-called Douglas-Rachford envelope [38] (see (25)) as the first term of our Lyapunov function.
>
>      (b). The role of $f$ and $g$ is different due to the nonconvexity in $f$. We cannot switch $f$ and $g$ in our Douglas-Rachford splitting scheme as in the convex case, where both are convex. This creates some challenges for analysis in the randomized case.
>
>      (c). If we apply ARock or other randomized coordinate methods, including [7, 8] to the FL model (1) in the convex case, then the randomized coordinate step on $f$ can be done after the full proximal step on $g$. However, in the nonconvex case, we need to apply the randomized coordinate step on $f$ first before having the full proximal step on $g$. This makes the analysis more challenging. So far, we do not know how to develop such a version where we can apply the proximal step on $g$ first and then on $f$ next in the nonconvex case.
>
> 6. Reviewer's minor concerns 2 and 3:
>
>    (2). On line 63, [nonsmooth] seems to be a typo.
>
>    (3). On line 79, early should be earliest.
>
> * Response: We will fix these issues.

---

> > ### Comment · Reviewer_A2nV · 2021-08-24
> > **Concerns are addressed**
> >
> >
> > After reading the detailed rebuttal, I believe that my concerns have been sufficiently addressed. Specifically, I think the warm-start strategy should work when combined with the proposed algorithm since the subproblem of evaluating \prox_{\eta f_i} will be strongly convex when \eta is small. I think that the bounded relative error will result in an additional term of \|x^{k+1} -x ^k\|^2 in the Lem A.5 of this work. This would make FedDR practical.
> >
> > The presentation will also be improved when the full computation complexity comparisons and comparison with ARock are added.
> >
> > I would like to raise my score to 7.

---

> > > ### Author Response · Authors · 2021-08-24
> > > **Thank you!**
> > >
> > > Thank you very much for your feedback. We will take your comments and suggestions into account in the next version.
> > > We highly appreciate your score increase.

---

### Official Review · Reviewer_ZGGt · 2021-07-16

**Rating:** 6
**Confidence:** 3

**Summary:**

This paper presents a new method for federated learning with a nonconvex but smooth objective function. The proposed method is based on Douglas Rachford Splitting to decompose the objective and randomized coordinate methodology to allow a random selection of users to update at each iteration. An extension to asynchronous settings is presented, allowing modeling of communication delays and similar practical considerations. The convergence of these methods (potential using inexact proximal calculations) is bounded by the typical O(\epsilon^-2) rate. Reasonable numerics are given showing the potential effectiveness of such a scheme on synthetic and real datasets.

**Limitations And Societal Impact:**

The paper discusses the important challenges to federated learning in terms of privacy concerns and homogeneity assumptions upfront. The work makes some modest progress towards designing methods around these societal impacts but is primarily focused on computational advances rather than societal ones.

**Main Review:**

-The use of $G_\beta(\tilde x)$ in Definition 2.2 as a stochastic quantity is confusing. In (4), $G_\beta(x)$ is defined deterministically. Is this definition meaning that a stochastic approximation of $G_\beta(x)$ generated by some algorithm is small? Or is $\tilde x$ not a point in the domain of $F$ but rather a random variable that tends to have small $G_\beta(\tilde x)$?

-The intuition for each $x_i$ and $y_i$ used in FedDR rely on the series of reformulations presented in Appendix A. Including some interpretation of these quantities in the main text would help readability.

-It is interesting that the different components do not need to be sample uniformly in Assumption 3.1. Some intuition on how varied $p_i\neq p_j$ is handled without leading to bias would be helpful.

-Deferring the full statement of convergence results to the appendix is disappointing. Guarantees under inexact proximal evaluations are one of the works claimed contributions but do not appear anywhere in the main text.
In particular, a full description of the cost of an inexact method would include the cost of the SGD subroutine employed to approximate each strongly convex subproblem. This cost seems particularly important to include since the $\epsilon_{i,k}$ choice recommended in the appendix is quite small (namely, $\epsilon/(k+1)$). Including such costs would likely lead to a runtime worse than the claimed $O(\epsilon^-2)$ rate.

**Time Spent Reviewing:**

5

---

> ### Author Response · Authors · 2021-08-08
> **Response to Reviewer ZGGt's comments**
>
> Thank you so much for your constructive comments. Below is our detailed response to each comment.
>
> 1. Reviewer's comment 1:
> The use of  $G_{\beta}(\tilde{x})$ in Definition 2.2 as a stochastic quantity is confusing. In (4), $G_{\beta}(x)$ is defined deterministically. Is this definition meaning that a stochastic approximation of $G_{\beta}(x)$ generated by some algorithm is small? Or is $\tilde{x}$ not a point in the domain of F but rather a random variable that tends to have small $G_{\beta}(\tilde{x})$?
>
> * Response:
> The formula (4) introduces the gradient mapping of (1) in general, and it is independent of algorithms. If the algorithm is stochastic, then the convergence guarantee can be in expectation (or almost surely, etc.)
>
>     Here, since $\bar{x}^k$ generated by Algorithm 1 or 2 is random due to Step 3, $G_{\beta}(\bar{x}^k)$ is also random. Theorem 3.1. or Theorem 3.2. states that the average (after $K$ iterations) of the expectation of the squared norm of gradient mapping is small.
>     Now, if we pick an output $\tilde{x}^K$ uniformly at random between $\{ \bar{x}^1, \cdots, \bar{x}^K \}$, then $ \Vert G_{\beta}(\tilde{x}^K) \Vert^2 $ satisfies the approximate optimality condition as defined in Definition 2.2. This guarantee is common in the literature to guarantee an approximate stationary point in randomized methods, see, e.g., [45].
>
>     When the computation of $\mathrm{prox}_{\eta f}$ is exact, $\bar{x}^k$ is always in the domain of $F$, though it is random. If $\bar{x}^k$ is approximately computed, then we need to make sure that $\bar{x}^k$ is in dom $F$. If $\mathrm{dom}(F)$ is not the entire space, then one may need an additional projection to guarantee $\bar{x}^k$ in $\mathrm{dom}(F)$.
>
> 2. Reviewer's comment 2:
> The intuition for each $x_i$ and $y_i$ used in FedDR rely on the series of reformulations presented in Appendix A. Including some interpretation of these quantities in the main text would help readability.
>
> * Response:
> Thank you very much for your suggestion. We will try to convey the main idea and describe the key steps of our algorithms in the main text in the next version.
>
> 3. Reviewer's comment 3:
> It is interesting that the different components do not need to be sample uniformly in Assumption 3.1. Some intuition on how varied pi≠pj is handled without leading to bias would be helpful.
>
> * Response:
> Our intention is to make Algorithm 1 flexible by allowing one to choose different sampling schemes, not necessarily uniformly. However, our sampling scheme must satisfy Assumption 3.1 guaranteeing that every user must have a non neglectable probability to be updated.
> Our bound in Theorem 3.1. depends on the lower bound probability in Assumption 3.1. In the uniform case, this probability is equal between users. Therefore, we believe that Assumption 3.1. allows us to capture the system heterogeneity between users, which is usually the case in federated learning. This observation motivates us to use a general probability distribution.
> Following your suggestion, we will add some discussion on the non-uniform case (i.e., $p_i\neq p_j$)  in the next version.
>
> 4. Reviewer's comment 4:
> Deferring the full statement of convergence results to the appendix is disappointing. Guarantees under inexact proximal evaluations are one of the works claimed contributions but do not appear anywhere in the main text. In particular, a full description of the cost of an inexact method would include the cost of the SGD subroutine employed to approximate each strongly convex subproblem. This cost seems particularly important to include since the ϵi,k choice recommended in the appendix is quite small (namely, $\epsilon/(k+1)$). Including such costs would likely lead to a runtime worse than the claimed $O(\varepsilon^{−2})$ rate.
>
> * Response:
> Thank you for bringing this concern to us. Following your suggestion, we will state our results on the inexact case in the main text in the next version. We should clearly mentioned that our complexity stated in Theorem 3.1. and Theorem 3.2. is the communication complexity, not the sample or oralce complexity.
>
>      In fact, our recommendation for the inner accuracy $\epsilon_{i,k} = \epsilon/(k+1)$ is only one choice to achieve the same order $O(\varepsilon^{-2})$ complexity as the reviewer mentioned. However, other choices are possible. More precisely, to obtain the  $O(\varepsilon^{-2})$ complexity bound in Theorem A.1, we require $\frac{1}{n}\sum_{i=1}^n\sum_{k=0}^{K+1}\epsilon_{i,k}^2 \leq M$, and other choices of $\epsilon_{i,k}$ still guarantee this condition. If this condition is violated, we can still get a convergence guarantee, but the complexity may be worse than $O(\varepsilon^{-2})$.
>
>     While our paper does not focus on the evaluation of $\mathrm{prox}_{\eta{f_i}}$, but following your suggestion, we will add a corollary to specify the overall oracle complexity when a specific subsolver (e.g., SGD) is applied to evaluate the proximal operator of $f_i$.

---

> > ### Comment · Reviewer_ZGGt · 2021-08-18
> > **Reply to Authors**
> >
> > Thank you for your careful reply clarifying the uncertainties detailed above. My score remains unchanged.
> >
> > Lastly, on the very minor point specifying the above concern about G_\beta(x) being stochastic. I believe the current writing is inconsistent and adding clarifying the language/notation would be beneficial. For example, assumption 2.2 uses x\in \dom f_i to introduce a point in this space, whereas Definition 2.2 uses x\in \dom F to introduce a random variable mapping into this space. There is no tell in the current notation to distinguish which quantities are points and which are random variables.

---

> > > ### Author Response · Authors · 2021-08-18
> > > **Response to the comment**
> > >
> > > Thank you very much for your additional comment.
> > > We will check and fix the inconsistence as indicated.
> > >
> > > For your information, we would like to clarify that $\mathrm{dom}{F} = \cap \mathrm{dom}f_i \cap\mathrm{dom}g$. In Assumption 2.2., we impose a condition on each $f_i$ only and do not touch $g$. Therefore, we only use $\mathrm{dom}f_i$. While, in Definition 2.2., we must use $\mathrm{dom}{F}$ since $x$ is now  related to the entire problem (1).
> > >
> > > However, we will carefully check other places to fix any possible inconsistence.
> > > Please let us know if you find any response that did not fully address your concerns.

---

### Official Review · Reviewer_irs2 · 2021-07-19

**Rating:** 6
**Confidence:** 4

**Summary:**

The authors present an application of Douglas-Rachford for consensus optimization in federated learning. Their algorithm, unlike in prior work, explicitly handles subsampled clients, bounded delays, and also provides guarantees under for hypoconvex problems (smooth but not necessarily convex).


**Limitations And Societal Impact:**

See the questions & comments section above.

**Main Review:**


The authors present a version of Douglas-Rachford that supports bounded delays and client subsampling for federated optimization. The techniques (stochastic approximation of the DR operator via client subsampling, and dealing with bounded delays) are more or less standard, and the results are not surprising.

Questions/comments:
- The authors claim that FedSplit and FedPD do not allow for sub-sampling among clients (e.g. ll. 53-54, ll 66-67, of the submission). However, it is unclear to me that FedSplit and FedPD are non-convergent under client subsampling; are the authors claiming that if these algorithms are modified to allow random sub-selection of devices/agents/users/clients on each round, that these algorithms will not converge?
- ll.103-104 is technically incorrect, FedSplit is an application of Peaceman-Rachford, not Douglas-Rachford; although similar, the latter is an "averaged" version of Peaceman-Rachford.
- Can the authors comment on the convergence properties of FedDR with random subsampling of the clients/users under more standard complexity assumptions (e.g. F is convex (but not necessarily strongly convex) and smooth / F is strongly convex and smooth; in the latter case, FedDR should be able to attain linear convergence in expectation)?
- It would be useful to state a corollary of Theorem 3.1 in the case alpha = 1, eta = c/L, and p_i = 1/n (uniform sampling), which corresponds to standard Douglas-Rachford with uniformly sampling and the optimal selection of the step-size.

**Time Spent Reviewing:**

1 hour

---

> ### Author Response · Authors · 2021-08-08
> **Response to Reviewer irs2's comments**
>
> Thank you so much for your constructive comments. Below is our detailed response to each comment.
>
> 1. Reviewer's comment on the contribution: The authors present a version of Douglas-Rachford that supports bounded delays and client subsampling for federated optimization. The techniques (stochastic approximation of the DR operator via client subsampling, and dealing with bounded delays) are more or less standard, and the results are not surprising.
>
> * Response: We believe that our results are new and significant. As we explained in the response point 4 below, the technical analysis of Algorithms 1 and 2 is quite challenging and fundamentally different from existing works, especially [7,8,33] for the convex case.
>
>     For Algorithm 2, our Lyapunov function has a tail part related to the delay as in ARock in [33] (or a much earlier work by Tseng (1991)), but our analysis is fundamentally different from ARock. ARock is for the convex case (or root finding of monotone operators), and its analysis relies on standard randomized block coordinate methods widely used in the literature combined with a new Lyapunov function to handle the delay. Similar works include [7, 8].
> Compared to the convex case, including ARock, we face the following difficulties when dealing with nonconvex problems:
>
>      (a). The choice of metric to form a Lyapunov function. Unlike using $\Vert x^k - x^{*}\Vert^2$ as in ARock or the objective residual $F(x^k) - F^{\star}$ in convex optimization methods, we use a so-called Douglas-Rachford envelope [38] (see (25)) as the first term of our Lyapunov function.
>
>      (b). The role of $f$ and $g$ is different due to the nonconvexity in $f$. We cannot switch $f$ and $g$ in our Douglas-Rachford splitting scheme as in the convex case, where both are convex. This creates some challenges for analysis in the randomized coordinate case.
>
>      (c). If we apply ARock or other randomized coordinate methods, including [7, 8], to the FL model (1) in the convex case, then the randomized coordinate step on $f$ can be done after the full proximal step on $g$. However, in the nonconvex case, we need to apply the randomized coordinate step on $f$ first before having the full proximal step on $g$. This makes the analysis more challenging. So far, we do not know how to develop a version where we can apply the proximal step on $g$ first and then on $f$ next in the nonconvex case.
>
>
> 2. Reviewer's comment 1:
> The authors claim that FedSplit and FedPD do not allow for sub-sampling among clients (e.g. ll. 53-54, ll 66-67, of the submission). However, it is unclear to me that FedSplit and FedPD are non-convergent under client subsampling; are the authors claiming that if these algorithms are modified to allow random sub-selection of devices/agents/users/clients on each round, that these algorithms will not converge?
>
> * Response:
> We are sorry for such confusing statements. We did not intend to claim that “FedSplit and FedPD do not allow for sub-sampling among clients” in our paper.
>
>     Up to the time our paper was submitted to NeurIPS, both FedSplit and FedPD were only developed for full client updates. We do not know if they still have convergence guarantee under client subsampling setting. We also do not know if they can be modified to take sub-selection of users, where the convergence guarantee still holds. Both original papers for FedSplit [32] and FedPD [47], respectively, did not present such a variant, at least in terms of convergence guarantees. They only presented the convergence guarantees for the full participation of users.
>
>     We will revise our text mentioned by the reviewer above to avoid any confusion.
>
>
> 3. Reviewer's comment 2:
> ll.103-104 is technically incorrect, FedSplit is an application of Peaceman-Rachford, not Douglas-Rachford; although similar, the latter is an "averaged" version of Peaceman-Rachford.
>
> * Response:
> Thank you for your comment. You are absolutely right. We will update this point in our next version.
>
>
> 4. Reviewer's comment 3:
> Can the authors comment on the convergence properties of FedDR with random subsampling of the clients/users under more standard complexity assumptions (e.g. F is convex (but not necessarily strongly convex) and smooth / F is strongly convex and smooth; in the latter case, FedDR should be able to attain linear convergence in expectation)?
>
> * Response:
> We believe that developing variants for the convex case is less challenging than that of  the nonconvex case as we do in this paper based on the following reasons:
>
>      (a). First, when f and g are convex in the FL model (1), we can switch them so that the randomized step on f is performed after the proximal step on g. Therefore, the analysis in existing works, e.g., ARock [33] and [7,8] can be utilized to obtain convergence guarantees. This is not the case in the nonconvex setting. In our paper, we must apply the randomized coordinate proximal step to $f$ first, and then the full proximal step to $g$. Since the randomized coordinate step is applied to three iterates $y^k_i$, $x^k_i$, and $\hat{x}_i^k$ simultaneously, we do not know if existing results can be applied to derive convergence. We instead use a different approach and develop several technical lemmas as in the Supp. Doc.
>
>      (b). Second, the analysis for the convex case relies on a different metric and techniques (see [7,8,33]), and we do not know if they can be applied to the nonconvex case due to the lack of convexity/monotonicity.
> At this point, developing variants for the convex case is out of scope of this paper since it will be based on a different approach and techniques. We believe that one can specify the methods in [7,8] and [33] to the federated learning models, and the theory in these papers can be applied to obtain the corresponding convergence guarantees.
>
> 5. Reviewer's comment 4:
> It would be useful to state a corollary of Theorem 3.1 in the case $\alpha = 1$, $\eta = c/L$, and $p_i = 1/n$ (uniform sampling), which corresponds to standard Douglas-Rachford with uniformly sampling and the optimal selection of the step-size.
>
> * Response:
> Thank you very much for a great suggestion. We will add this particular case to the next version.

---

> > ### Author Response · Authors · 2021-08-26
> > **Discussion**
> >
> > Dear Reviewer irs2:
> >
> > Since the deadline of discussion is approaching, we would like to see if our response to your comments is sufficient.
> > Would you mind taking some time to go over our response? Please let us know if you still have concerns or further comments.
> > We hope to communicate closely with you to improve our paper based on your constructive comments and suggestions.
> >
> > We understand that your comments and judgements are greatly important to us.
> > We really appreciate if you can consider to re-evaluate our work given its contribution and the significant difference between ours and existing work.
> >
> > Thank you very much for your time and effort.
> >
> > Sincerely,
> >
> > The authors

---

> > > ### Comment · Reviewer_irs2 · 2021-08-31
> > > **Response to authors**
> > >
> > > I have read the authors' response. While I agree that there are certain technical innovations in this work, the idea of the Douglas Rachford Envelope is not a new one -- and is in particular something that has been employed in prior works analyzing DR type algorithms under non-convex (e.g. hypo convex) problems.
> > >
> > > I continue to believe comparisons to the convex case are particularly important for this community (e.g. many ML-oriented researchers are much more familiar with convex optimization theory).
> > >
> > > As a result, I have decided to keep my score.

---

### Decision · Program_Chairs · 2021-09-27

**Decision:**

Accept (Poster)

**Comment:**

Federated learning algorithms based on operator splitting or ADMM are more sophisticated than simple algorithms such as FedAvg or FedProx, and they bring more flexibility and rigor in algorithm design. The proposed FedDR method and its asynchronous versions extends the classical Douglas-Rachford splitting method, combining with randomized block-coordinate updates, to the federated learning setting with nonconvex loss functions.

The idea of applying operator splitting methods to federated learning has been proposed in a few recent works, but the reviewers are supportive of the generality and technical results obtained in this paper. They make solid contribution to the Federated Learning literature.